# Capacitated Fair-Range Clustering: Hardness and Approximation Algorithms

Ameet Gadekar [* 1]    Suhas Thejaswi [* 2]

## Abstract

Capacitated fair-range $k$-clustering generalizes classical $k$-clustering by incorporating both capacity constraints and demographic fairness. In this setting, data points are categorized as clients and facilities; each facility has a capacity and may belong to one or more possibly intersecting demographic groups. The task is to select $k$ facilities as centers and assign each client to a center so that: $(a)$ no center exceeds its capacity, $(b)$ the number of centers selected from each group lies within specified lower and upper bounds (fair-range constraints), and $(c)$ the clustering cost (e.g., $k$-median or $k$-means) is minimized.

In a prior work, Thejaswi et al. (2022) showed that even satisfying fair-range constraints is NP-hard, thereby making the problem inapproximable to any polynomial factor. Our first main result strengthens this by showing that inapproximability persists even when the fair-range constraints are trivially satisfiable, highlighting the intrinsic computational complexity of the clustering task itself. These inapproximability results hold even on tree metrics and when the number of groups is logarithmic in the size of the facility set.

In light of strong inapproximability results, we focus on a practical setting where the number of groups is constant. Our second main result is a polynomial-time $O(\log k)$- and $O(\log^2 k)$-approximation algorithm for $k$-median and $k$-means objectives, respectively, in this regime. Next, we design constant factor approximation algorithms for these problems that run in fixed parameterized tractable time in $k$. All our approximation guarantees match the best bounds for capacitated clustering without fair-range con-

straints. Finally, as our third main contribution, we show that our polynomial-time algorithms are, to our knowledge, the first to have provable approximation guarantees that can practically solve problem instances of modest size.

## Funding and Acknowledgments

Suhas Thejaswi acknowledges support from the Technology Industries of Finland Centennial Foundation through a grant awarded to Aalto University. The authors declare that the funding source does not create any conflict of interest in relation to this work. Part of this work was carried out while Suhas Thejaswi was affiliated with the Max Planck Institute for Software Systems.

## 1. Introduction

Clustering is the task of partitioning a set of data points into clusters by choosing $k$ representative points, known as cluster centers (or simply centers, when the context is clear), and assigning each data point to a center to form a clustering solution. The quality of a clustering solution is typically measured using clustering cost, most commonly defined by the $k$-median (or $k$-means) objective, where the goal is to minimize the sum of (squared) distances between each data point and its assigned center. In a more general setting, data points are distinguished as clients and facilities—which may or may not overlap—with a constraint that cluster centers must be chosen from the set of facilities. Further, in capacitated clustering, each facility is also associated with a capacity that limits the number of clients that can be assigned to it. Here, the task is to choose $k$ centers and assign clients to centers in a way that does not exceed their capacities, while minimizing the clustering cost.

In real-world applications, data points can be associated with attributes such as sex, education level, or language skills, forming possibly intersecting groups corresponding to these attributes. In this setting, and with the growing focus on algorithmic fairness, clustering problems that require selecting centers from different demographic groups have been studied under the umbrella of fair clustering (Chhabra et al., 2021). This line of work[1], specifically addressing cluster center fairness, has introduced several problem for-

---
[*]Equal contribution  [1]CISPA Hemholtz Center for Information Security, Saarbrüken, Germany  [2]Aalto University, Espoo, Finland. Correspondence to: Ameet Gadekar <firstname.lastname@cispa.de>, Suhas Thejaswi <firstname.lastname@aalto.fi>.

*Proceedings of the 43rd International Conference on Machine Learning*, Seoul, South Korea. PMLR 306, 2026. Copyright 2026 by the author(s).

---
[1]See Appendix A for a review of further related work.

mulations that impose lower bounds, upper bounds, or equality constraints on the number of centers chosen from each group (Hajiaghayi et al., 2012; Krishnaswamy et al., 2011; Kleindessner et al., 2019; Thejaswi et al., 2021; 2022) A more general variant, known as fair-range clustering that has both lower and upper bounds on the number of centers chosen from each group (Hotegni et al., 2023; Zhang et al., 2024a; Thejaswi et al., 2024). While prior efforts have primarily focused on fairness in uncapacitated settings, many real-world applications often impose capacity limitations for cluster centers, which is the focus of our work.

To further motivate the relevance of studying this setting, consider a university mentorship initiative to support incoming students from diverse academic, socioeconomic, and cultural backgrounds. The program aims to assign each student (client) to a mentor (facility) who will serve as their primary point of contact for guidance. Each mentor has a limited capacity—they can support only a fixed number of students due to time constraints—and mentors belong to one or more demographic groups—*e.g.*, based on sex, country of origin, or academic discipline—forming possibly intersecting groups. To ensure the program to be effective, the university should solve a clustering task: ($i$) assigning students to mentors based on shared academic goals or proximity in fields of study (minimizing a clustering objective), ($ii$) respecting mentor capacity limits, and ($iii$) ensuring diversity in mentor selection—*e.g.*, ensuring representation from women, international faculty, or underrepresented scientific disciplines.

This example highlights a broader class of real-world (clustering) problems where diversity, capacity, and proximity must all be considered when designing algorithmic decision-support systems. Such problems can be formalized as the *capacitated fair-range clustering* problem, where the goal is to select $k$ centers from a set of facilities and assign each client to a center such that the number of clients assigned to each center does not exceed its capacity (*capacity constraints*), and ensure that the number of centers selected from each group lies within specified lower and upper bounds (*fair-range constraints*). The clustering objective can be $k$-median or $k$-means, resulting in the *capacitated fair-range $k$-median* or *capacitated fair-range $k$-means* problem.

To place the capacitated fair-range clustering problem in context, we first review known complexity results for fair-range clustering in the uncapacitated setting. In light of the growing interest in fair clustering, there has been remarkable progress in understanding the computational complexity as well as the design of algorithms for these problems, both in polynomial-time and fixed-parameter tractable[2] (FPT)

setting.[3] When the groups are disjoint, polynomial-time approximation algorithms are known for fair-range clustering (Hotegni et al., 2023). However, when the groups intersect, Thejaswi et al. (2021) showed[4] that the problem is inapproximable to any multiplicative factor. A key insight in their result is that, with intersecting groups, even satisfying the fair-range constraints becomes NP-hard regardless of the clustering objective being optimized. As a consequence, the fair-range $k$-median ($k$-means) problem is inapproximable to any multiplicative factor, both in polynomial-time and in $\mathsf{FPT}(k)$-time, even for structured inputs such as Euclidean and tree metrics. Naturally, these results extend to the capacitated variants of these fair-range clustering problems, as they capture the corresponding uncapacitated versions.

While their inapproximability result is significant, it clearly falls short of capturing the true complexity of the underlying clustering task, as it focuses solely on the hardness of satisfying the fair-range constraints. In practice, there exist instances—including those with intersecting groups—where a feasible solution (one that satisfies fair-range constraints) can be found efficiently (or in polynomial-time). For example, a simple greedy strategy that selects facilities covering the most constraints may produce a feasible solution. However, such solutions can be arbitrarily far from being optimal in terms of the clustering cost. To further strengthen the complexity landscape of this problem as well as its capacitated variant, we ask:

> **Question:** *Is it possible to approximate the (capacitated) fair-range clustering problem when feasible solutions can be found in polynomial-time?*

In this work, we answer this question negatively even for the uncapacitated version, revealing the intrinsic hardness of the underlying clustering problem. Moreover, for the capacitated variant, we identify instances that are of practical interest and bypass the above hardness result, and design polynomial-time and $\mathsf{FPT}(k)$-time approximation algorithms. In detail, our contributions are as follows.[5] We use $n$ to denote the number of data points in the instance.

**Hardness of Approximation.** We strengthen the inapproximability landscape by showing that the hardness does not

---

[2] A parameterized problem $P$ is fixed-parameter tractable (approximable) with respect to a parameter $k$, if there exists an algorithm that for any instance $(x, k) \in P$ computes an exact (approximate) solution in time $f(k) \cdot |x|^{\mathsf{O}(1)}$, for some computable function $f$. We denote by $\mathsf{FPT}(k)$ for such running times.

[3] In prior work, the hardness and approximation results are given for uncapacitated variants, whereas our results cover both capacitated and uncapacitated settings.

[4] In fact, their reduction produces instances with lower-bound only requirements. However, our results can be extended to produce instances with lower-bound only requirements. See Appendix B for details.

[5] All proofs are available in the Appendix.

arise solely from the complexity of satisfying the fair-range constraints. Specifically, we prove that the fair-range $k$-median (and $k$-means) problem remains NP-hard to approximate to any polynomial factor, even when feasible solutions can be found in polynomial-time. While our inapproximability factor matches that of Thejaswi et al. (2021), our result is fundamentally stronger, as the hardness arises from the underlying clustering task itself (see Theorem 3.2 for a precise statement). Since capacitated variants generalize their uncapacitated counterparts, our inapproximability results naturally extend to the capacitated setting. We further strengthen our hardness result in two ways. First, observe that any feasible solution, which can be found efficiently in this case, is a $\Delta$ (or $\Delta^2$) approximate solution for fair-range $k$-median (or $k$-means), where $\Delta$ is the distance aspect ratio of the instance.[6] In stark contrast, we show that this factor is essentially optimal under $\mathsf{P} \neq \mathsf{NP}$ conjecture (see Theorem 3.3 for details). Next, assuming Gap-ETH,[7] we show a stronger result (see Theorem 3.4): there is no $n^{o(k)}$-time algorithm that can approximate the (capacitated) fair-range $k$-median (or $k$-means) problem to any polynomial factor, even when feasible solutions can be found in polynomial-time. Note that the trivial brute-force algorithm, which enumerates all $k$-tuples of facilities, runs in time $n^{O(k)}$. Our hardness result implies that this is essentially the best possible—even when seeking only an approximate solution. Furthermore, our inapproximability result holds even when the number of groups is logarithmic in the size of the facility set, and also on tree metrics. To the best of our knowledge, this hardness result was previously unknown.

**Polynomial-time approximation algorithms.** In light of strong inapproximability results, we turn our attention to identifying instances, for which we can obtain non-trivial approximations in the capacitated setting. One regime that bypasses the above theoretical hardness barrier, and is simultaneously of practical interest is when the number of groups is constant. This setting has been extensively studied in prior work (Kleindessner et al., 2019; Thejaswi et al., 2021; 2022; Zhang et al., 2024b).

For a constant number of groups, we present $O(\log k)$- and $O(\log^2 k)$-approximation algorithms for the $k$-median and $k$-means objectives, respectively (see Theorem 4.1). Our algorithms run in polynomial-time and match the best-known approximation factors for their non-fair counter-

parts (Charikar et al., 1998). Our approach relies on embedding the original instance into a tree metric, followed by solving the problem exactly on the tree using dynamic programming. Such tree embeddings are well studied (Bartal, 1996; 1998; Fakcharoenphol et al., 2004) and have been applied to obtain approximation algorithms for clustering problems (Charikar et al., 1998; Bartal, 1998; Adamczyk et al., 2019), among other optimization problems. However, naively embedding all data points into a tree yields a $O(\log n)$-approximation (or $O(\log^2 n)$, respectively), since these embeddings incur $O(\log n)$ distortion in the distances. To address this, we build upon the embedding framework of (Adamczyk et al., 2019), which obtains similar approximation guarantees for the capacitated setting without fairness constraints. However, directly applying their techniques faces challenges in satisfying the fairness constraints for the selected centers. In this work, we show how this framework can be adapted to our setting. Additionally, we design a simple dynamic program for the problem in tree metrics, which is a crucial component of our algorithm.

*Constant-factor* $\mathsf{FPT}(k)$-*approximation algorithms.* We also present constant-factor approximation algorithms in the $\mathsf{FPT}$ regime with respect to parameter $k$, the number of centers. While our inapproximability result rules out $n^{o(k)}$-time approximation algorithms in the general setting, this hardness result no longer applies when the number of groups is constant. As our next contribution, in Theorem 4.5, we give $(3 + \epsilon)$ and $(9 + \epsilon)$-approximation algorithms, for any $\epsilon > 0$, for the capacitated fair-range $k$-median and $k$-means problems, respectively. These algorithms run in time $(O(k\epsilon^{-1} \log n))^{O(k)} \cdot n^{O(1)}$, for constant number of groups, and match the best-known approximation guarantees for their unfair counterparts (Cohen-Addad & Li, 2019). Our algorithm is based on the leader-guessing framework, which has been applied to solve several clustering problems in recent years (Cohen-Addad & Li, 2019; Cohen-Addad et al., 2019; Thejaswi et al., 2022; Zhang et al., 2024a; Chen et al., 2024), and on a simple observation that allows us to consider instances where the group requirements are exactly one.

**Experimental results.** Finally, using our implementation,[8] we demonstrate that our polynomial-time algorithms are, to our knowledge, the first to be able to solve the general variant of fair-range clustering (and its capacitated variant) with intersecting groups in practice while offering provable approximation guarantees. On synthetic data, for example, we solve instances with $n = 1600$, $t = 3$, and $k = 5$ in under 5100 seconds ($\approx$ 1.5 hours, and the empirical approximation factor is below 2.7 across all instances, despite the worst-case $O(\log k)$ bound. On real-world datasets of modest size ($n = 714$), our methods scale to $k = 10$ centers

---

[6] In a metric space $(X, d)$, the aspect ratio $\Delta$ is the ratio between the maximum and minimum pairwise distances, *i.e.*, $\Delta := \frac{d_{\max}}{d_{\min}}$, where $d_{\max} = \max_{x,y \in X} d(x, y)$ and $d_{\min} = \min_{x \neq y \in X} d(x, y)$.

[7] Roughly speaking, Gap-ETH says that there exists an $\epsilon > 0$ such that there is no $2^{o(n')}$ time algorithm that decides if the given 3-SAT formula $\phi$ on $n'$ variables has a satisfying assignment or every assignment satisfies at most $(1 - \epsilon)$ fraction of clauses of $\phi$. See Hypothesis B.9 for a precise formulation.

[8] Source code is available at https://github.com/suhastheju/fair-range-clustering.

and $t = 4$ groups in under $10^4$ seconds ($\approx 2.8$ hours) on a commodity `MacBook Air` laptop with an M2 processor.

The rest of the paper is organized as follows: Section 2 defines the problem, Section 3 presents inapproximability results, Section 4 describes our approximation algorithms, Section 5 discusses our experimental results. And, finally, Section 6 offers concluding remarks.

## 2. Problem Definition

We formally define of our problem.

**Definition 2.1** (The capacitated fair-range $k$-median (and $k$-means) problem). An instance $\mathcal{I} = (C, F, \mathbb{G}, \vec{\alpha}, \vec{\beta}, \zeta, k, t)$ of the *capacitated fair-range $k$-clustering problem* is defined by positive integers $k$ and $t$, a set $C$ of clients, a set $F$ of facilities, and a metric $d$ over $C \cup F$. Each facility in $F$ belongs to one or more demographic groups, forming possibly intersecting groups denoted by $\mathbb{G} = \{G_i\}_{i \in [t]}$. Each group $G_i$ is associated with a lower bound requirement $\alpha_i$ and an upper bound requirement $\beta_i$. The requirements are represented by vectors $\vec{\alpha} = (\alpha_i)_{i \in [t]}$ and $\vec{\beta} = (\beta_i)_{i \in [t]}$. Furthermore, each facility $f \in F$ has a capacity $\zeta : F \to \mathbb{Z}_{\geq 0}$. The task is to select a subset $S \subseteq F$ of at most $k$ facilities and find an assignment function $\rho : C \to S$ that assigns each client $c \in C$ to a facility $f \in S$, to form a clustering solution $(S, \rho)$. A solution $(S, \rho)$ is feasible if it satisfies:

- $\forall G_i \in \mathbb{G}$, the number of selected centers from $G_i$ lies within $\alpha_i$ and $\beta_i$, *i.e.*, $\alpha_i \leq |S \cap G_i| \leq \beta_i$,
- $\forall f \in S$, $f$ is assigned at most $\zeta(f)$ clients, *i.e.*, $|\{c \in C : \rho(c) = f\}| \leq \zeta(f)$.

The objective of the *capacitated fair-range $k$-median* is to find a feasible solution $(S, \rho)$ that minimizes $\text{COST}_{\mathcal{I}}(C, S) := \sum_{c \in C} d(c, \rho(c))$, while for capacitated fair-range $k$-means, the objective is to minimize $\text{COST}_{\mathcal{I}}(C, S) := \sum_{c \in C} d(c, \rho(c))^2$, over all feasible solutions $(S, \rho)$. We succinctly denote these problems as CFR$k$MED and CFR$k$MEANS, respectively.

When facilities have unlimited capacities and can serve any number of clients, the problem is referred to as the fair-range $k$-median (and $k$-means) problem and denoted succinctly as FR$k$MED (and FR$k$MEANS). When the client set $C$ is clear from context, we write $\text{COST}_{\mathcal{I}}(S)$ for $\text{COST}_{\mathcal{I}}(C, S)$; when both $\mathcal{I}$ and $C$ are clear, we use $\text{COST}(S)$. For discrete metrics, we assume that $d$ is defined by a weighted graph $H$ whose vertex set contains $C \cup F$, and where $d$ corresponds to the shortest-path metric on $H$. We say that $d$ is a tree metric if $H$ is a tree. We use $n$ to denote the size of the vertex set of $H$. For a positive integer $\kappa$, we use $[\kappa] = \{1, \ldots, \kappa\}$. We assume the distance aspect ratio, denoted as $\Delta$ (see Footnote 6 for the definition), of the metric space of

the given instance is polynomially bounded in $n$.

## 3. Hardness of Approximation

In this section, we show that even when feasible solutions can be found trivially, no polynomial-time algorithm can approximate (capacitated) fair-range clustering to any polynomial factor. To formalize this, we first define FR$k$MED$^{\mathcal{O}}$ (and FR$k$MEANS$^{\mathcal{O}}$) as those instances of FR$k$MED (and FR$k$MEANS, resp.) which admit a polynomial-time algorithm for finding feasible solutions. For brevity, we formally define FR$k$MED$^{\mathcal{O}}$, and similar definition follows for FR$k$MEANS$^{\mathcal{O}}$.

**Definition 3.1** (FR$k$MED$^{\mathcal{O}}$). Consider an instance $\mathcal{I} = (C, F, \mathbb{G}, \vec{\alpha}, \vec{\beta}, \zeta, k, t)$ of FR$k$MED, and let $\mathcal{F}(\mathcal{I})$ be the set of feasible solutions to $\mathcal{I}$. We say the instance $\mathcal{I}$ is *oracle-feasible* if there exists an (instance specific) oracle $O_{\mathcal{I}}$ that runs in polynomial time and outputs some feasible solution in $\mathcal{F}(\mathcal{I})$.[9] The FR$k$MED$^{\mathcal{O}}$ problem, given an instance $\mathcal{I}$ of FR$k$MED with a promise that $\mathcal{I}$ is oracle-feasible, asks to find a feasible solution $(S, \rho)$ from $\mathcal{F}(\mathcal{I})$ that minimizes $\text{COST}_{\mathcal{I}}(C, S) = \sum_{c \in C} d(c, \rho(c))$.[10]

Now, we are ready to state our first hardness of approximation result.

**Theorem 3.2.** *Unless* $\mathsf{P} = \mathsf{NP}$*, there is no polynomial-time algorithm that can approximate* FR$k$MED$^{\mathcal{O}}$ *(or* FR$k$MEANS$^{\mathcal{O}}$*) within polynomial factor, even on tree metrics.*

We further strengthen this result in two ways: first, we show a stronger inapproximability factor for polynomial time algorithms, and second, we rule out super polynomial time algorithms with the same inapproximability guarantee as in Theorem 3.2. For the former, note that, any feasible solution is a $\Delta$-approximation, where recall that $\Delta$ is the distance aspect ratio of $d$. In the following theorem, we show that this trivial bound is essentially optimal.

**Theorem 3.3** (Informal version of Theorem B.3). *Assuming* $\mathsf{P} \neq \mathsf{NP}$*, there is no polynomial time algorithm to approximate* FR$k$MED$^{\mathcal{O}}$ *(*FR$k$MEANS$^{\mathcal{O}}$ *resp.) within a factor of* $(\Delta - 2)/16$*, even on tree metrics.*

Next, we strengthen the running time of Theorem 3.2. Note that (capacitated) FR$k$MED and FR$k$MEANS can be exactly solved in time $n^{O(k)}$, by enumerating all $k$-sized subsets of the facility set. Our next result shows that, under Gap-ETH[7], even for finding a non-trivial approximation requires $n^{\Omega(k)}$ time, making brute-force algorithm essentially our best hope, even in this case. This also rules out any FPT$(k)$-time approximation algorithms for these problems.

---

[9]We emphasize that the oracle may be different for each instance $\mathcal{I}$.

[10]Equivalently, FR$k$MED$^{\mathcal{O}} = \{\mathcal{I} \in \text{FR}k\text{MED} \mid \mathcal{F}(\mathcal{I}) \neq \emptyset\}$.

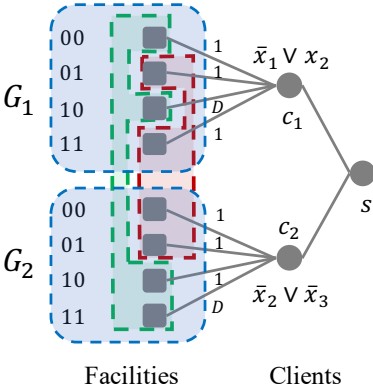

*Figure 1.* Illustration of reduction from 2-SAT formula: $(\bar{x}_1 \vee x_2) \wedge (\bar{x}_2 \vee \bar{x}_3)$.

**Theorem 3.4.** *Assuming* Gap-ETH, *there is no* $f(k) \cdot n^{o(k)}$ *algorithm, for any computable function $f$, that can approximate* FR$k$MED$^{\mathcal{O}}$ *(or* FR$k$MEANS$^{\mathcal{O}}$*) within any polynomial factor, even when the number of groups is* $\mathsf{O}(k^3 \log n)$, *and when the metric space is a tree.*

*Remark* 3.5. Our inapproximability results are stated for the range setting, where both upper and lower bounds are specified for each group. In contrast, Thejaswi et al. (Thejaswi et al., 2021) show hardness results even when only lower bounds are present, making their result appear formally stronger. However, we note that our hardness constructions can be adapted to obtain the same inapproximability guarantees under lower-bound-only constraints as well. See Appendix B.3 for details.

**Technical Overview.** Our inapproximability results are based on reductions from the 3-SAT problem to FR$k$MED$^{\mathcal{O}}$ (and FR$k$MEANS$^{\mathcal{O}}$). The high level idea is that (see Theorems B.4 and B.11), given an instance $\phi$ of 3-SAT on $n'$ variables and $m'$ clauses and $D \geq 1$, we construct an instance $\mathcal{I}$ of FR$k$MED such that $(i)$ the distance aspect ratio in $\mathcal{I}$ is $2D + 2$, $(ii)$ if $\phi$ is satisfiable, then there is a feasible solution to $\mathcal{I}$ with cost at most $k$, and $(iii)$ if every assignment satisfies at most $(1 - \epsilon)m$ clauses of $\phi$, for some constant $\epsilon > 0$, then every feasible solution to $\mathcal{I}$ has cost $\Omega(D)$. This immediately rules out $o(D)$-approximation for the problem, for arbitrary values of $D$. The main crux of our construction is that we use the group structure with requirements to create instances where $D$ can have arbitrary values, without breaking the metric property. This is in contrast with the hardness constructions for the vanilla $k$-Median ($k$-Means) problem, where we do not have such a flexibility.

In more detail, for each clause $C_i$ in $\phi$, we create a client $c_i$ and a set $F_i$ of facilities, each facility representing a possible assignment to the variables in $C_i$. We connect $c_i$ to all facilities in $F_i$, assigning edge weight 1 if the assignment satisfies $C_i$, and $D$ otherwise. Finally, we add a dummy

client $s$ and connect it to each $c_i$ with an edge of weight 1. Observe that this creates a tree metric and hence the metric property works for all values of $D$. However, note that since we are enumerating all the partial assignments of every clause, we would also like to enforce the constraint that the partial assignments corresponding to the selected facilities in the solution must be consistent. This can precisely be achieved by creating suitable groups and adding corresponding requirements. In particular, we create two types of groups—*clause groups* and *assignment groups*. For each clause $C_i$ in $\phi$, we create a clause group $G_i$ that contains all the facilities in $F_i$ and set the requirements $\alpha_i = \beta_i = 1$, to enforce the selection of exactly one partial assignment for $C_i$. To ensure consistency across assignments, we create assignment groups: for each variable $X_j$, each pair of clauses $C_i, C_{i'}$ containing $X_j$, and each assignment $a \in \{0, 1\}$, we create a group $G_{X_j \mapsto a}^{(C_i, C_{i'})}$. This group includes all facilities in $F_i$ assigning $a$ to $X_j$, and all in $F_{i'}$ assigning $1 - a$ to $X_j$, with both lower and upper bounds set to 1. Finally, we set $k = m$. The idea is that any feasible solution to this CFR$k$MED instance should correspond to a set of consistent partial assignments that allows us to obtain a global assignment. Furthermore, note that, for every assignment to the variables of $\phi$, the corresponding set of facilities form a feasible solution. Hence, we can find feasible solutions to this instance trivially. An illustration of the reduction is shown in Figure 1. Due to space constraints, we depict a reduction from a 2-SAT formula with two clauses (the construction for 3-SAT is similar): $C_1 = \bar{x}_1 \vee x_2$ and $C_2 = \bar{x}_2 \vee \bar{x}_3$. We highlight clause groups $G_1$ and $G_2$ in blue, and two assignment groups—$G_{x_2 \mapsto 0}^{(C_1, C_2)}$ and $G_{x_2 \mapsto 1}^{(C_1, C_2)}$— in green and red, respectively. The complete analysis is deferred to Appendix B.

## 4. Approximation Algorithms for Constant Number of Groups

In this section, we focus our attention towards a setting that is more practical, but simultaneously avoids the hardness results of the previous section. Specifically, we consider the problem when the number of groups is constant. Moreover, this setting has been extensively explored in the literature (*e.g.*, (Kleindessner et al., 2019; Thejaswi et al., 2021; 2022; Zhang et al., 2024b)) across various notions of fair clustering. We believe that studying the capacitated fair-range setting under this regime is both natural and promising. To this end, we present polynomial-time approximation algorithms in Section 4.1 and FPT($k$)-approximation algorithms in Section 4.2.

### 4.1. Polynomial-time approximation algorithm

In this subsection, we design polynomial-time $\mathsf{O}(\log k)$- and $\mathsf{O}(\log^2 k)$-approximation algorithms for CFR$k$MED

and CFR$k$MEANS, respectively, when the number of groups is constant. For simplicity, we focus on $O(\log k)$-approximation algorithm for CFR$k$MED. Our approach can be easily generalized to CFR$k$MEANS to obtain $O(\log^2 k)$-approximation (see Appendix C for details).

**Theorem 4.1.** *There exists a $O(\log k)$ (and $O(\log^2 k)$) approximation algorithm for* CFR$k$MED *(*CFR$k$MEANS, *resp.) that runs in $(nk^t)^{O(1)}$ time.*

At a high level, the algorithm proceeds in two steps. First, given an instance $\mathcal{I}$ of CFR$k$MED on general metrics, we embed it into a tree metric. Second, we design a polynomial-time exact dynamic program to solve CFR$k$MED on the resulting tree metric. As mentioned earlier, standard techniques (Bartal, 1998; Fakcharoenphol et al., 2004) allow embedding any metric $\mathcal{M}$ on $n$ points into a tree metric with $O(\log n)$ distortion in the distances.[11] Thus, if we can solve CFR$k$MED exactly on tree metrics, combining this with the tree embedding yields a $O(\log n)$-approximation for CFR$k$MED on general metrics. To obtain $O(\log k)$-approximation, we build on the ideas of (Adamczyk et al., 2019), who designed a $O(\log k)$-approximation algorithm for capacitated $k$-median, extending the techniques from (Charikar et al., 1998).

An overview of our approach is shown in Figure 2. In Phase 1, we embed the given instance $\mathcal{I}$ of CFR$k$MED on metric $d$ into a new instance $\mathcal{I}'$ on metric $d'$ such that $d'$ dominates $d$,[12] and has properties that enable us to obtain better approximation guarantees. We remark that instances $\mathcal{I}'$ and $\mathcal{I}$ differ only in the underlying metric. Specifically, $d'$ corresponds to the shortest-path metric on a graph, consisting of a complete graph (or clique) on $k$ nodes, and remaining $n - k$ nodes connected to exactly one node in the clique. Here $n := C \cup F$. We refer to this metric as $k$-clique-star. To construct such an embedding we make use of a polynomial-time $O(1)$-approximation algorithm $\mathcal{A}$ for $k$-Median ((Cohen-Addad et al., 2025; Arya et al., 2004; Ahmadian et al., 2019)). Below we state our result formally.

**Lemma 4.2.** *Given an instance $\mathcal{I}$ of* CFR$k$MED *on a general metric $d$, and a polynomial-time $\eta$-approximation algorithm $\mathcal{A}$ for $k$-median, we can construct, in $n^{O(1)}$ time, an instance $\mathcal{I}'$ of* CFR$k$MED *on $k$-clique-star metric $d'$ such that*

$$\text{COST}_{\mathcal{I}'}(O') \leq \text{COST}_{\mathcal{I}'}(O) \leq (4\eta + 3) \cdot \text{COST}_{\mathcal{I}}(O),$$

*where $O, O' \subseteq F$ are optimal solutions to $\mathcal{I}$ and $\mathcal{I}'$, respectively.*

The idea to build $d'$ using $\mathcal{A}$ is as follows. First, we obtain an $\eta$-approximate set $S$ to $\mathcal{I}$ using $\mathcal{A}$. Note, however, that

since $\mathcal{A}$ works only for $k$-median, the set $S$ may not be feasible (both capacity and fairness wise). In $d'$, we create a clique on the nodes of $S$ with weights of the edges being the distance between the pairs in $d$. Finally, we connect the remaining points to the closest node in $S$ with weight being the corresponding distance in $d$.

In Phase 2, we design $O(\log k)$-approximation algorithm for CFR$k$MED on $k$-clique-star metrics. Towards this, we first replace the clique of $d'$ on $k$ vertices by a tree obtained from tree embeddings mentioned earlier (Fakcharoenphol et al., 2004), to obtain a tree metric $d''$. Note that, for any pair $u, v \in C \cup F$, we have $d'(u, v) \leq d''(u, v) \leq O(\log k) \cdot d'(u, v)$, due to the guarantees of the embedding. In fact, we prove the following stronger result.

**Lemma 4.3.** *Given an instance $\mathcal{I}'$ of* CFR$k$MED *of size $n$ on a $k$-clique-star metric $d'$, we can construct, in time $n^{O(1)}$, an instance $\mathcal{I}''$ of* CFR$k$MED *on a tree metric $d''$ such that for any set $S'$ of facilities, it holds that*

$$\text{COST}_{\mathcal{I}'}(S') \leq \text{COST}_{\mathcal{I}''}(S') \leq O(\log k) \cdot \text{COST}_{\mathcal{I}'}(S').$$

Finally, we show a simple dynamic programming algorithm for CFR$k$MED on tree metrics.

**Lemma 4.4.** *There exists an exact algorithm for* CFR$k$MED *on tree metrics in $k^{2t} \cdot n^{O(1)}$ time.*

We begin by transforming the given tree into a rooted full binary tree where all clients and leaves appear at leaves using standard techniques. We then design a dynamic program over this binary tree. At a high level, the dynamic programming table $T(e, \vec{\kappa}, b)$ stores the minimum cost (and the corresponding solution) over all feasible solutions for the subtree $T_e$ rooted at edge $e$, with respect to $\vec{\kappa}$ and $b$. Here, $\vec{\kappa} = (\kappa_i)_{i \in [t]}$ specifies that the solution must open $\alpha_i \leq \kappa_i \leq \beta_i$ facilities from group $G_i$ in $T_e$, and $b \in \{-n, \ldots, +n\}$ indicates $|b|$ clients must be routed through the edge $e$ (routed out of $T_e$ when $b > 0$ and routed into $T_e$ when $b < 0$). To compute this entry, we split $\kappa$ and $b$ between left and right subtrees—connected via edges $e^\ell$ and $e^r$—such that $\vec{\kappa} = \vec{\kappa}^\ell + \vec{\kappa}^r$ and $b = b^\ell + b^r$. For each configuration $(e, \vec{\kappa}, b)$, we select the tuple $(\vec{\kappa}^\ell, \vec{\kappa}^r, b^\ell, b^r)$ that minimizes the cost and proceed in bottom-up fashion from the leaves to the root to find an optimal solution.

Combining Lemmas 4.2, 4.3 and 4.4 yields $O(\log k)$- and $O(\log^2 k)$-approximation algorithms for CFR$k$MED and CFR$k$MEANS, respectively, running in $(nk^t)^{O(1)}$ time.

### 4.2. FPT time approximation algorithm

For completeness, we present $\mathsf{FPT}(k)$-approximation algorithms for the regime with constant $t$, building upon the leader-guessing framework of Cohen-Addad et al. (2019) and subsequent work on FPT-approximations for clustering problems (Cohen-Addad & Li, 2019; Thejaswi et al.,

---

[11]The distortion is on expectation over the probabilistic embedding of $\mathcal{M}$ based on a distribution on tree metrics. However, such embeddings can be derandomized. See Appendix C for details.

[12]That is, $d(u, v) \leq d'(u, v)$, for all pairs $u, v \in C \cup F$.

**Lemma 4.2**     **Lemma 4.3**

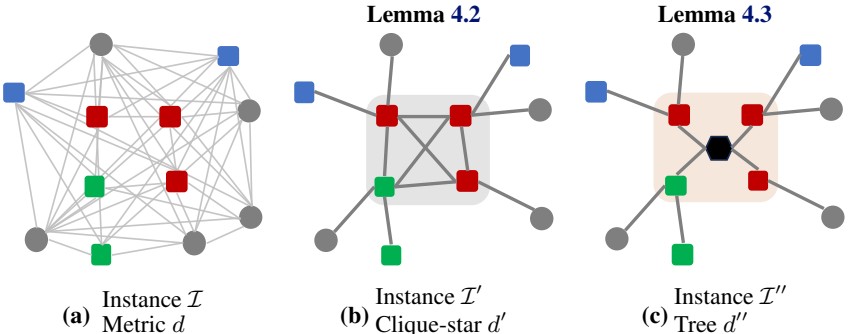

(a) Instance $\mathcal{I}$
Metric $d$

(b) Instance $\mathcal{I}'$
Clique-star $d'$

(c) Instance $\mathcal{I}''$
Tree $d''$

*Figure 2.* Overview of our algorithm for Theorem 4.1: squares represent facilities, (gray) circles represent clients, and (black) hexagons are dummy nodes introduced in the tree embedding. Colors (red, blue and green) indicate facility groups. Panel (a) shows the original instance $\mathcal{I}$ of CFR$k$MED in a general metric space $d$. Panel (b) depicts the transformed instance $\mathcal{I}'$ in $k$-clique-star $d'$ obtained from Lemma 4.2 using an $\eta$-approximation solution $S$ treating $\mathcal{I}$ as a vanilla $k$-median instance; $S$ is highlighted with shaded area. Panel (c) illustrates the instance $\mathcal{I}''$ in the tree metric $d''$ obtained from Lemma 4.3 with the tree embedding of $S$ again highlighted with shaded area.

2022; Zhang et al., 2024a; Chen et al., 2024). Though this is not a main contribution of our work, it shows that in the capacitated fair-range setting we can match the best-known approximation guarantees for capacitated clustering without fairness constraints. See Appendix D for detailed proofs.

**Theorem 4.5.** *For any $\epsilon > 0$, there exists a randomized $(3 + \epsilon)$-approximation algorithm for CFR$k$MED with running time $(\mathsf{O}(2^t k \epsilon^{-1} \log n))^{\mathsf{O}(k)} \cdot n^{\mathsf{O}(1)}$. With the same running time, a $(9 + \epsilon)$-approximation algorithm exists for CFR$k$MEANS.*

**Step 1: enumerating feasible constraint patterns.** Given an instance $\mathcal{I}$ of CFR$k$MED, we first reduce it to a collection of structured subinstances. For each facility $f$, let $\chi_f \in \{0,1\}^t$ be its group-membership vector, and group facilities $E(\gamma) = \{f : \chi_f = \gamma\}$ for $\gamma \in \{0,1\}^t$. Thus, there are at most $2^t$ disjoint facility groups. A feasible solution corresponds to a multiset $\{\gamma_1, \ldots, \gamma_k\}$ of *constraint patterns* with $\vec{\alpha} \leq \sum_{i=1}^k \gamma_i \leq \vec{\beta}$ (elementwise). We show that all such feasible patterns can be enumerated in FPT time:

**Lemma 4.6.** *For any instance of CFR$k$MED (or CFR$k$MEANS), there is a deterministic algorithm that enumerates all constraint patterns satisfying the fair-range constraints in time $2^{tk} \cdot n^{\mathcal{O}(1)}$.*

Fixing a feasible pattern $(\gamma_1, \ldots, \gamma_k)$ reduces CFR$k$MED to a one-per-group variant in which we must choose exactly one facility from each (now disjoint) group $E(\gamma_i)$. This gives an instance of the one-per-group weighted capacitated fair-range $k$-median problem (OPG-WC$k$MED$^\emptyset$), for which the optimal cost is at least that of $\mathcal{I}$. Since we enumerate all feasible patterns, one instance corresponds to the pattern in optimal solution, ensuring that our approximation factor is guaranteed.

**Step 2: coresets and leader-guessing.** To make the prob-

lem FPT in $k$, we first compress the clients via a coreset. For capacitated $k$-median / $k$-means, Cohen-Addad & Li (2019) construct weighted coresets $(W, \omega)$ of size $|W| = \mathsf{poly}(k, \epsilon_1^{-1}, \log n)$ that approximately preserve the cost of every set of $k$-centers. Since fair-range constraints only restrict which facilities can be chosen, the same coresets for clients apply in our setting.

On top of this coreset, we adapt the standard leader-guessing framework (Cohen-Addad et al., 2019). For OPG-WC$k$MED$^\emptyset$, we (i) guess a multiset of $k$ leaders $L^* = \{\ell_1^*, \ldots, \ell_k^*\} \subseteq W$ (each the closest client to a center in an optimal solution), and (ii) guess their radii corresponding $\{r_i^*\}$, discretized to $\mathsf{O}(\epsilon_2^{-1} \log n)$ values. For each leader $\ell_i^*$ and guess radius $r_i^*$, we then select exactly one facility from the corresponding group $E(\gamma_i)$ within distance $r_i^*$, while respecting capacities and avoiding duplicates. We use standard color-coding to ensure that centers in an optimal solution receive distinct colors with high probability, and then restrict attention to color-consistent facility choices. Fractional assignments on the weighted clients are finally rounded via a min-cost flow, as in prior works. This yields us the following lemma.

**Lemma 4.7.** *For any $\epsilon > 0$, there exists a randomized $(3 + \epsilon)$-approximation algorithm for OPG-WC$k$MED$^\emptyset$ with running time $\mathsf{O}(k\epsilon^{-1} \log n))^{\mathsf{O}(k)} \cdot n^{\mathsf{O}(1)}$. The same running time gives a $(9 + \epsilon)$-approximation for OPG-WC$k$MEANS$^\emptyset$.*

**Putting things together.** To prove Theorem 4.5, we enumerate all feasible constraint patterns via Lemma 4.6, transform each feasible pattern into an OPG-WC$k$MED$^\emptyset$ instance on coreset, and run the algorithm of Lemma 4.7 on each instance. We return the best solution over all patterns. Since the pattern corresponding to the optimal solution is included in one of these instances, this gives $(3 + \epsilon)$- and $(9 + \epsilon)$-approximations for CFR$k$MED and CFR$k$MEANS, respec-

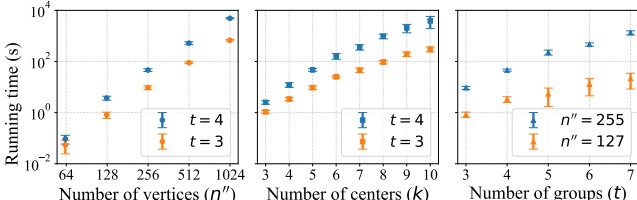

*Figure 3.* Scalability of dynamic program for computing exact solution on binary-tree metric $(U'', d'')$. The left panel shows runtime as a function of the number of vertices $n'' \in \{63, 127, \ldots, 1023\}$, with the number of centers $k = 5$ (fixed) and groups $t \in \{3, 4\}$. The middle panel shows scaling with respect to the number of centers $k \in \{3, 4, \ldots, 10\}$ for a fixed tree size $n'' = 255$ and $t \in \{3, 4\}$. The right panel shows runtime as a function of the number of groups $t \in \{3, 4, \ldots, 7\}$, with the number of centers $k = 5$ (fixed) and vertices $n'' \in \{127, 255\}$.

tively, with total running time $O(2^t k \epsilon^{-1} \log n))^{O(k)} \cdot n^{O(1)}$, which simplifies to $(O(k \epsilon^{-1} \log n))^{O(k)} \cdot n^{O(1)}$ when $t$ is constant.[13]

*Table 1.* Comparison of running time and solution quality of $O(\log k)$-approximation algorithm and the brute-force baseline on synthetic instances with number of groups $t = 3$ and centers $k = 5$ fixed. Here, $n$ and $n''$ are the number of data points in the original metric $(U, d)$ and the binary-tree metric $(U'', d'')$. Columns `Embed`, `DP`, and `BF` report the mean and standard deviation of running times (in seconds) for the embedding step, the dynamic program on the tree, and brute-force enumeration, respectively. The column (`Approx`) reports the empirical approximation ratio of our algorithm relative to the optimal solution returned by brute force. The last column $\left(\frac{\text{avg}(d'')}{\text{avg}(d)}\right)$ report the ratio of average pairwise distances in binary-tree embedding and original metric.

| | | | Running time (s) | | Embed + DP | Embed |
|---|---|---|---|---|---|---|
| $n$ | $n''$ | Embed | DP | BF | Approx | $\frac{\text{avg}(d'')}{\text{avg}(d)}$ |
| 30 | 49 | $0.1 \pm 0.0$ | $0.1 \pm 0.0$ | $3.8 \pm 2.6$ | $2.30 \pm 0.31$ | $2.40 \pm 0.24$ |
| 40 | 69 | $0.1 \pm 0.0$ | $0.1 \pm 0.0$ | $17.6 \pm 16.4$ | $2.12 \pm 0.23$ | $2.44 \pm 0.19$ |
| 60 | 109 | $0.1 \pm 0.0$ | $0.3 \pm 0.1$ | $372.3 \pm 164.4$ | $1.90 \pm 0.06$ | $2.48 \pm 0.22$ |
| 80 | 149 | $0.1 \pm 0.0$ | $0.8 \pm 0.1$ | $1769.8 \pm 639.3$ | $1.83 \pm 0.09$ | $2.36 \pm 0.22$ |
| 100 | 189 | $0.1 \pm 0.0$ | $1.3 \pm 0.4$ | – | – | $2.49 \pm 0.23$ |
| 200 | 389 | $0.2 \pm 0.0$ | $13.8 \pm 1.8$ | – | – | $2.24 \pm 0.23$ |
| 400 | 789 | $0.5 \pm 0.1$ | $124.2 \pm 8.7$ | – | – | $2.42 \pm 0.11$ |
| 800 | 1589 | $1.4 \pm 0.1$ | $763.8 \pm 20.5$ | – | – | $2.26 \pm 0.12$ |
| 1600 | 3189 | $4.7 \pm 0.1$ | $5000.4 \pm 60.8$ | – | – | $2.19 \pm 0.08$ |

## 5. Experimental Results

In this section, we present our main experimental findings, focusing on the scalability of our polynomial-time approximation algorithms, their empirical approximation factors obtained on synthetic instances and scalability in real-world data. A detailed description of the experimental setup, together with a comprehensive analysis on both synthetic as well as real-world data is available in Appendix E. All implementations are written in `python` and executed on a commodity `MacBook Air` laptop. We report the experimental results for $k$-median objective and the results for $k$-means are qualitatively similar. We use $n$ and $n''$ to indicate the number of data points in original metric and binary-tree metric, respectively.

**Scalability.** In Table 1, we report the running time and empirical approximation factor of $O(\log k)$ approximation algorithm in comparison with brute-force enumeration, as the number of data points $n$ increases while keeping $t = 3$ and $k = 5$ fixed. For each configuration of $n, t, k$, we generate five random instances $(U, d)$ in $\mathbb{R}^{10}$ (with different seeds) and report the mean and standard deviation of all reported quantities. We set $\vec{\alpha} = (1)^t$ and $\vec{\beta} = (k)^t$ to make sure that the scalability experiments are reported for the worst case scenario, where we need to consider all feasible solutions. We observed that the embedding time is negligible compared to the dynamic program; for $n = 1600$, our

---

[13]In parameterized complexity, running times of the form $((\log n)^{O(k)})$ are fixed-parameter tractable in $k$. Indeed, $(\log n)^{O(k)} = 2^{O(k \log \log n)}$. If $k < \log \log n$, this term polynomial in $n$; otherwise, $\log n \leq 2^k$, so it is bounded by $2^{O(k^2)}$. Therefore, the overall running time is $2^{O(k^2)} \cdot n^{O(1)}$, as desired.

method takes less than 1.5 hours with little variance across independent instances. In contrast, the brute-force method failed to scale beyond $n = 80$ points within reasonable time ($< 10^4$ seconds) and those experiments were terminated. Since dynamic program dominates the overall running time, we perform a more fine-grained scalability analysis in Figure 8 by varying each parameters $n'', t$, and $k$ in turn—while keeping the other two fixed—using synthetically generated binary-tree metric instances $(U'', d'')$. For $t = 4$ and $k = 5$, our implementation solves instances with $n'' = 1023$ in less than 2 hours. For $n'' = 255$ and $t = 4$, it scales to $k = 10$ within the same time budget. When $n'' = 255$ and $k = 5$, we can handle up to $t = 7$ groups in under 20 minutes. These results show that, despite the inherent computational complexity of the problem, our algorithms remains practical on modest-sized instances.

**Approximation ratio.** Across instances where brute-force enumeration is feasible, the empirical approximation factor is less than 2.7, with little variance across independent executions (see Table 1 `Approx` column). $\frac{\text{avg}(d'')}{\text{avg}(d)}$ is the ratio of average pairwise distances in binary-tree embedding and original metric. It serves as a proxy for the embedding quality and it is consistently less than factor 2.8 across all instances in our experiments. We suspect that this measure of distortion influences the resulting approximation factors. To reiterate, though the theoretical guarantees are loose, in practice the obtained approximation factors are practically useful and so far the best achievable even with `FPT` time.

**Experiments on real-world data.** We use real-world datasets from the UCI Machine Learning Repository and preprocess them by one-hot encoding all categorical at-

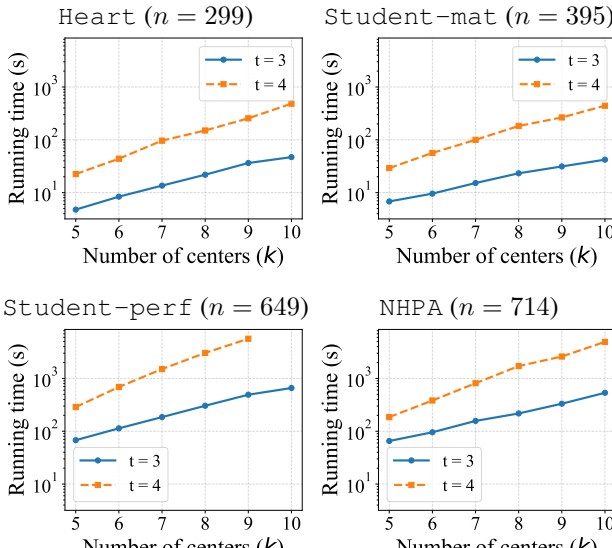

*Figure 4.* Scalability of polynomial-time approximation algorithm on real-world datasets. For each dataset (indicated in the plot title together with the number of data points), we report runtime of our pipeline (embedding + dynamic program) for $t \in \{3, 4\}$ and $k \in \{5, \dots, 10\}$. Instances with execution time that exceeded $10^4$ seconds were terminated.

tributes, followed by normalizing each column to unit norm. We then construct four intersecting demographic groups based on dataset-specific attributes (see Appendix E.1 for details). In Figure 4, we report the running time of our polynomial-time approximation algorithm as the number of centers increases, $k \in \{5, \dots, 10\}$. The reported times include both the embedding phase and the dynamic program on the tree. For all datasets, we obtain solutions in less than $10^4$ seconds ($\approx 2.8$ hours), except for `Student-perf` with $k = 10$, where we terminate the run after reaching this time limit. Since our baseline (brute-force) could not solve even the smallest real-world instance, we cannot do compare the empirical approximation factors on these datasets.

## 6. Conclusions

In this paper, we provide a comprehensive analysis of the computational complexity of (capacitated) fair-range clustering with intersecting groups, focusing on its inapproximability. Most notably, assuming Gap-ETH, no $n^{o(k)}$-time algorithm can approximate the problem to any nontrivial factor, even when feasible solutions can be found in polynomial time. On a positive note, when the number of groups $t$ is constant, we present polynomial-time $O(\log k)$ and $O(\log^2 k)$-approximation algorithms for the $k$-median and $k$-means objectives, respectively. We also give $(3 + \epsilon)$ and $(9 + \epsilon)$-approximation algorithms that run in FPT($k$)-time for these objectives. Finally, we establish that our methods are practical on modest sized problem instances in

the regime where both $k$ and $t$ are small.

**Limitations of our methodology.** In theory, coreset construction is expected to introduce only a small $\epsilon$-factor of distortion in distances. However, in practice, building smaller-sized coresets that enable us to do brute-force enumeration requires a larger $\epsilon$, resulting to higher approximation factors. As noted in prior work (Thejaswi et al., 2022), the proposed FPT($k$)-algorithms may not scale to large problem instances, given that the exponential factors are large. In contrast, our polynomial-time approximation algorithms are more practical and easier to implement, as demonstrated by our experimental results.

**Fairness in practice.** Our work focuses on fair clustering, offering theoretical insights into the challenges of designing (approximation) algorithms that support design of responsible algorithmic decision-support systems. While our algorithms provide theoretical guarantees under a formal notion of fairness, this does not automatically justify indiscriminate application. The definition of demographic groups and fairness metrics plays a crucial role in how fairness is realized in practice. Accordingly, we emphasize that our contributions are theoretical, and due caution is necessary when applying these methods in real-world settings.

## Impact Statement

Ensuring responsible design and deployment of algorithmic decision-making has profound societal, economic and ethical implications. As in the recent years, algorithmic systems are being deployed for hiring, credit rating, healthcare and law enforcement, where biased decisions can reinforce systemic inequalities and disproportionately impact marginalized communities in the society. In our view, addressing these biases through design of fairness-aware algorithms will improve social equity and builds public trust in (AI-driven) algorithmic decision-making system.

We emphasize that designing practically deployable algorithms that account for social equity is crucial for long-term societal well-being, especially when such algorithms are used in high-stakes decision making. With this motivation, it is important not only to develop theoretically sound methods, but also to ensure that they are implementable at scale. In this work, we take a significant step in this direction by presenting, to the best of our knowledge, the first algorithm for (capacitated) fair-range clustering that simultaneously offers formal approximation guarantees and can be put to work in practice.

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

## A. Further Related Work

Our work builds on prior research in clustering and algorithmic fairness. For comprehensive surveys on clustering and fair clustering, we refer the reader to these surveys (Jain et al., 1999; Chhabra et al., 2021).

Clustering is a fundamental problem in computer science, extensively studied in both theoretical and applied domains (Jain & Dubes, 1988; Vazirani, 2001). Among the most well-known clustering formulations are the $k$-median and $k$-means problems (Vazirani, 2001), along with their capacitated variants, where each facility can serve only a limited number of clients (Charikar et al., 1998). A seminal line of work by Bartal (1996) introduced approximation algorithms based on probabilistic tree embeddings, yielding an $O(\log^2 n)$-approximation for capacitated $k$-median, and later improved to $O(\log n)$ (Fakcharoenphol et al., 2004). Despite their practical relevance, the best-known polynomial-time approximations remain at $O(\log k)$ for $k$-median and $O(\log^2 k)$ for $k$-means (Adamczyk et al., 2019), with no improvements in recent years. In the FPT regime, Adamczyk et al. (2019) gave a $(7 + \epsilon)$-approximation for capacitated $k$-median in $2^{O(k \log k)} \cdot n^{O(1)}$, and it was later improved to $(3 + \epsilon)$ and $(9 + \epsilon)$ for capacitated $k$-median and $k$-means in $(O(k\epsilon^{-1} \log n))^{O(k)} \cdot n^{O(1)}$ time (Cohen-Addad & Li, 2019).

Fairness in unsupervised machine learning tasks—such as clustering, feature selection, and dimensionality reduction—has gained prominence in recent years as part of a broader focus on algorithmic fairness (Matakos et al., 2024; Gadekar et al., 2025; Kleindessner et al., 2019; Chierichetti et al., 2017; Samadi et al., 2018; Abbasi et al., 2023; Ebadian & Micha, 2025; Chen et al., 2019; Thejaswi, 2026; Kalayci & Micha, 2026; Gadekar et al., 2026). However, fair clustering was studied even before algorithmic fairness became a prominent research focus. For example, the red-blue median problem limited the maximum number of servers chosen from each type (*e.g.*, red or blue) (Hajiaghayi et al., 2012), and its generalization, the matroid median problem, captured broader fairness-like constraints (Krishnaswamy et al., 2011). Related problems also appear in robustness-based clustering, which aims to prevent disproportionately high costs for any clients (Bhattacharya et al., 2014). Our work focuses on cluster center fairness, which has seen substantial progress in recent years through formulations imposing lower bounds, upper bounds, or equality constraints on the number of centers selected from each group (Gadekar et al., 2025; Kleindessner et al., 2019; Thejaswi et al., 2021; 2022; Jones et al., 2020). We study the most general formulation—fair-range clustering—which enforces both lower and upper bounds on the number of centers selected from each group.

Hotegni et al. (2023) gave a polynomial-time approximation algorithm for the uncapacitated fair-range clustering with disjoint groups under $(\ell, p)$-norm objective. Thejaswi et al. (2024; 2022) addressed the case of intersecting groups, giving $(1 + \frac{2}{e} + \epsilon)$- and $(1 + \frac{9}{e} + \epsilon)$-approximations for $k$-median and $k$-means, respectively, in FPT($k$)-time, when the number of groups is constant. More recently, Zhang et al. (2024a) presented a $(1 + \epsilon)$-approximation for fair-range $k$-median in Euclidean metrics in FPT($k$)-time, and asked about the possibility of designing FPT-approximation algorithms when facilities have capacity constraints. Quy et al. (2021) studied fair clustering with capacity constraints, but their setting differs to us in two ways: first, fairness is imposed on clients via proportional fairness, and capacities limit the size of each cluster. In contrast, we impose fairness on center selection with lower and upper bounds on the number of centers per group. Our capacity limits are tied to facilities—each facility with its own limit—so the cluster size depends on the selected center.

# B. Hardness of Approximation

We present our reduction for uncapacitated FR$k$MED. Since the reduction is independent of the clustering objective, the inapproximability result also applies to FR$k$MEANS. As capacitated variants generalize the uncapacitated case, the hardness extends to them as well.

**Definition B.1** (3-SAT). An instance $(\phi, X)$ of the 3-SAT problem is defined on Boolean formula $\phi = C_1 \wedge C_2 \wedge \cdots \wedge C_m$ consisting of $m$ clauses, where each clause $C_i = (\ell_{i,1} \vee \ell_{i,2} \vee \ell_{i,3})$ is a disjunction of exactly three literals $\ell_{i,j} \in \{x_1, \bar{x}_1, \ldots, x_{n'}, \bar{x}_{n'}\}$ over a set of variables $X = \{x_1, \ldots, x_{n'}\}$. The goal is to decide whether there exists an assignment to the variables in $X$ that evaluates $\phi$ to true.

We use the following hardness result for 3-SAT from Håstad (2001) in our inapproximability proofs.

**Theorem B.2** (Håstad (2001), Theorem 6.1). *For every $\epsilon > 0$, it is* NP-*hard to decide if a given 3-SAT formula $\phi$ has a satisfying assignment or all assignments satisfy $< \frac{7}{8} + \epsilon$ fraction of clauses.*

To formalize our hardness results, we define a subclass of fair-range $k$-median (and $k$-means) problems—denoted FR$k$MED$^{\mathcal{O}}$ (and FR$k$MEANS$^{\mathcal{O}}$)—as instances of FR$k$MED (and FR$k$MEANS) that admit a polynomial-time algorithm for finding feasible solutions.[14] Recall that a solution is feasible if it satisfies the fair-range constraints. Our inapproximability results are established for this variant.

Next, we present polynomial-time inapproximability results for FR$k$MED$^{\mathcal{O}}$ in Section B.1, followed by parameterized inapproximability with respect to $k$ in Section B.2. The same reductions apply to FR$k$MEANS$^{\mathcal{O}}$, up to squaring of the approximation factors.

## B.1. Polynomial time inapproximability

In this subsection, we prove the following hardness result.

**Theorem 3.2.** *Unless* P = NP, *there is no polynomial-time algorithm that can approximate* FR$k$MED$^{\mathcal{O}}$ *(or* FR$k$MEANS$^{\mathcal{O}}$*) within polynomial factor, even on tree metrics.*

Note that the trivial algorithm for FR$k$MED$^{\mathcal{O}}$ (or FR$k$MEANS$^{\mathcal{O}}$) that returns any feasible solution obtained from the oracle is a factor $\Delta$ (or $\Delta^2$) approximation, where $\Delta$ is the distance aspect ratio of the input instance.[6] Towards proving Theorem 3.2, we show the following stronger statement that implies that this factor is essentially our best hope. For any function $g : \mathbb{N} \to \mathbb{R}_{>0}$, we denote by $g(n)$-FR$k$MED$^{\mathcal{O}}$ as the problem of solving FR$k$MED$^{\mathcal{O}}$ on instances of size $n$ with distance aspect ratio of the metric bounded from above by $g(n)$.

**Theorem B.3.** *For every polynomial $g : \mathbb{N} \to \mathbb{R}_{\geq 4}$ and every constant $\epsilon > 0$, it is* NP-*hard to approximate $g(n)$-FR$k$MED$^{\mathcal{O}}$ to a factor $(1 - \epsilon)\frac{g(n)-2}{16}$ on tree metrics. Furthermore, for general metrics, it is* NP-*hard to approximate $g(n)$-FR$k$MED$^{\mathcal{O}}$ to a factor $(1 - \epsilon)\frac{g(n)-2}{8}$.*

*In particular, the following holds assuming* P $\neq$ NP. *For every polynomial $g$ and for every constant $\epsilon > 0$, there is no $n^{O(1)}$ time algorithm that can decide if a given instance of $g(n)$-FR$k$MED$^{\mathcal{O}}$ has cost at most $k$ or every feasible solution has cost $> (1 - \epsilon)\frac{g(n)-2}{8} \cdot k$.*

*Finally, there is a trivial algorithm for $g(n)$-FR$k$MED$^{\mathcal{O}}$ that is a factor $g(n)$-approximation.*

We prove the above theorem by showing a reduction from 3-SAT to $g(n)$-FR$k$MED$^{\mathcal{O}}$.

### B.1.1. REDUCTION FROM 3-SAT TO $g(n)$-FR$k$MED$^{\mathcal{O}}$

Here we show the following result.

---

[14]We formalize feasibility of satisfying fair-range constraints via access to instance-specific oracles, which naturally model domain experts, who can provide feasible solutions based on implicit knowledge, heuristics, or experience, without explicitly revealing the underlying process. We do not model how such solutions are obtained, but only assume access to them. More broadly, this abstraction captures settings where feasibility arises from external or black-box mechanisms, such as legacy systems, heuristic or learning-based methods, simulation engines, or regulatory checks. While this is formally equivalent to restricting attention to instances with FR$k$MED$^{\mathcal{O}} \neq \emptyset$, meaning there exists a feasible solution for the problem instance. We believe the oracle viewpoint provides a more faithful and flexible abstraction of these practical scenarios.

**Theorem B.4.** *Given a polynomial $g : \mathbb{N} \to \mathbb{R}_{>0}$, a constant $\epsilon > 0$, and an instance $\phi$ of 3-SAT on $n'$ variable set $X = \{X_1, \ldots, X_{n'}\}$ with $m$ clauses $C = \{C_1, \ldots, C_m\}$, there is a $(m\,n')^{O(1)}$ time algorithm that computes an instance $\mathcal{I}$ of $g(n)$-FR$k$MED$^{\mathcal{O}}$ of size $n$ such that the following holds.*

1. *Parameters: $|C| = m, |F| = 8m, n = 9m + 1, k = m, t \leq 2n'm^2 + m$*

2. *(Yes case) If there is a satisfying assignment $\sigma$ to $X$ that satisfies all the clauses of $\phi$, then there is a feasible solution $S_\sigma$ to $\mathcal{I}$ that has cost at most $k$*

3. *(No case) If every assignment satisfies $< (\frac{7}{8} + \epsilon)$ fraction of clauses, then every feasible solution to $\mathcal{I}$ has cost $> (1 - 8\epsilon) \frac{g(n)-2}{16} k$.*

*Proof.* Let $(\phi, X)$ be the given instance of 3-SAT. We construct an instance $\mathcal{I}$ of FR$k$MED$^{\mathcal{O}}$ using $(\phi, X)$ as follows.

**Construction.** Let $D \geq 1$ be a fixed number. For every clause $C_i$ of $\phi$, we create a client $c_i$, and create 8 facilities $f_i^1, \ldots, f_i^8$ in $\mathcal{I}$, corresponding to the partial assignments to the variables of $C_i$. Next, we add a dummy node $s$. Now, we construct a metric over $C \cup F \cup \{s\}$ of $\mathcal{I}$ as follows. First, we create a weighted bipartite graph with left partition $C$ and right partition $F$. For each $i \in [m]$, add edges between $c_i$ and $f_i^j$, for $j \in [8]$. Furthermore, if the partial assignment corresponding to $f_i^j$ satisfies the clause $C_i$, corresponding to $c_i$, assign the weight of the edge to be 1, otherwise assign the weight to be $D$. Finally, add unit weight edges from $s$ to all clients. Next, we create the groups in $\mathbb{G}$ as follows. Specifically, we create two types of groups – *clause groups* and *assignment groups*. For every clause $C_i$ of $\phi$, create a clause group $G_i$ that contain all the facilities $f_i^1, \ldots, f_i^8$, and set $\alpha_i = \beta_i = 1$.[15] Let $G_C = \{G_1, \ldots, G_m\}$ be the set of clause groups. Next, for every variable $X_j$, for every assignment $a \in \{0, 1\}$, and for every pair of clauses $C_i$ and $C_{i'}$ containing $X_j$, we create an assignment group $G_{X_j \mapsto a}^{(C_i, C_{i'})}$ that contains all facilities in $G_i \in G_C$ that assign $a$ to $X_j$ and all facilities in $G_{i'} \in G_C$ that assign $1 - a$ to $X_j$. We set the corresponding requirements as $\alpha_{X_j \mapsto a}^{(C_i, C_{i'})} = \beta_{X_j \mapsto a}^{(C_i, C_{i'})} = 1$. Next, we set $k = m$. This completes the construction. See Section 3 for a pictorial depiction of the reduction.

We first verify the parameters of the instance $\mathcal{I}$. Since, we create client for every clause, we have $|C| = m$. For every clause, we create 8 facilities, and hence $|F| = 8m$. Additionally, we have a dummy node in the metric space, implying $n = 9m + 1$. Finally, we create $m$ clause groups, and at most $2n'm^2$ assignment groups, and hence $t \leq m + 2n'm^2$, as desired. Now, we claim that $\mathcal{I}$ is an instance of FR$k$MED$^{\mathcal{O}}$.

**Claim B.5.** *For every assignment $\sigma : X \to \{0, 1\}$, there is a feasible solution $S_\sigma$ to $\mathcal{I}$. Therefore, $\mathcal{I}$ is an instance of FR$k$MED$^{\mathcal{O}}$.*

*Proof.* Consider the solution $S_\sigma \subseteq F$ of size $k$ that contains, for every $i \in [k] = [m]$, a facility $f_i^j \in G_i, j \in [8]$ such that $f_i^j$ corresponds to the partial assignment on the variables of $C_i$ due to $\sigma$. First, we claim that $S_\sigma$ is a feasible solution to $\mathcal{I}$. To see this, note that, for every clause group $G_i \in G_C$, we have $|S_\sigma \cap G_i| = 1$, by construction. Furthermore, for a variable $X_j$, and clauses $C_i$ and $C_{i'}$ containing $X_j$, we claim that $|S_\sigma \cap G_{X_j \mapsto \sigma(X_j)}^{(C_i, C_{i'})}| = 1$. To see this, let $G_{i, X_j \mapsto \sigma(X_j)} \subseteq G_i$ be the set of facilities of $G_i$ that correspond to the partial assignments to the variables of $C_i$ that assign $\sigma(X_j)$ to $X_j$. Similarly, let $G_{i', X_j \mapsto 1 - \sigma(X_j)} \subseteq G_i$ be the set of facilities of $G_i$ that correspond to the partial assignments to the variables of $C_i$ that assign $1 - \sigma(X_j)$ to $X_j$. Then, note that $G_{X_j \mapsto \sigma(X_j)}^{(C_i, C_j)} = G_{i, X_j \mapsto \sigma(X_j)} \cup G_{i', X_j \mapsto 1 - \sigma(X_j)}$. Now, let $f_i \in G_i$ and $f_{i'} \in G_{i'}$ be the facility corresponding to the partial assignment to the variable of $C_i$ and $C_{i'}$, respectively, due to $\sigma$. First note that $f_i, f_{i'} \in S_\sigma$. Next we have, $f_i \in G_{i, X_j \mapsto \sigma(X_j)}$, hence $f_i \in G_{X_j \mapsto \sigma(X_j)}^{(C_i, C_j)}$. Now, observe that $|S_\sigma \cap G_{i, X_j \mapsto \sigma(X_j)}| = 1$, since $G_{i, X_j \mapsto \sigma(X_j)} \subseteq G_i$ and $|S_\sigma \cap G_i| = 1$ by construction. However, since $|S_\sigma \cap G_{i'}| = 1$ and since $G_{i', X_j \mapsto \sigma(X_j)} \subseteq G_{i'}$, we have that $|S_\sigma \cap G_{i', X_j \mapsto 1 - \sigma(X_j)}| = 0$, as $f_{i'} \in S_\sigma$ but $f_{i'} \notin G_{i', X_j \mapsto 1 - \sigma(X_j)}$. Therefore, $|S_\sigma \cap G_{X_j \mapsto \sigma(X_j)}^{(C_i, C_{i'})}| = 1$, and hence $S_\sigma$ is a feasible solution to $\mathcal{I}$. □

**Lemma B.6** (Yes case). *If $\phi$ has a satisfying assignment, then there exists a feasible solution to $\mathcal{I}$ with cost at most $k$.*

---

[15]In fact, we can set $\alpha_i = 1$ and $\beta_i = m$, which captures the lower bound setting of (Thejaswi et al., 2021). See Section B.3 more details.

*Proof.* Suppose there is an assignment $\sigma : X \to \{0,1\}$ to $X$ such that $\phi$ is satisfiable. Then, consider the solution $S_\sigma \subseteq F$ of size $k$ obtained from Claim B.5. As $S_\sigma$ is a feasible solution, we have $|S_\sigma \cap G_i| = 1$, for all $i \in [m]$. Therefore, let $f_i \in G_i$ be the facility that was picked from $G_i$ during the construction of $S_\sigma$. Since $\sigma$ satisfies clause $C_i$ of $\phi$, we have that the weight of the edge between $f_i$ and $c_i$ is 1. Hence, the cost of $S_\sigma$ is $m = k$. $\qquad \square$

**Lemma B.7** (No case). *If every assignment to $\phi$ satisfies at most $(7/8 + \epsilon)$ fraction of clauses, then every feasible solution to $\mathcal{I}$ has cost more than $\frac{(1-8\epsilon)D \cdot k}{8}$.*

*Proof.* We will prove the contrapositive of the statement. Suppose there is a feasible solution $S$ to $\mathcal{I}$ of size $k$ with cost at most $\frac{(1-8\epsilon)D \cdot k}{8}$. We will show an assignment $\sigma_S : X \to \{0,1\}$ to the variables of $\phi$ such that $\sigma_S$ satisfies at least $(7/8 + \epsilon)$ fraction of clauses. Since $S$ satisfies the diversity constraints on the clause groups, we have that $|S \cap G_i| = 1$, for every $i \in [m]$. Let $f_i \in G_i$ be the facility in $S$, for the clause group $G_i$. Let $\sigma_i : X|_{C_i} \to \{0,1\}$ be the partial assignment to the variables of $C_i$ corresponding to facility $f_i \in G_i \cap S$. We claim that the partial assignments $\{\sigma_i\}_{i \in [m]}$ are consistent, i.e., there is no variable $X_j \in X$ that receives different assignments from $\sigma_i, \sigma_{i'}$, for some $i, i' \in [m]$. Suppose, for the contradiction, there exist $X_j \in X$, and $i, i' \in [m]$ such that $\sigma_i(X_j) \neq \sigma_{i'}(X_j)$. Without loss of generality, assume that $\sigma_i(X_j) = 1$ and $\sigma_{i'}(X_j) = 0$, and consider the assignment group $G_{X_j \mapsto 1}^{(C_i, C_{i'})}$. Then note that both $f_i, f_{i'} \in G_{X_j \mapsto 1}^{(C_i, C_{i'})}$, since $f_i \in G_i$ corresponds to $\sigma_i$ such that $\sigma_i(X_j) = 1$, whereas $f_{i'} \in G_{i'}$ corresponds to $\sigma_{i'}$ such that $\sigma_{i'}(X_j) = 0$. Hence, $|S \cap G_{X_j \mapsto 1}^{(C_i, C_{i'})}| = 2 > \beta_{X_j \mapsto 1}^{(C_i, C_{i'})} = 1$, contradicting the fact that $S$ is a feasible solution to $\mathcal{I}$. Therefore, the partial assignments $\{\sigma_i\}_{i \in [m]}$ are consistent. Now consider the global assignment $\sigma_S : X \to \{0,1\}$ obtained from these partial assignments.[16] The following claim says that $\sigma_S$ satisfies at least $(\frac{7}{8} + \epsilon)m$ clauses, contracting Theorem B.2.

**Claim B.8.** $\sigma_S$ satisfies at least $(\frac{7}{8} + \epsilon)m$ clauses of $\phi$.

*Proof.* Let $C' \subseteq C$ be the clients that have a center in $S$ at a distance 1, and let $|C'| = \ell$. Note that the closest center for client $c_i \in C \setminus C'$ in $S$ is at a distance $D$ since $|S \cap G_i| = 1$, and the distance between $c_i$ and facilities in $G_i$ is either 1 or $D$. Therefore, the cost of $S$ is $\sum_{c \in C'} d(c, S) + \sum_{c \in C \setminus C'} d(c, S) = \ell \cdot 1 + (m - \ell)D$. Since, we assumed that the cost of $S$ is at most $\frac{(1-8\epsilon)D \cdot k}{8}$, we have that $|C'| = \ell > (\frac{7}{8} + \epsilon)m$. This means that for at least $(\frac{7}{8} + \epsilon)m$ clients in $\mathcal{I}$, there is a facility in $S$ at a distance 1. Hence, for every such client $c_i \in C'$, the corresponding clause $C_i$ is satisfied by the partial assignment $\sigma_i$, implying that the number of clauses satisfied by $\sigma_S$ is at least $(\frac{7}{8} + \epsilon)m$. $\qquad \square$

This finishes the proof of the lemma. $\qquad \square$

We finish the proof of the theorem by setting $D = g(n)/2 - 1$. $\qquad \square$

### B.1.2. PROOFS

**Proof of Theorem B.3.** Fix $g(n)$ and $\epsilon > 0$. Let $\phi$ be an instance of 3-SAT obtained from Theorem B.2. Using the construction of Theorem B.4 on $g(n)$, $\epsilon/8$, and $\phi$, we obtain an instance $\mathcal{I}$ of $g(n)$-FR$k$MED$^{\mathcal{O}}$ in $(|\phi|^{O(1)})$ such that

- If $\phi$ has a satisfying assignment, then there exists a feasible solution to $\mathcal{I}$ with cost $k$

- If every assignment to $\phi$ satisfies $< (\frac{7+\epsilon}{8})$ fraction of clauses, then every feasible solution to $\mathcal{I}$ has cost $> (1 - \epsilon)\frac{g(n)-2}{16}k$.

Therefore, it is NP-hard to decide if a given instance of $g(n)$-FR$k$MED$^{\mathcal{O}}$ has cost at most $k$ or $> (1 - \epsilon)\frac{g(n)-2}{16}k$. Finally, observe that $\mathcal{I}$ is defined on a tree metric (in fact, a depth 2 rooted tree with root $s$).

For general metrics, we obtain slightly better constants in the lower bound. The idea is that, in the construction of Theorem B.4, instead of adding the dummy vertex $S$, we add the missing edges on the graph on $C \cup F$ with weight $D$. Therefore, the distance aspect ratio $g(n) = D$, and hence the bound follows from Lemma B.14.

Finally, for the upper bound, consider an instance $\mathcal{I}$ of $g(n)$-FR$k$MED$^{\mathcal{O}}$ of size $n$, for some polynomial $g(n)$ with OPT as the optimal cost. Let $d_{\max}$ and $d_{\min}$ be the largest and the smallest distances in the metric $d$ of $\mathcal{I}$, respectively. Note that

---

[16] If a variable is not assigned by any partial assignment, we assign it an arbitrary value from $\{0,1\}$.

$g(n) = \frac{d_{\max}}{d_{\min}}$. Let $S$ be a feasible solution obtained from the oracle $\mathcal{O}$ in $(n)^{O(1)}$ time. Then, we claim that $S$ is a factor $g(n)$-approximate solution to $\mathcal{I}$. This follows since, $S$ is a feasible solution with cost

$$= \sum_{c \in C} d(c, S) \leq n \cdot d_{\max} \leq g(n) \cdot \mathsf{OPT},$$

since $\mathsf{OPT} \geq n \cdot d_{min}$.

**Proof of Theorem 3.2.** Suppose there is an algorithm $\mathcal{A}$ that, given an instance $\mathcal{I}$ of FR$k$MED$^{\mathcal{O}}$ of size $n$ outputs a feasible solution $S_{\mathcal{I}}$ in $(n)^{O(1)}$ time with cost at most $p(n) \cdot \mathsf{OPT}(\mathcal{I})$, for some polynomial function $p$, where $\mathsf{OPT}(\mathcal{I})$ is the optimal cost of $\mathcal{I}$. Let $\mathcal{I}$ be the hard instance of $g(n)$-FR$k$MED$^{\mathcal{O}}$ on tree metric obtained from Theorem 3.2 with $g(n) = 32p(n) + 2$ and $\epsilon = 1/2$. We will use $(n)^{O(1)}$-time algorithm $\mathcal{A}$ to construct a polynomial time algorithm $\mathcal{B}$ that decides if (Yes) there is a feasible solution to $\mathcal{I}$ with cost at most $k$ or (No) every feasible solution to $\mathcal{I}$ has cost $> (1-\epsilon)\frac{g(n)-2}{16} \cdot k = \frac{g(n)-2}{32} \cdot k$, contradicting the assumption $\mathsf{P} \neq \mathsf{NP}$. Our algorithm $\mathcal{B}$ first computes a feasible solution $S$ to $\mathcal{I}$ using $\mathcal{A}$. Then, $\mathcal{B}$ says Yes if the cost of $S$ is at most $p(n) \cdot k$, and No otherwise. To see that $\mathcal{B}$ correctly decides on $\mathcal{I}$, note that the cost of $S$ is at most $p(n) \cdot \mathsf{OPT}(\mathcal{I})$. If $\mathsf{OPT}(\mathcal{I}) \leq k$, then the cost of $S$ is at most $p(n) \cdot k = (\frac{g(n)}{32} - 2) \cdot k$, while if $\mathsf{OPT}(\mathcal{I}) > (1-\epsilon)\frac{g(n)-2}{16} \cdot k = \frac{g(n)-2}{32} \cdot k = p(n) \cdot k$, finishing the proof.

### B.2. Parameterized inapproximability

In this section, we strengthen Theorem 3.2 to rule out $n^{o(k)}$ time algorithms for obtaining the corresponding guarantee. Our lower bound is based on the following assumption, called Gap-ETH.

**Hypothesis B.9** ((Randomized) Gap Exponential Time Hypothesis (Gap-ETH) (Dinur, 2016; Manurangsi & Raghavendra, 2017))**.** There exists constants $\epsilon, \tau > 0$ such that no randomized algorithm when given an instance $\phi$ of 3-SAT on $n'$ variables and $O(n')$ clauses can distinguish the following cases correctly with probability $2/3$ in time $O(2^{\tau n'})$.

- there exists an assignment for $\phi$ that satisfies all the clauses

- every assignment satisfies $< (1-\epsilon)$ fraction of clauses in $\phi$.

In particular, Gap-ETH implies the following statement.

**Theorem B.10.** *Assuming **Gap-ETH**, there exist $\epsilon > 0$ such that there is no $2^{o(n')}$-time algorithm that given an instance $\phi$ of 3-SAT on $n'$ variables and $O(n')$ clauses can decide correctly with probability $2/3$ if there exists an assignment for $\phi$ that satisfies all the clauses or every assignment satisfies $< (1-\epsilon)$ fraction of clauses in $\phi$.*

We show the following hardness results.

**Theorem 3.4.** *Assuming **Gap-ETH**, there is no $f(k) \cdot n^{o(k)}$ algorithm, for any computable function $f$, that can approximate FR$k$MED$^{\mathcal{O}}$ (or FR$k$MEANS$^{\mathcal{O}}$) within any polynomial factor, even when the number of groups is $O(k^3 \log n)$, and when the metric space is a tree.*

Towards proving this, we show the following reduction which is similar to Theorem B.4.

**Theorem B.11.** *Given a polynomial $p : \mathbb{N} \rightarrow \mathbb{R}_{\geq 4}$, a constant $1 \geq \epsilon > 0$, an integer $\kappa \in \mathbb{Z}_+$, and an instance $\phi$ of 3-SAT on $n'$ variable set $X = \{X_1, \ldots, X_{n'}\}$ with $m = O(n')$ clauses $C = \{C_1, \ldots, C_m\}$, there is a $2^{O(n'/k)}(n')^{O(1)}$-time algorithm that computes an instance $\mathcal{I}$ of $p(n)$-FR$k$MED$^{\mathcal{O}}$ of size $n$, such that the following holds.*

1. *Parameters: $|C| = \kappa, |F| = \kappa \cdot 2^{O(n'/\kappa)}, n = k(2^{O(n'/\kappa)} + 1) + 1, k = \kappa, t \leq \kappa + 2n'\kappa^2$*

2. *(Yes case) If there is a satisfying assignment $\sigma$ to $X$ that satisfies all the clauses of $\phi$, then there is a feasible solution $S_\sigma$ to $\mathcal{I}$ that has cost at most $k$*

3. *(No case) If every assignment satisfies $< (1-\epsilon)$ fraction of clauses, then every feasible solution to $\mathcal{I}$ has cost $> p(n) \cdot k$.*

*Proof.* Let $(\phi, X)$ be the given instance of 3-SAT. We construct an instance $\mathcal{I}$ of FR$k$MED$^{\mathcal{O}}$ using $(\phi, X)$ as follows.

**Construction.** Without loss of generality, we assume that $k$ divides $m$, and let $m/k = \ell \in \mathbb{Z}_+$. We start by partitioning the clauses $C_1, \ldots, C_m$ of $\phi$ into $\kappa$ parts $\tilde{C}_1, \ldots, \tilde{C}_\kappa$ arbitrarily, where each part contains $\ell = m/k$ clauses of $\phi$. We call each

part $\tilde{C}_i$, $i \in [\ell]$, a *super clause*. Let $D \geq 1$ be a fixed real, which will be decided later. Let $\rho_i$ be the number of partial assignments to the variables of clauses in $\tilde{C}_i$. Then, note that $\rho_i \leq 2^{O(m/\kappa)} = 2^{O(\ell)}$, since $\tilde{C}_i$ contains $\ell$ clauses of $\phi$. For every super clause $\tilde{C}_i$ of $\phi$, we create a client $c_i$, and create $\rho_i = 2^{O(m/k)}$ facilities $f_i^1, \ldots, f_i^{\rho_i}$ in $\mathcal{I}$, corresponding to the partial assignments to the variables of clauses in $\tilde{C}_i$. Next, we add a dummy node $s$. Now, we construct a metric over $C \cup F \cup \{s\}$ of $\mathcal{I}$ as follows. First, we create a weighted bipartite graph with left partition $C$ and right partition $F$. For each $i \in [\kappa]$, add edges between $c_i$ and $f_i^j$, for $j \in [\rho_i]$. Furthermore, if the partial assignment corresponding to $f_i^j$ satisfies all the clauses in $\tilde{C}_i$, corresponding to $c_i$, assign the weight of the edge to be 1, otherwise assign the weight to be $D$. Finally, add unit weight edges from $s$ to all clients. Next, we create the groups in $\mathbb{G}$ as follows. Specifically, we create two types of groups – *super clause groups* and *assignment groups*. For every super clause $\tilde{C}_i$ of $\phi$, create a super clause group $G_i$ that contain all the facilities $f_i^1, \ldots, f_i^{\rho_i}$, and set $\alpha_i = \beta_i = 1$.[17] Let $G_{\tilde{C}} = \{G_1, \ldots, G_\kappa\}$ be the set of super clause groups. For a super clause $\tilde{C}_i$, we say that $\tilde{C}_i$ *contains* variable $X_j \in X$ if there is a clause in $\tilde{C}_i$ that contains $X_j$. Now, for every variable $X_j$, for every assignment $a \in \{0, 1\}$, and for every pair of super clauses $\tilde{C}_i$ and $\tilde{C}_{i'}$ that both contain $X_j$, we create an assignment group $G_{X_j \mapsto a}^{(\tilde{C}_i, \tilde{C}_{i'})}$ that contains all facilities in $G_i \in G_{\tilde{C}}$ that assign $a$ to $X_j$ and all facilities in $G_{i'} \in G_{\tilde{C}}$ that assign $1 - a$ to $X_j$. We set the corresponding requirements as $\alpha_{X_j \mapsto a}^{(\tilde{C}_i, \tilde{C}_{i'})} = \beta_{X_j \mapsto a}^{(\tilde{C}_i, \tilde{C}_{i'})} = 1$. Finally, we set $k = \kappa$. This completes the construction.

We first verify the parameters of the instance $\mathcal{I}$. Since, we create a client for every super clause, we have $|C| = \kappa = k$. For every super clause $\tilde{C}_i$, we create $\rho_i$ facilities, and hence $|F| = \sum_{i \in [\kappa]} \rho_i = k \cdot 2^{O(m/\kappa)}$. Additionally, we have a dummy node in the metric space, implying $n = \kappa \cdot (2^{O(m/k)} + 1) + 1$. Finally, we create $\kappa$ clause groups, and at most $2n'\kappa^2$ assignment groups, and hence $t \leq \kappa + 2n'\kappa^2$, as desired. Now, we claim that $\mathcal{I}$ is an instance of $\text{FR}k\text{MED}^{\mathcal{O}}$.

**Claim B.12.** For every assignment $\sigma : X \to \{0, 1\}$, there is a feasible solution $S_\sigma$ to $\mathcal{I}$. Therefore, $\mathcal{I}$ is an instance of $\text{FR}k\text{MED}^{\mathcal{O}}$.

*Proof.* Consider the solution $S_\sigma \subseteq F$ of size $k$ that contains, for every $i \in [k]$, a facility $f_i^j \in G_i$, $j \in [\rho_i]$ such that $f_i^j$ corresponds to the partial assignment on the variables of the clauses in $\tilde{C}_i$ due to $\sigma$. We claim that $S_\sigma$ is a feasible solution to $\mathcal{I}$. To see this, note that, for every super clause group $G_i \in G_{\tilde{C}}$, we have $|S_\sigma \cap G_i| = 1$, by construction. Furthermore, for a variable $X_j$, and super clauses $\tilde{C}_i$ and $\tilde{C}_{i'}$ containing $X_j$, we claim that $|S_\sigma \cap G_{X_j \mapsto \sigma(X_j)}^{(\tilde{C}_i, \tilde{C}_{i'})}| = 1$. To see this, let $G_{i, X_j \mapsto \sigma(X_j)} \subseteq G_i$ be the set of facilities of $G_i$ that correspond to the partial assignments to the variables of $\tilde{C}_i$ that assign $\sigma(X_j)$ to $X_j$. Similarly, let $G_{i', X_j \mapsto 1 - \sigma(X_j)} \subseteq G_i$ be the set of facilities of $G_i$ that correspond to the partial assignments to the variables of $\tilde{C}_i$ that assign $1 - \sigma(X_j)$ to $X_j$. Then, note that $G_{X_j \mapsto \sigma(X_j)}^{(\tilde{C}_i, \tilde{C}_j)} = G_{i, X_j \mapsto \sigma(X_j)} \cup G_{i', X_j \mapsto 1 - \sigma(X_j)}$. Now, let $f_i \in G_i$ and $f_{i'} \in G_{i'}$ be the facilities corresponding to the partial assignment to the variables of clauses of $\tilde{C}_i$ and $\tilde{C}_{i'}$, respectively, due to $\sigma$. First note that $f_i, f_{i'} \in S_\sigma$. Next we have, $f_i \in G_{i, X_j \mapsto \sigma(X_j)}$, hence $f_i \in G_{X_j \mapsto \sigma(X_j)}^{(\tilde{C}_i, \tilde{C}_j)}$. Now, observe that $|S_\sigma \cap G_{i, X_j \mapsto \sigma(X_j)}| = 1$, since $G_{i, X_j \mapsto \sigma(X_j)} \subseteq G_i$ and $|S_\sigma \cap G_i| = 1$ by construction. However, since $|S_\sigma \cap G_{i'}| = 1$ and since $G_{i', X_j \mapsto \sigma(X_j)} \subseteq G_{i'}$, we have that $|S_\sigma \cap G_{i', X_j \mapsto 1 - \sigma(X_j)}| = 0$, as $f_{i'} \in S_\sigma$ but $f_{i'} \notin G_{i', X_j \mapsto 1 - \sigma(X_j)}$. Therefore, $|S_\sigma \cap G_{X_j \mapsto \sigma(X_j)}^{(\tilde{C}_i, \tilde{C}_{i'})}| = 1$, and hence $S_\sigma$ is a feasible solution to $\mathcal{I}$. $\square$

**Lemma B.13** (Yes case). *If $\phi$ has a satisfying assignment, then there exists a feasible solution to $\mathcal{I}$ with cost at most $k$.*

*Proof.* Suppose there is an assignment $\sigma : X \to \{0, 1\}$ to $X$ such that $\phi$ is satisfiable. Then, consider the solution $S_\sigma \subseteq F$ of size $k$ obtained from Claim B.12.

As $S_\sigma$ is a feasible solution, we have $|S_\sigma \cap G_i| = 1$, for all $i \in [m]$. Therefore, let $f_i \in G_i$ be the facility that was picked from $G_i$ during the construction of $S_\sigma$. Since $\sigma$ satisfies all the clauses in $\tilde{C}_i$ of $\phi$, we have that the weight of the edge between $f_i$ and $c_i$ is 1. Hence, the cost of $S_\sigma$ is $k$. $\square$

**Lemma B.14** (No case). *If every assignment to $\phi$ satisfies at most $(1 - \epsilon)$ fraction of clauses, then every feasible solution to $\mathcal{I}$ has cost more than $(1 + \epsilon \cdot (D - 1))k$.*

---

[17]In fact, we can set $\alpha_i = 1$ and $\beta_i = m$, which captures the lower bound setting of (Thejaswi et al., 2021). See Section B.3 more details.

*Proof.* We will prove the contrapositive of the statement. Suppose there is a feasible solution $S$ to $\mathcal{I}$ of size $k$ with cost at most $(1 + \epsilon \cdot (D - 1))k$. We will show an assignment $\sigma_S : X \to \{0, 1\}$ to the variables of $\phi$ such that $\sigma_S$ satisfies at least $(1 - \epsilon)$ fraction of clauses. Since $S$ satisfies the diversity constraints on the super clause groups, we have that $|S \cap G_i| = 1$, for every $i \in [k]$. Let $f_i \in G_i$ be the facility in $S$, for the super clause group $G_i$. Let $\sigma_i : X|_{\tilde{C}_i} \to \{0, 1\}$ be the partial assignment to the variables of the clauses in $\tilde{C}_i$ corresponding to facility $f_i \in G_i \cap S$. We claim that the partial assignments $\{\sigma_i\}_{i \in [k]}$ are consistent, i.e., there is no variable $X_j \in X$ that receives different assignments from $\sigma_i, \sigma_{i'}$, for some $i, i' \in [k]$. Suppose, for the contradiction, there exist $X_j \in X$, and $i \neq i' \in [k]$ such that $\sigma_i(X_j) \neq \sigma_{i'}(X_j)$. Without loss of generality, assume that $\sigma_i(X_j) = 1$ and $\sigma_{i'}(X_j) = 0$ and consider the assignment group $G^{(\tilde{C}_i, \tilde{C}_{i'})}_{X_j \mapsto 1}$. Then, note that both $f_i, f_{i'} \in G^{(\tilde{C}_i, \tilde{C}_{i'})}_{X_j \mapsto 1}$, since $f_i \in G_i$ corresponds to $\sigma_i$ such that $\sigma_i(X_j) = 1$, whereas $f_{i'} \in G_{i'}$ corresponds to $\sigma_{i'}$ such that $\sigma_{i'}(X_j) = 0$. Hence, $|S \cap G^{(\tilde{C}_i, \tilde{C}_{i'})}_{X_j \mapsto 1}| \geq 2 > 1 = \beta^{(\tilde{C}_i, \tilde{C}_{i'})}_{X_j \mapsto 1}$, contradicting the fact that $S$ is a feasible solution to $\mathcal{I}$. Therefore, the partial assignments $\{\sigma_i\}_{i \in [k]}$ are consistent. Now consider the global assignment $\sigma_S : X \to \{0, 1\}$ obtained from these partial assignments.[16] The following claim says that $\sigma_S$ satisfies at least $(1 - \epsilon)m$ clauses, contracting Theorem B.2.

**Claim B.15.** $\sigma_S$ satisfies at least $(1 - \epsilon) m$ clauses of $\phi$.

*Proof.* Let $C' \subseteq C$ be the clients that have a center in $S$ at a distance 1, and let $|C'| = \mu$. Note that the closest center for client $c_i \in C \setminus C'$ in $S$ is at a distance $D$. This is due to the fact that $|S \cap G_i| = 1$, and the closest center to $c_i$ in $S_\sigma$ is in $G_i$, which is at a distance $D$ from $c_i$. Therefore, the cost of $S$ is $\sum_{c \in C'} d(c, S) + \sum_{c \in C \setminus C'} d(c, S) = \mu \cdot 1 + (k - \mu) \cdot D$. Since, we assumed that the cost of $S$ is at most $(1 + \epsilon \cdot (D - 1))k$, we have that $|C'| = \mu > (1 - \epsilon)k$. This means that for more than $(1 - \epsilon)k$ clients in $\mathcal{I}$, there is a facility in $S$ at a distance 1. We call such a client, a *good* client for $S$, the corresponding super clause $\tilde{C}_i$ a *good* super clause for $\sigma_S$. Now, note that a super clause contains $\ell = m/k$ clauses of $\phi$. Hence, all the clauses in a good super clause $\tilde{C}_i$ for $\sigma_S$, corresponding to a good client $c_i \in C'$ for $S$, are satisfied by the partial assignment $\sigma_i$, implying that the number of clauses satisfied by $\sigma_S$ is at least $(1 - \epsilon)m$. $\qquad\square$

This finishes the proof the lemma. $\qquad\square$

We finish the proof of the theorem by using $D = \frac{p(n)}{\epsilon}$. $\qquad\square$

**Proof of Theorem 3.4.** Suppose there is an algorithm $\mathcal{A}$ that, given an instance $\mathcal{I}$ FR$k$MED$^{\mathcal{O}}$ of size $n$ on tree metric, runs in time $f(k)n^{O(k/h(k))}$, for some non-decreasing and unbounded functions $f$ and $h$, and produces a feasible solution with cost at most $p(n) \cdot \mathsf{OPT}(\mathcal{I})$, for some polynomial $p$. Then, using $\mathcal{A}$, we will design an algorithm $\mathcal{B}$ that, given an instance $\phi$ of 3-SAT on $n'$ variables and $O(n')$ clauses and any $\epsilon' > 0$, correctly decides if $\phi$ has a satisfying assignment or every assignment satisfies $< (1 - \epsilon')$ fraction of clauses in $\phi$, and runs in time $2^{o(n')}$, contradicting Theorem B.10.

Given a 3-SAT formula $\phi$ on $n'$ variables and $O(n')$ clauses, and $1 \geq \epsilon > 0$, the algorithm $\mathcal{B}$ does the following, using the algorithm $\mathcal{A}$ for FR$k$MED$^{\mathcal{O}}$ that runs in time $f(k)n^{O(k/h(k))}$. Without loss of generality, we assume that $f(k) \geq 2^k$. Given $\phi$, let $\kappa$ be the largest integer such that $f(\kappa) \leq n'$. Note that, the value of $\kappa$ thus computed depends on $n'$, and hence let $\kappa = q(n')$, for some non-decreasing and unbounded function $q$. Since, $f(\kappa) \geq 2^\kappa \geq \kappa$, and $f(\kappa) \leq n'$, we have that $\kappa \leq 2 \log n'$ and $q(n') \leq n'$. Given $\phi$ and $\epsilon$, $\mathcal{B}$ first uses the reduction of Theorem B.11 on $\kappa = q(n')$ and polynomial $p$, to obtain an instance $\mathcal{I}$ of FR$k$MED$^{\mathcal{O}}$. Next, $\mathcal{B}$ runs algorithm $\mathcal{A}$ to obtain a feasible solution $S$ with cost $p(n) \cdot \mathsf{OPT}(\mathcal{I})$, returns Yes if cost of $S$ is at most $p(n) \cdot k$, and No otherwise. To see the correctness of $\mathcal{B}$ on $\phi$, note that if $\phi$ has a satisfying assignment then $\mathcal{I}$ has a feasible solution with cost $k = \kappa$. In this case, the cost of $S$ is at most $p(n) \cdot k$. On the other hand, if every assignment satisfied $< (1 - \epsilon)$ fraction of clauses of $\phi$, then every feasible solution to $\mathcal{I}$ has cost $> p(n) \cdot k$, and hence the cost of $S$ is $> p(n) \cdot k$. Therefore, $\mathcal{B}$ correctly decides if $\phi$ has a satisfying assignment or every assignment satisfies $< (1 - \epsilon')$ fraction of clauses. Finally, the running time of $\mathcal{B}$ is bounded by

$$O(f(k)n^{O(k/h(k))}) \leq O(n' \cdot (2k \cdot 2^{O(n'/k)})^{\frac{k}{h(k)}}) \leq O(n' \cdot 2^{o(n')}),$$

as desired, since $k = \kappa \leq 2 \log n'$. $\qquad\square$

### B.3. Hardness for the lower-bound only FR$k$MED$^{\mathcal{O}}$ (and FR$k$MEANS$^{\mathcal{O}}$)

In this section, we sketch the changes required in the reductions of Theorem B.4 and Theorem B.11, such that these reductions construct instances with lower-bound only requirements. Note that, in both the constructions, we create two types of groups: *clause groups* and *assignment groups*, and make the lower and upper bound requirements for every group to be 1. Therefore, without particularly fixing any construction, we focus on showing how to transform the group requirements to have lower bound only requirements such that Theorem B.4 and B.11 remain true.

**Proof sketch.** As mentioned before, every group $G_i$ constructed by Theorem B.4 and B.11 for the instance $\mathcal{I}$ of FR$k$MED$^{\mathcal{O}}$ has $\alpha_i = \beta_i = 1$. For the lower-bound only construction, we simply drop the $\beta_i$'s. Hence, $G_i$ has a lower-bound only requirement $\alpha_i = 1$. Let us denote the obtained instance as $\mathcal{I}_L$, to denote the fact that it is obtained from $\mathcal{I}$ by keeping only the lower-bounds. To argue that Theorem B.4 and Theorem B.11 remain true, we claim that a solution $S$ is feasible to $\mathcal{I}$ if and only if $S$ is feasible to $\mathcal{I}_L$. Since we only relaxed the requirements for $\mathcal{I}_L$, any feasible solution to $\mathcal{I}$ is also a feasible solution to $\mathcal{I}_L$, proving the forward direction. Now, for the reverse direction, consider a feasible solution $S$ to $\mathcal{I}_L$. We claim that, for every group $G_i$ of $\mathcal{I}$, it holds that $|S \cap G_i| = 1$, and hence $S$ satisfies the upper bound and lower bound requirements for $G_i$, since $\alpha_i = \beta_i = 1$. Therefore, $S$ is a feasible solution to $\mathcal{I}$, finishing the claim. Towards this, suppose $G_i$ is a clause group. Then, observe that there are $k$ clause groups in $\mathcal{I}_L$ which are mutually disjoint with each other. Since $|S| \leq k$, and each clause group has a lower-bound requirement of 1, it holds that $|S \cap G_i| = 1$, as required for $\mathcal{I}$. Finally, suppose $G_i$ is an assignment group. Without loss of generality, suppose $G_i$ corresponds to the assignment group $G_{X_j \mapsto a}^{(C_i, C_{i'})}$, for some pair of (super) clauses $C_i$ and $C_{i'}$, and $a \in \{0, 1\}$. Suppose for the contradiction, we have that $|S \cap G_{X_j \mapsto a}^{(C_i, C_{i'})}| = 2$. Recall that $G_{X_j \mapsto a}^{(C_i, C_{i'})}$ contains all the facilities of clause group $G_i$ that assign $a$ to $X_j$ and all facilities in clause group $G_{i'}$ that assign $1 - a$ to $X_j$. Let us denote by $G_{X_j \mapsto a}^{C_i} \subseteq G_i$, and $G_{X_j \mapsto 1-a}^{C_{i'}} \subseteq G_{i'}$, for these facilities respectively. Therefore, $G_{X_j \mapsto a}^{(C_i, C_{i'})} = G_{X_j \mapsto a}^{C_i} \cup G_{X_j \mapsto 1-a}^{C_{i'}}$.

Since, we established that $|S \cap G_i| = |S \cap G_{i'}| = 1$, it must be the case that $|S \cap G_{X_j \mapsto a}^{C_i}| = |S \cap G_{X_j \mapsto 1-a}^{C_{i'}}| = 1$, as $|S \cap G_{X_j \mapsto a}^{(C_i, C_{i'})}| = 2$. However, this means that $|S \cap G_{X_j \mapsto 1-a}^{C_i}| = |S \cap G_{X_j \mapsto a}^{C_{i'}}| = 0$, again due to $|S \cap G_i| = |S \cap G_{i'}| = 1$, since $G_{X_j \mapsto 1-a}^{C_i} \subseteq G_i$ and $G_{X_j \mapsto a}^{C_{i'}} \subseteq G_{i'}$. But since the assignment group $G_{X_j \mapsto 1-a}^{(C_i, C_{i'})} = G_{X_j \mapsto 1-a}^{C_i} \cup G_{X_j \mapsto a}^{C_{i'}}$, and therefore $|S \cap G_{X_j \mapsto 1-a}^{(C_i, C_{i'})}| = 0$, while the lower bound requirement of $G_{X_j \mapsto 1-a}^{(C_i, C_{i'})}$ is set to 1. This contradicts the fact that $S$ is a feasible solution to $\mathcal{I}_L$. Therefore, $S$ is a feasible solution to $\mathcal{I}$. $\qquad\square$

# C. A Polynomial Time Approximation Algorithm

In this section, we design a polynomial time factor $O(\log k)$-approximation for CFR$k$MED with constant number of groups (when $t$ is constant). We formally state this result in the following theorem.

**Theorem 4.1.** *There exists a $O(\log k)$ (and $O(\log^2 k)$) approximation algorithm for CFR$k$MED (CFR$k$MEANS, resp.) that runs in $(nk^t)^{O(1)}$ time.*

For the sake of clarity, we focus on presenting our results for CFR$k$MED. Although the extension to CFR$k$MEANS is straightforward, we will indicate the parts of the analysis that differ for CFR$k$MEANS.

On a high level, our algorithm works in two phases. In Phase 1, which is described in Section C.1, we embed the given instance $\mathcal{I}$ of CFR$k$MED in general metrics into another instance $\mathcal{I}'$ of CFR$k$MED on a special metric, which we call, *clique-star* metric, such that the optimal cost of $\mathcal{I}'$ is at most a constant factor of the optimal cost of $\mathcal{I}$. In Phase 2, described in Section C.2, we design a polynomial time $O(\log k)$-approximation for CFR$k$MED on clique-star metrics, thus, yielding $O(\log k)$-approximation for the CFR$k$MED in general metrics. We note that the transformations of our algorithm only modifies the underlying metric of $\mathcal{I}$, leaving everything else untouched, i.e., the instance $\mathcal{I}'$ is exactly same as $\mathcal{I}$, except that the underlying metric $d'$ in $\mathcal{I}'$ is different than the metric $d$ of $\mathcal{I}$.

**Preliminaries.** For an instance $\mathcal{I}$ of CFR$k$MED, we assume that the metric $d$ is specified by a weighted graph $H$ on a vertex set $V$ such that $V \supseteq C \cup F$, and $d$ corresponds to the shortest path metric on $H$. Furthermore, we say $d$ is $k$-*clique-star*, if $H$ consist of a $k$-clique $K_k$ and the remaining $n - k$ vertices are connected to exactly one vertex in $K_k$, i.e., they are pendant vertices to the clique nodes. We say that a given instance $\mathcal{I}$ of fair-range clustering is defined over a tree (or $k$-clique-star) metric if $d$ is a tree ($k$-clique-star, resp.).

## C.1. Embedding general metric spaces into $k$-clique-star metric

In this section, we show the following result that allows us to embed the instance $\mathcal{I}$ using an $\eta$-approximation algorithm for $k$-median into $k$-clique-star metric such that the optimal cost of the embedded instance is at most $O(\eta)$ times the optimal cost of $\mathcal{I}$.

**Lemma C.1.** *Given an instance $\mathcal{I}$ of CFR$k$MED on a general metric $d$, and a polynomial-time $\eta$-approximation algorithm $\mathcal{A}$ for $k$-median, we can construct, in $n^{O(1)}$ time, an instance $\mathcal{I}'$ of CFR$k$MED on $k$-clique-star metric $d'$ such that*

$$\text{COST}_{\mathcal{I}'}(O') \leq \text{COST}_{\mathcal{I}'}(O) \leq (4\eta + 3) \cdot \text{COST}_{\mathcal{I}}(O),$$

*where $O, O' \subseteq F$ are optimal solutions to $\mathcal{I}$ and $\mathcal{I}'$, respectively.*

*Proof.* Given an instance $\mathcal{I}$ of CFR$k$MED in metric space $d$, we first obtain $S \subseteq F$ by running algorithm $\mathcal{A}$ on $\mathcal{I}$. Note that, $S$ might be infeasible for $\mathcal{I}$, as $\mathcal{A}$ is an $\eta$-approximation algorithm for $k$-median. However, it holds that $\text{COST}_{\mathcal{I}}(S) \leq \eta \cdot \text{OPT}(\mathcal{I})$, since $\text{OPT}(\mathcal{I})$ is at most the optimal cost of $k$-median on $\mathcal{I}$. Using $S$, we construct the new metric $d'$ as follows. For every $s \in S$ and for every client $c \in C$ that is assigned to $s$, add an edge $(c, s)$ with weight $d(c, S)$. Next, for every facility $f \in F$, add an edge $(f, s_f)$, where $s_f \in S$ is the closest facility to $f$ in $S$, with weight $d(f, s_f)$. Finally, for every $s \neq s' \in S$, add edge $(s, s')$ with weight $d(s, s')$. Let $H'$ be the resultant graph. Then, $d'$ is defined as the shortest path metric on $H'$. Note that $d'$ is $k$-clique-star metric.

Consider an optimal solution $O$ to $\mathcal{I}$. For a client $c \in C$, let $o_c \in O$ be the facility serving $c$ in $O$. Similarly, for $c \in C$, let $s_c \in S$ be the facility serving $c$ in $S$, and let $s_{o_c} \in S$ be the closest facility to $o_c$ in $S$. Then, we have the following claim.

**Claim C.2.** *For every client $c \in C$ and the corresponding facilities $o_c \in O$ and $s_c \in S$, we have that*

$$d'(c, o_c) \leq 4 \cdot d(c, s_c) + 3 \cdot d(c, o_c).$$

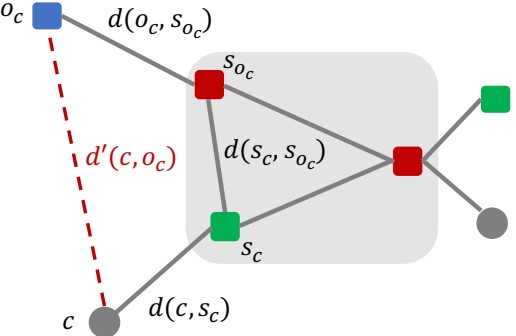

*Figure 5.* An illustration of the clique-star metric $d'$ for a client $c$. Facilities are represented as squares and clients as circles. The shaded area highlights the cluster centers $S$ selected by the $\eta$-approximation algorithm for vanilla $k$-median. In this example, client $c$ is assigned to facility $s_c \in S$, while in the optimal solution, it is served by facility $o_c$, which is connected to center $s_{o_c} \in S$ in $d'$. Our goal (Claim C.2) is to bound the rerouting cost $d'(c, o_c)$ in terms of distances in the original metric space $d$.

*Proof.* The scenario of this claim is illustrated in Figure 5. In the metric space $d'$, we have that,

$$
\begin{aligned}
d'(c, o_c) &= d(c, s_c) + d(s_c, s_{o_c}) + d(s_{o_c}, o_c) \\
&\overset{(i)}{\leq} d(c, s_c) + d(s_c, c) + d(c, o_c) + d(o_c, s_{o_c}) + d(s_{o_c}, o_c) \\
&\leq 2d(c, s_c) + d(c, o_c) + 2d(s_{o_c}, o_c) \\
&\overset{(ii)}{\leq} 2d(c, s_c) + d(c, o_c) + 2d(o_c, s_c) \\
&\overset{(iii)}{\leq} 2d(c, s_c) + d(c, o_c) + 2(d(o_c, c) + d(c, s_c)) \\
&\leq 4 \cdot d(c, s_c) + 3 \cdot d(c, o_c),
\end{aligned}
$$

where, inequality $(i)$ holds because of the triangle inequality $d(s_c, s_{o_c}) \leq d(s_c, c) + d(c, o_c) + d(o_c, s_{o_c})$, inequality $(ii)$ holds because $s_{o_c}$ was the closest facility in $S$ to $o_c$, and hence $d(o_c, s_{o_c}) \leq d(o_c, s_c)$, and finally inequality $(iii)$ holds because of the triangle inequality $d(o_c, s_c) \leq d(o_c, c) + d(c, s_c)$. Concluding this claim. □

By summing the guarantee of Claim C.2 over all $c \in C$, we obtain,

$$
\begin{aligned}
\mathrm{COST}_{\mathcal{I}'}(O) = \sum_{c \in C} d'(c, o_c) &\leq 4 \cdot \sum_{c \in C} d(c, s_c) + 3 \cdot \sum_{c \in C} d(c, o_c) \\
&\overset{(iv)}{\leq} 4\eta \cdot \mathsf{OPT}(\mathcal{I}) + 3 \cdot \mathsf{OPT}(\mathcal{I}) \\
&\leq (4\eta + 3) \cdot \mathrm{COST}_{\mathcal{I}}(O),
\end{aligned}
$$

where, inequality $(iv)$ holds because $\sum_{c \in C} d(c, s_c) = \mathrm{COST}_{\mathcal{I}}(S) \leq \eta \cdot \mathsf{OPT}(\mathcal{I})$, due to algorithm $\mathcal{A}$, and $\sum_{c \in C} d(c, o_c) = \mathsf{OPT}(\mathcal{I})$. Finally, note that $O'$ is an optimal solution to $\mathcal{I}'$, and hence, we have that $\mathrm{COST}_{\mathcal{I}'}(O') \leq \mathrm{COST}_{\mathcal{I}'}(O)$, as desired. Since $\mathcal{A}$ runs in polynomial time, and construction of $k$-clique-star $d'$ can be done in polynomial time, the overall running time is $n^{\mathsf{O}(1)}$, which concludes the proof of lemma. □

### C.2. $O(\log k)$-approximation algorithm for $k$-clique-star metrics

In this section, we show a $O(\log k)$-approximation algorithm for CFR$k$MED on $k$-clique-star metrics. Towards this, in section C.2.1, we show how to embed $k$-clique-star metric instance of CFR$k$MED into tree, at a cost of a multiplicative factor $O(\log k)$ in the cost. Finally, in section C.2.2, we show how to solve CFR$k$MED exactly on trees using a polynomial time dynamic program, resulting in a $\mathsf{O}(\log k)$-approximation for CFR$k$MED on $k$-clique-star metrics.

#### C.2.1. EMBEDDING $k$-CLIQUE-STAR METRIC INTO TREES

**Lemma C.3.** *Given an instance $\mathcal{I}'$ of CFR$k$MED of size $n$ on a $k$-clique-star metric $d'$, we can construct, in time $n^{\mathsf{O}(1)}$, an instance $\mathcal{I}''$ of CFR$k$MED on a tree metric $d''$ such that for any set $S'$ of facilities, it holds that*

$$
\mathrm{COST}_{\mathcal{I}'}(S') \leq \mathrm{COST}_{\mathcal{I}''}(S') \leq \mathsf{O}(\log k) \cdot \mathrm{COST}_{\mathcal{I}'}(S').
$$

*Proof.* Recall that the $k$-clique-star metric $d'$ has a complete graph $Q$ on $k$ nodes, while remaining $n - k$ vertices are pendant to the nodes of $Q$. According to the seminal result of Bartal (1998), any graph with $k$ nodes can be embedded into a tree metric on $\mathsf{O}(k)$ nodes with distortion at most a $\mathsf{O}(\log k)$ in the distances. [18] We use this tree embedding result to obtain $\mathcal{I}''$ on a tree metric $d''$, by simply replacing the complete graph $Q$ of $k$-clique-star metric $d'$ by the tree $T$ obtained from (Bartal, 1998). Without loss of generality, we assume that all the nodes of $Q$ of $d'$ are mapped to leaves of the $T$. By relabling the vertices of $T$, we assume that a vertex $u$ in $Q$ of $d'$ is mapped to $u$ in $T$ of $d''$. Note that, for any pair $u, v \in Q$, we have $d'(u, v) \leq d''(u, v) \leq \mathsf{O}(\log k) \cdot d'(u, v)$.

Now, conside any subset $S'$ of facilities. We claim that $\mathrm{COST}_{\mathcal{I}'}(S') \leq \mathrm{COST}_{\mathcal{I}''}(S') \leq \mathsf{O}(\log k) \cdot \mathrm{COST}_{\mathcal{I}'}(S')$. Since $d''$ dominates $d$, we have that $\mathrm{COST}_{\mathcal{I}'}(S') \leq \mathrm{COST}_{\mathcal{I}''}(S')$. Furthermore, consider a client $c \in C$ and let $f \in S'$ be the facility serving $c$. Let $q_c, q_{f_c} \in Q$ be the nodes of $Q$ that are closest to $c$ and $f_c$ in $Q$, respectively. Then, since $d''(c, f_c) = d'(c, q_c) + d''(q_c, q_{f_c}) + d'(q_{f_c}, f)$, we have,

$$d''(c, f_c) \leq d'(c, q_c) + \mathsf{O}(\log k) \cdot d'(q_c, q_{f_c}) + d'(q_f, f) \leq \mathsf{O}(\log k) \cdot d'(c, f_c),$$

since $d'(c, f_c) = d'(c, q_c) + d'(q_c, q_{f_c}) + d(q_{f_c}, f_c)$.

By summing the above equation over all $c \in C$, we obtain:

$$\mathrm{COST}_{\mathcal{I}''}(S') = \sum_{c \in C} d''(c, f_c) \leq \mathsf{O}(\log k) \sum_{c \in C} d'(c, f_c)) = \mathsf{O}(\log k) \cdot \mathrm{COST}_{\mathcal{I}'}(S').$$

The running time of the algorithm is polynomial, as the algorithm for embedding to tree metrics by Fakcharoenphol et al. (2004) runs in polynomial time and other operations also take polynomial time. This concludes the proof of lemma. $\square$

### C.2.2. AN EXACT DYNAMIC PROGRAMMING ALGORITHM FOR TREE METRICS

The final ingredient for our approach is an exact algorithm for solving CFR$k$MED on tree metrics. Our approach is inspired by the classical dynamic-programming algorithm for the vanilla $k$-median problem on tree metrics due to Tamir (1996) and its extension to the capacitated $k$-median problem by Adamczyk et al. (2019).

Let $T(e, \vec{\kappa}, b)$ denote the minimum clustering cost for the subtree rooted below edge $e$, where $\vec{\kappa} = (\kappa_1, \ldots, \kappa_t)$ specifies that exactly $\kappa_i \in [t]$ facilities from group $G_i$ are opened, and $b \in \{-n, \ldots, n\}$ represents the number of clients routed (in or out) through edge $e$. Here, $b < 0$ (or $b > 0$ resp.) indicates a capacity deficit (or surplus) of $|b|$ clients—meaning that $-b$ clients are routed upward (or $+b$ clients downward) through $e$. We now prove the following claim.

**Lemma C.4.** *There exists an exact algorithm for* CFR$k$MED *on tree metrics in* $k^{2t} \cdot n^{\mathsf{O}(1)}$ *time.*

*Proof.* Given an instance of CFR$k$MED on the tree metric $d''$, we first preprocess the tree into a rooted binary tree using standard techniques, placing all clients and facilities are placed at the leaves. The root has one child, and all internal vertices have left and right children. This can be done adding dummy nodes and zero-distance edges ensuring that the total number of nodes and edges remains in $\mathsf{O}(n)$, and the distances (for brevity, the metric $d''$) remain unchanged. For each non-root node $v$, let $e$ denote the edge connecting $v$ to its parent, and $e^\ell, e^r$ the edges connecting $v$ to its left and right children, respectively. The root node is connected by a special edge $e^\partial$. See Figure 6 for an illustration.

On this transformed tree, we proceed bottom-up, starting from edges connected to leaf nodes and moving towards the root. In the base case, if an edge $e$ connects to a client, we set $T(e, \vec{\kappa}, b) = \infty$ for all $\vec{\kappa} \in [k]^t$ and $q \in \{-n, \ldots, n\}$, as no facility is available to serve the client directly at this point, except for $T(e, \vec{0}, -1) = d''(e)$ because a client needs to be served by some facility. If an edge $e$ connects to a facility $f$ belonging to group $G_i$ with capacity $\zeta(f)$, then for $\vec{\kappa} = (0, \ldots, 1, \ldots, 0)$ with $\kappa_i = 1$ and all other other entries zero, and for $b \in \{0, \ldots, \zeta(f)\}$ we set $T(e, \vec{\kappa}, b) = d''(e) \cdot |b|$; otherwise, we set $T(e, \vec{\kappa}, b) = \infty$.

For each edge $e$, computing $T(e, \vec{\kappa}, q)$ would requires minimizing over all valid decompositions such that $\vec{\kappa}^\ell + \vec{\kappa}^r = \vec{\kappa}$ (element-wise) and $b^\ell + b^r = b$, where $\vec{\kappa}^\ell, \vec{\kappa}^r \in [k]^t$. After identifying the tuples $\vec{\kappa}^\ell, \vec{\kappa}^r, b^\ell, b^r$, we set:

$$T(e, \vec{\kappa}, b) = \min_{\substack{\vec{\kappa}^\ell + \vec{\kappa}^r = \vec{\kappa} \\ b^\ell + b^r = b}} \{T(e^\ell, \vec{\kappa}^\ell, b^\ell) + T(e^r, \vec{\kappa}^r, b^r)\} + |b| \cdot d''(e), \tag{1}$$

---

[18]The original construction of Bartal is randomized, however, as mentioned in(Adamczyk et al., 2019), this construction can be derandomized.

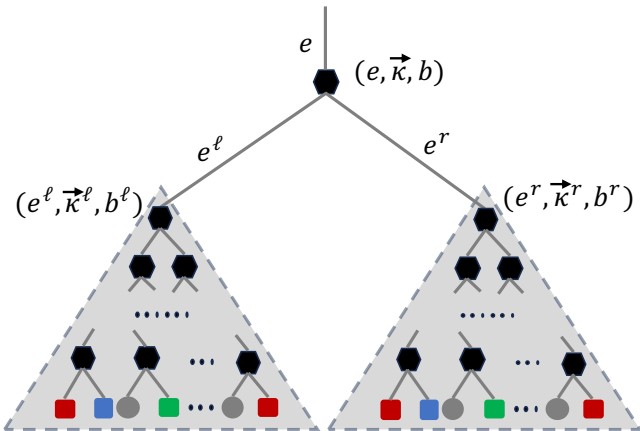

*Figure 6.* An illustration of the dynamic programming algorithm. Clients are represented as circles, facilities as squares, and internal (dummy) nodes as hexagons. The color of each facility (red, blue, green) indicates its group membership. In the dynamic programming step to compute $T(e, \vec{\kappa}, b)$, we find the minimum cost over all decompositions of the subtree solutions connected via edges $e^\ell$ and $e^r$, such that $\vec{\kappa}^\ell + \vec{\kappa}^r = \vec{\kappa}$ for all $\vec{\kappa}^\ell, \vec{\kappa}^r \in [k]^t$ and $b^\ell + b^r = b$. There is an additional cost for re-routing $|b|$ clients, which is $d''(e) \cdot |b|$, where $d''(e)$ is the distance of edge $e$ in the tree metric $d''$.

where $d''(e)$ is the length of edge $e$ in the tree. The final solution is obtained by minimizing over the pseudo-root edge $e^\partial$ over all feasible $\vec{\kappa} \in [k]^t$ satisfying the fair-range constraints, *i.e.*, $\vec{\alpha} \le \vec{\kappa} \le \beta$, as follows:

$$\min_{\substack{\vec{\kappa} \in [k]^t \\ \text{s.t. } \vec{\alpha} \le \vec{\kappa} \le \vec{\beta}}} T(e^\partial, \vec{\kappa}, 0) \tag{2}$$

The optimal subset of facilities can be recovered by storing the corresponding facility subset, rather than just the clustering cost, at each entry $T(e, \vec{\kappa}, b)$.

**Optimality of the solution.** We prove optimality by induction. In the base case, for edges connecting to leaf nodes, each entry $T(e, \vec{\kappa}, b)$ stores the optimal cost for that corresponding configuration. For the induction step, assume that $T(e^\ell, \vec{\kappa}^\ell, b^\ell)$ and $T(e^r, \vec{\kappa}^r, b^r)$ store the optimal costs for all configurations $\vec{\kappa}^\ell, \vec{\kappa}^r \in [k]^t$ and $b^\ell, b^r \in \{-n, \ldots, +n\}$. In Eq.1, we iterate over all valid decompositions satisfying $\vec{\kappa}^\ell + \vec{\kappa}^r = \vec{\kappa}$ and $b^\ell + b^r = b$, selecting the tuple $(\vec{\kappa}^\ell, \vec{\kappa}^r, b^\ell, b^r)$ that minimizes the total cost. Note that $b$ clients are routed through edge $e$ (either inwards or outwards) via edge $e$, incurring an additional cost of $|b| \cdot d''(e)$. Thus, the value stored in $T(e, \vec{\kappa}, q)$ is guaranteed to be optimal among all such combinations.

**Running time.** For each $e$, computing all configuration $(e, \vec{\kappa}, q)$ and their corresponding tuples $(\vec{\kappa}^\ell, \vec{\kappa}^r, b^\ell, b^r)$ minimizing Eq 1 requires $\mathsf{O}(k^{2t} \cdot n)$ time. Since the binary tree has $\mathsf{O}(n)$ edges, the total running time of the algorithm is $k^{2t} \cdot n^{\mathsf{O}(1)}$, concluding the proof. □

### C.3. Proof of Theorem 4.1

Our algorithm for CFR$k$MED (and CFR$k$MEANS) is as follows. Let $\mathcal{I}$ be the input instance with metric $d$. First, we use Lemma 4.2 on $\mathcal{I}$ to obtain an instance $\mathcal{I}'$ on a $k$-clique-star metric $d'$, using $\mathsf{O}(1)$-approximation algorithm for $k$-median (or $k$-Means) (Cohen-Addad et al., 2025; Arya et al., 2004; Ahmadian et al., 2019). Then, we obtain $\mathsf{O}(\log k)$-approximate ($\mathsf{O}(\log^2 k)$-approximate, resp.) solution $S$ to $\mathcal{I}'$ using the algorithm of Section C.2. Finally, we return $S$ as the solution. We claim that $S$ is an $\mathsf{O}(\log k)$-approximate ($\mathsf{O}(\log^2 k)$-approximate, resp.) solution to $\mathcal{I}$.

First, note that $S$ is a feasible solution to $\mathcal{I}$ due to the guarantee of Section C.2, and due to the fact that $\mathcal{I}$ and $\mathcal{I}'$ are same, except with different metrics. Then, note that, $\mathsf{OPT}(\mathcal{I}) \le \mathrm{COST}_{\mathcal{I}}(S) \le \mathrm{COST}_{\mathcal{I}'}(S')$, because the metric $d'$ dominates $d$, i.e., $d'(u, v) \ge d(u, v)$, for all pairs $u, v \in C \cup F$. We finish the proof by observing that, for CFR$k$MED, we have

$$\mathrm{COST}_{\mathcal{I}'}(S) \le \mathsf{O}(\log k) \cdot \mathsf{OPT}(\mathcal{I}') \le \mathsf{O}(\log k) \cdot \mathsf{OPT}(\mathcal{I}),$$

due to Lemma 4.2, since $\eta = O(1)$. Similarly, for CFR$k$MEANS, we have,

$$\mathrm{COST}_{\mathcal{I}'}(S) \le \mathsf{O}(\log^2 k) \cdot \mathsf{OPT}(\mathcal{I}') \le \mathsf{O}(\log^2 k) \cdot \mathsf{OPT}(\mathcal{I}),$$

Finally, all the parts of our algorithm, individually run in polynomial time, concluding the proof. □

---

**Algorithm 1** Polynomial-time algorithm for CFR$k$MED

---

**Input:** $I = (C, F, \mathbb{G}, \vec{\alpha}, \vec{\beta}, \zeta, k, t)$, an instance of CFR$k$MED

        $d$, the distance metric space

        $\epsilon$, a real number

**Output:** $S^*$, a subset of facilities

1   $S \leftarrow \eta\text{-ApxVanillaClustering}(C, F, k, d)$

2   $(I', d') \leftarrow k\text{-CliqueStarMetric}(I, S, d)$

3   $(I'', d'') \leftarrow \text{TreeEmbedding}(I', d')$

4   $S^* \leftarrow \text{DynamicProgram}(C, F, E, \alpha, \beta, \zeta, k, d'')$

5   **return** $S^*$

---

# D. A Fixed Parameter Tractable Time Approximation Algorithm

In this section, we present $(3 + \epsilon)$- and $(9 + \epsilon)$-approximation algorithm for CFR$k$MED and CFR$k$MEANS, respectively, running in FPT($k$)-time when the number of groups $t$ is constant. We formally state the result below.

**Theorem 4.5.** *For any $\epsilon > 0$, there exists a randomized $(3 + \epsilon)$-approximation algorithm for* CFR$k$MED *with running time* $(\mathsf{O}(2^t k \epsilon^{-1} \log n))^{\mathsf{O}(k)} \cdot n^{\mathsf{O}(1)}$. *With the same running time, a $(9 + \epsilon)$-approximation algorithm exists for* CFR$k$MEANS.

Our algorithm proceeds in three phases. Given an instance of CFR$k$MED (or CFR$k$MEANS), Phase 1 (Section D.1) reduces the number of clients to a small, weighted set called a coreset, such that the clustering cost on the coreset approximates that of the original instance. In Phase 2 (Section D.2), we transform the given instance with intersecting groups to multiple simplified instances with $k$-disjoint facility sets and replace clients with the coreset. In these simplified instances, selecting exactly one facility from each group satisfies the fair-range constraints, by construction. However, choosing arbitrary facilities may result in unbounded clustering cost relative to the optimal. In Phase 3 (described in Section D.3), we design an approximation algorithm for each simplified instance using the leader-guessing framework. Applying this framework directly would not ensure capacity and fairness constraints are met, so we modify this framework to properly account for these requirements. By combining all three phases, we obtain approximation algorithms for CFR$k$MED (or CFR$k$MEANS) with the claimed guarantees. Throughout, we describe our algorithm in the context of CFR$k$MED, noting only the necessary modifications for extending the results to CFR$k$MEANS.

## D.1. Constructing coresets for CFR$k$MED and CFR$k$MEANS

Our first step is to reduce the number of clients using coresets—a small, weighted subset of clients that approximates the clustering cost of the original client set within a bounded factor. To accommodate this, we consider each client to be associated with a weight. We now formally define the weighted variants of the capacitated fair-range $k$-median and $k$-means problems. We remark that, the unweighted variant corresponds to each client having unit weight.

**Definition D.1** (The weighted capacitated fair-range $k$-median (and $k$-means) problem). An instance $(C, F, \mathbb{G}, \vec{\alpha}, \vec{\beta}, \zeta, k, t)$ of the capacitated fair-range $k$-median (or $k$-means) problem can be extended to its weighted version, denoted as $(C, F, \mathbb{G}, \vec{\alpha}, \vec{\beta}, \zeta, \omega, k, t)$, by associating each client $c \in C$ with a weight $\omega(c) \geq 1$. The clustering cost of a solution $S \subseteq F$ is then defined as $\sum_{c \in C} \omega(c) \cdot d(c, \rho(c))$ for the $k$-median objective, and $\sum_{c \in C} \omega(c) \cdot d(c, \rho(c))^2$ for the $k$-means objective.

When clients are associated with positive real weights, it may be necessary to assign each client fractionally to multiple cluster centers, via an assignment function $\mu : C \times S \to \mathbb{R}_{\geq 0}$. This variant of the problem is particularly relevant to us, as the coresets we construct will be weighted—where the weight, loosely speaking, represents the number of original clients that a coreset client stands in for. To properly handle these weighted clients, we require the flexibility to assign their weights fractionally across multiple centers, in a way redistributing portions of the original client mass. We formally define the fractional weighted variant of the capacitated fair-range clustering problem as follows.

**Definition D.2** (The fractional weighted capacitated fair-range $k$-median (and $k$-means) problem). An instance $(C, F, \mathbb{G}, \vec{\alpha}, \vec{\beta}, \zeta, \omega, k, t)$ of the weighted capacitated fair-range clustering problem can be extended to its factional variant, where the $c \in C$ with weight $\omega(c)$ can be assigned fractionally via a function $\mu : C \times S \to \mathbb{R}_{\geq 0}$ such that $\mu$ is a proper assignment for all clients, *i.e.*, for all $c \in C$, $\sum_{f \in S} \mu(c, f) = \omega(c)$. The clustering cost of a solution $S \subseteq F$ is FRAC-COST$(C, S) = \sum_{c \in C, f \in S} \mu(c, f) \, d(c, f)$ for $k$-median and FRAC-COST$(C, S) = \sum_{c \in C, f \in S} \mu(c, f) \, d(c, f)^2$ for $k$-means.

With necessary definitions in place, let us formally define coresets for CFR$k$MED and CFR$k$MEANS.

**Definition D.3** (Coresets). For a given instance $(C, F, \mathbb{G}, \vec{\alpha}, \vec{\beta}, \zeta, k, t)$ of CFR$k$MED (or CFR$k$MEANS), a coreset is a pair $(W, \omega)$, where $W \subseteq C$ is a subset of clients and $\omega : W \to \mathbb{R}_{\geq 0}$ is a weight function assigning non-negative weights to clients in $W$. The coreset approximates the clustering cost of the full client set $C$ such that, for every subset of centers $S \subseteq F$ of size $|S| = k$, and for some small $\epsilon_1 > 0$, the following holds:

$$\text{FRAC-COST}(W, S) \in (1 \pm \epsilon_1) \, \text{COST}(C, S).$$

For any sufficiently small constant $\epsilon_1 > 0$, Cohen-Addad & Li (2019) constructed coresets of size $\mathsf{O}(k^2 \epsilon_1^{-3} \log^2 n)$ for capacitated $k$-median and $\mathsf{O}(k^5 \epsilon_1^{-3} \log^5 n)$ for capacitated $k$-means. The result is formally stated in the following theorem.

**Theorem D.4** (Cohen-Addad & Li (2019), Theorem 11, 12). *For every $\epsilon_1 > 0$, there exists a randomized algorithm that, given an instance $(k, C, F, \zeta)$ of the capacitated $k$-median or $k$-means problem, computes a coreset $(W, \mu)$, where $W \subseteq C$ and $\mu : W \to \mathbb{R}_{\geq 0}$ in $n^{O(1)}$ time. The coreset has size $|W| = O(k^2 \epsilon_1^{-3} \log^2 n)$ for the capacitated $k$-median problem and $|W| = O(k^5 \epsilon_1^{-3} \log^5 n)$ for the capacitated $k$-means problem.*

The above coreset applies to any subset of facilities $S \subseteq F$, regardless of whether it satisfies the fair-range constraints or not. Thus, a coreset constructed for the vanilla capacitated $k$-median (or $k$-means) problem can be directly used for our setting. We state this formally in the following corollary.

**Corollary D.5** (Coresets for CFR$k$MED and CFR$k$MEANS). *For every $\epsilon_1 > 0$, there exists a randomized algorithm that, given an instance $(C, F, \mathbb{G}, \vec{\alpha}, \vec{\beta}, \zeta, k, t)$ of CFR$k$MED or CFR$k$MEANS, computes a coreset $(W, \mu)$, where $W \subseteq C$ and $\mu : W \to \mathbb{R}_{\geq 0}$ in $\mathsf{poly}(n, k)$ time. The coreset has size $|W| = O(k^2 \epsilon_1^{-3} \log^2 n)$ for CFR$k$MED and $|W| = O(k^5 \epsilon_1^{-3} \log^5 n)$ for CFR$k$MEANS.*

### D.2. Enumerating feasible constraint patterns

In this subsection, we reduce CFR$k$MED (and CFR$k$MEANS) to multiple instances of a simpler problem variant, which has a restricted structure that allow us to design efficient algorithmic solutions. To do so, we adopt the framework of Thejaswi et al. (2022; 2024) to our setting. For simplicity, we describe our approach in the context of CFR$k$MED, however, since finding a feasible solution is independent of the clustering objective, the approach naturally extends to CFR$k$MEANS. We begin by introducing the notions of characteristic vector and constraint pattern.

**Definition D.6** (Characteristic vector). Given an instance $(C, F, \mathbb{G}, \vec{\alpha}, \vec{\beta}, \zeta, k, t)$ of CFR$k$MED with $\mathbb{G} = \{G_i\}_{i \in [t]}$. A characteristic vector of a facility $f \in F$ is a binary vector $\vec{\chi}_f \in \{0, 1\}^t$, where the $i$-th index $\vec{\chi}_f[i]$ is set to 1 if $f \in G_i$ and 0 otherwise. Consider the set $\{\vec{\chi}_f\}_{f \in F}$ of characteristic vectors of facilities in $F$. For each $\vec{\gamma} \in \{0, 1\}^t$, let $E(\vec{\gamma}) = \{f \in F : \vec{\chi}_f = \vec{\gamma}\}$ denote the set of all facilities with characteristic vector $\vec{\gamma}$. Finally, $\mathbb{P} = \{E(\vec{\gamma})\}_{\vec{\gamma} \in \{0,1\}^t}$ induces a partition on $F$.

**Definition D.7** (Constraint pattern). Given a $k$-multiset $\mathbb{E} = \{E(\vec{\gamma}_i)\}_{i \in [k]}$, where each $E(\vec{\gamma}_i) \in \mathbb{P}$, the constraint pattern associated with $\mathbb{E}$ is the vector obtained by the element-wise sum of the characteristic vectors $\{\vec{\gamma}_1, \ldots, \vec{\gamma}_k\}$, *i.e.*, $\sum_{i \in [k]} \vec{\gamma}_i$.

A constraint pattern is said to be feasible if $\vec{\alpha} \leq \sum_{i \in [k]} \vec{\gamma}_i \leq \vec{\beta}$, where the inequalities are taken element-wise.

In the following lemma, we establish that enumerating all feasible constraint patterns for CFR$k$MED can be done in $\mathsf{FPT}(k, t)$-time.

**Lemma D.8.** *For any instance of CFR$k$MED (or CFR$k$MEANS), there is a deterministic algorithm that enumerates all constraint patterns satisfying the fair-range constraints in time $2^{tk} \cdot n^{O(1)}$.*

*Proof.* The partition $\mathbb{P}$ can be constructed in $O(|F| t)$ time, as there are at most $|F|$ facilities and computing each characteristic vector takes $O(t)$ time. We then enumerate all possible $k$-multisets over $\mathbb{P}$ and check if they form a feasible constraint pattern. Since there are $\binom{|\mathbb{P}|+k-1}{k}$ such multisets, enumeration takes $O(|\mathbb{P}|^k t)$ time. Given $|\mathbb{P}| \leq 2^t$ and $|F| \leq n$, the running time is $O(2^{tk} tn)$. $\qquad\square$

Given $(\vec{\gamma}_i)_{i \in [k]}$ be a feasible constraint pattern. By selecting exactly one facility from each $E(\vec{\gamma}_i)$ arbitrarily for every $i \in [k]$ yields a feasible solution for CFR$k$MED. However, such an arbitrary selection may not guarantee a bounded approximation factor relative to the optimal solution. Therefore, we need to select facilities more carefully. We remark that a feasible constraint pattern $(\vec{\gamma}_i)_{i \in [k]}$ may contain two (or more) identical vectors, *i.e.*, $\vec{\gamma}_i = \vec{\gamma}_j$ for some $i \neq j$, when multiple facilities must be selected from the same group. In such cases, we duplicate facilities in $E(\vec{\gamma}_i)$, introducing infinitesimal small distortion $\epsilon_2 > 0$ to create a new group $E(\vec{\gamma}_j)$. As a result, we treat every collection $\mathbb{E}$ corresponding to a feasible constraint pattern as consisting of disjoint groups. This transformation reduces the problem to a simpler variant: the one-per-group weighted capacitated $k$-median (or $k$-means) problem with disjoint facility groups, where the goal is to choose exactly one facility from each group $E(\gamma_i)$, for every $i \in [k]$ and the clients are replaced by a coreset with associated weights. We now define this problem formally.

**Definition D.9** (The one-Per-Group Weighted Capacitated $k$-Median (or $k$-means) problem). An instance $(W, \mathbb{E}, \zeta, \omega, k)$ of the One-Per-Group Weighted Capacitated $k$-Median (or $k$-means) problem is defined by a positive integer $k$, a coreset $(W, \omega)$, where $\omega : W \to \mathbb{R}_{\geq 0}$, a collection of disjoint facility groups $\mathbb{E} = \{E_i\}_{i \in [k]}$, facility capacities $\zeta : \bigcup_{i \in [k]} E(\gamma_i) \to \mathbb{R}_{\geq 0}$.

The task is to choose exactly one center from each group $E_i$, for all $i \in [k]$, and assign clients (fractionally) to the selected centers $S$ via an assignment function $\mu : W \times S \to \mathbb{R}_{\geq 0}$, such that:

- for all clients $c \in W$, $\sum_{f \in S} \mu(c, f) = \omega(c)$,
- for all selected facilities $f \in S$, $\sum_{c \in W} \mu(c, f) \leq \zeta(f)$.

The objective is to minimize the clustering cost $\sum_{c \in W, f \in S} \mu(c, f) \cdot d(c, f)$ for $k$-median and $\sum_{c \in W, f \in S} \mu(c, f) \cdot d(c, f)^2$ for $k$-means. We succinctly denote these problems as OPG-WC$k$MED$^\emptyset$ and OPG-WC$k$MEANS$^\emptyset$, respectively.

## D.3. An **FPT** approximation algorithm for OPG-WC$k$MED$^\emptyset$ and OPG-WC$k$MEANS$^\emptyset$

In this subsection, we give a $(3 + \epsilon_3)$-approximation algorithm for OPG-WC$k$MED$^\emptyset$, for any $\epsilon_3 > 0$. The proof for OPG-WC$k$MEANS$^\emptyset$ is similar and can be obtained by replacing the distances with squared distances, resulting in the claimed $(9 + \epsilon_3)$-approximation.

**Lemma D.10.** *There exists a randomized $(3 + \epsilon_3)$-approximation algorithm, for every $\epsilon_3 > 0$, for* OPG-WC$k$MED$^\emptyset$ *in time $(\mathsf{O}(|W|\,k\,\epsilon_3^{-1} \log n))^{\mathsf{O}(k)} \cdot n^{\mathsf{O}(1)}$, where $|W|$ is the size of the coreset. With the same running time, there exists a randomized $(9 + \epsilon_3)$-approximation algorithm for* OPG-WC$k$MEANS$^\emptyset$.

*Proof.* Given an instance $(W, \mathbb{E}, \zeta, \omega, k)$ of OPG-WC$k$MED$^\emptyset$ with $\mathbb{E} = \{E_i\}_{i \in [k]}$. First we guess a set of $k$-leaders $L^* = (\ell_1^*, \ldots, \ell_k^*) \subseteq W$ that will be closest to the facilities in the optimal solution $S^* = \{f_1^*, \ldots, f_k^*\}$, with $\ell_i^*$ being closest to $f_i^*$. Additionally, we guess the radii $R^* = (r_1^*, \ldots, r_k^*)$ corresponding to each of the $k$-clusters in the optimal solution. More precisely, the aspect ratio of the metric space is bounded in interval $[1, (n)^{\mathsf{O}(1)}]$ which we discretize to $[[(n)^{\mathsf{O}(1)}]_{\epsilon_3}]$, for some $\epsilon_3 > 0$. So there are at most $[(n)^{\mathsf{O}(1)}]_{\epsilon_3} \leq \lceil \log_{1+\epsilon_3} \Delta \rceil = \mathsf{O}(\epsilon_3^{-1} \log n)$ discrete possible radii. Out of these $\mathsf{O}(\epsilon_3^{-1} \log n)$ possible radii, we guess the correct radius $r_i^*$ from the leader $\ell_i^*$ to the facility $f_i^*$ that will be serving $\ell_i^*$ in the optimal solution, *i.e.*, $d(\ell_i^*, f_i^*) = r_i^*$.

Although we can guess the set of leaders $L^*$ and their corresponding optimal radii $R^*$ from the optimal solution $S^*$, multiple facilities may lie within each radius, and it remains unclear which one to choose. The selected facilities must satisfy the following conditions: $(i)$ exactly one facility must be chosen from each group $E_i$, $(ii)$ the number of assigned clients should not exceed the capacity of facilities, $(iii)$ the clustering cost must bounded with respect to the cost of the optimal solution, $(iv)$ finding the facility set satisfying $i$, $ii$ and $iii$ must be in $\mathsf{FPT}(k, t)$ time.

**One-per-group constraint.** As mentioned earlier, there can be multiple facilities belonging to different $E(\gamma)$ may be present within the radius $r_i^*$ from a leader $\ell_i^*$, so it is unclear facility belonging to which group we need to select for $\ell_i^*$. As there are $k$ groups, we can find this in $\mathsf{O}(k^k)$ time via brute-force enumeration. For the remainder of the proof, we assume that the correct group $E(\gamma_j)$ is known for each leader $\ell_i^*$, and without loss of generality, we assume that leader $\ell_i^*$ selects a facility from group $E(\gamma_i)$ for all $i \in [k]$. Specifically, we choose the facility $f_i \in E(\gamma_i)$ with the largest capacity within radius $r_i^*$ from $\ell_i^*$. Note that $f_i$ and $f_i^*$ (the facility used in the optimal solution) may not belong to the same group. While we enforce $f_i \in E(\gamma_i)$, we make no assumptions about the group membership of $f_i^*$, which may be from any group.

**Capacity requirements.** Since $f_i$ has the highest capacity among all candidates in $E(\gamma_i)$ such that $d(\ell_i^*, f_i) \leq r_i^*$, it can serve at least as much capacity as $f_i^*$, the corresponding facility in the optimal solution, *i.e.*, $\zeta(f_i^*) \leq \zeta(f_i)$. For the purpose of bounding the approximation ratio, we assume that for each $i \in [k]$, all clients fractionally assigned to $f_i^*$ in the optimal solution are reassigned to $f_i$, *i.e.*, $\mu(c, f_i) = \mu^*(c, f_i^*)$. Such an assignment is valid because $\zeta(f_i^*) \leq \zeta(f_i)$.

**Approximation factor.** Consider the illustration in Figure 7. The distance incurred when redirecting any client $c$, originally fractionally served by $f_i^*$, to the selected facility $f_i$ can be expressed as follows:

$$
\begin{aligned}
d(c, f_i) &\overset{(i)}{\leq} d(c, f_i^*) + d(f_i^*, \ell_i^*) + d(\ell_i^*, f_i) \\
&\overset{(ii)}{\leq} d(c, f_i^*) + 2 \cdot (1 + \epsilon_3)\, r_i^* \\
&\overset{(iii)}{\leq} d(c, f_i^*) + 2 \cdot (1 + \epsilon_3)\, d(c, f_i^*) \\
&\overset{(iv)}{\leq} (3 + 2\epsilon_3)\, d(c, f_i^*),
\end{aligned}
$$

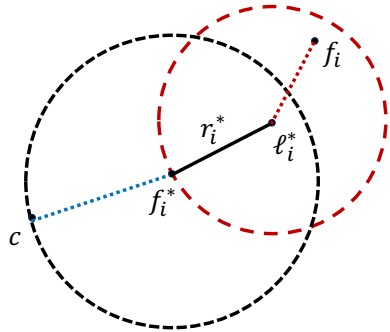

*Figure 7.* An illustration for bounding the approximation factor of the FPT algorithm for OPG-WC$k$MED$^\emptyset$. Here, $\ell_i^*$ is the leader of cluster $i \in [k]$, $f_i$ is the facility closest to $\ell_i^*$ within the guessed radius, and $f_i^*$ is the facility that (partially) serves both the client $c$ in the optimal solution $S^* = \{f_i^*\}_{i \in [k]}$. If $\mu(c, f_i^*)$ denotes the fraction of $c$ served by $f_i^*$, we aim to bound the term $\mu(c, f_i^*) \cdot d(c, f_i)$ in terms of $\mu(c, f_i^*) \cdot d(c, f_i^*)$, the former corresponds to the cost incurred in our approximate solution while the latter is corresponds to the cost incurred in the optimal solution if $c$ were to be (partially) served by $f_i^*$.

where the inequality $(i)$ follows from the triangle inequality by splitting the distance appropriately, the inequality $(ii)$ holds because we have guessed the radius $r_i^*$ such that $d(\ell_i^*, f_i) \leq r_i^*$. The additive factor of $\epsilon_3$ is coming for the distortion in the distances due to discretization of radii. The inequality $(iii)$ follows from the choice of $\ell_i^*$ as the closest client to $f_i^*$, implying $d(c, f_i^*) \geq d(\ell_i^*, f_i^*) = r_i^*$, allowing us to replace $r_i^*$ with $d(c, f_i^*)$. Since only $\mu(c, f_i^*)$ is the fraction of $c$ that is being served by $f_i^*$ in the optimal solution, which we will redirect to $f_i$ in our suboptimal solution, it holds that,

$$\mu(c, f_i^*) \cdot d(c, f_i) \leq (3 + 2\epsilon_3) \cdot \mu(c, f_i^*) \cdot d(c, f_i^*)$$

By summing over all clients $c \in W$, the total assignment cost can be bounded as follows.

$$\sum_{c \in W, f_i \in S} \mu(c, f_i^*) \cdot d(c, f_i) \leq (3 + 2\epsilon_3) \sum_{c \in C, f_i^* \in S^*} \mu(c, f_i^*) \cdot d(c, f_i^*)$$

$$\text{FRAC-COST}(C, S) \leq (3 + 2\epsilon_3) \cdot \text{OPT}$$

**Running time analysis.** Brute-force enumeration over all $k$-multisets of clients in coreset $W$ can be done in $\mathsf{O}(|W|^k)$ time. Since we assume that the aspect ratio of the metric space is bounded by a polynomial in n , we can discretize the distances into at most $\mathsf{O}(\epsilon_3^{-1} \log n)$ distinct values using standard techniques. Enumerating all possible $k$-multisets of discretized radii to guess the optimal radii can thus be done in $\mathsf{O}((\epsilon_3^{-1} \log n)^k)$ time. Guessing the correct facility group $E(\gamma_i)^*$ for each leader $\ell_i^*$ can be done in $\mathsf{O}(k^k)$ time by brute-force enumeration. Finally, for each guess, we need to compute the cost of the corresponding solution to select the one with the minimum cost. This computation can be performed in $n^{\mathsf{O}(1)}$ time. Thus, the total running time of the algorithm is:

$$\mathsf{O}\left(|W|^k \cdot (\mathsf{O}(\epsilon_3^{-1} \log n))^k \cdot \mathsf{O}(k^k) \cdot n^{\mathsf{O}(1)}\right) = \left(\mathsf{O}(|W| \, k \, \epsilon_3^{-1} \log n)\right)^{\mathsf{O}(k)} \cdot n^{\mathsf{O}(1)}.$$

The proof of $(9 + \epsilon_3)$-approximation algorithm for OPG-WC$k$MEANS$^\emptyset$ follows by replacing distances with squared distances. This concludes our proof. $\qquad\square$

### D.4. Proof of Theorem 4.5

Our algorithm for solving CFR$k$MED (or CFR$k$MEANS) proceeds as follows. For clarity, we describe the approach for CFR$k$MED; the extension to CFR$k$MEANS is similar. Given an instance $(C, F, \mathbb{G}, \vec{\alpha}, \vec{\beta}, \zeta, k, t)$ of CFR$k$MED, we first construct a coreset $(W, \omega)$ for the client set $C$ using Corollary D.5. Next, we partition $F$ into at most $\mathsf{O}(2^t)$ disjoint subsets $\mathbb{P} = \{E(\vec{\gamma})\}_{\vec{\gamma} \in \{0,1\}^t}$, where each subset $E(\vec{\gamma})$ contains facilities with a common characteristic vector $\vec{\gamma} \in \{0, 1\}^t$. Using Lemma 4.6, we enumerate all feasible $k$-multisets of $\mathbb{P}$, denoted as $\mathbb{E} = \{E(\vec{\gamma}_i)\}_{i \in [k]}$. To ensure the groups in $\mathbb{E}$ are disjoint, we duplicate facilities if needed and perturb distances slightly by $\epsilon_2 > 0$. This yields a set of at most $\mathsf{O}(2^{tk})$ instances of OPG-WC$k$MED$^\emptyset$, each defined as $J = (W, \mathbb{E}, \zeta, \omega, k)$. For each instance $J$ of OPG-WC$k$MED$^\emptyset$, for any $\epsilon_3 > 0$, we compute a $(3 + \epsilon_3)$-approximate solution using in Lemma D.10, and choose the solution with minimum cost.

Recall that we may have duplicated facility groups in $\mathbb{E}$ to ensure disjointness when constructing instances of OPG-WC$k$MED$^\emptyset$ (or OPG-WC$k$MEANS$^\emptyset$). If the same facility is selected more than once due to duplication, it would violate capacity constraints—since its capacity cannot be double-counted in the original instance. This issue is unique to the

capacitated setting and does not arise in the uncapacitated case. To handle this, we follow the technique of color coding the facilities similar to (Cohen-Addad & Li, 2019). The key idea is that a random $k$-coloring of the facilities would color each center of an optimal solution with different color with high probability ($= 1/2^{\Omega(k)}$). Hence, in our algorithm we assume that each facility in the input instance has one of the $k$ colors such that there is a feasible solution with distinct colored centers whose cost is equal to the optimal cost of the original instance. Now, for the leader $\ell_i$ and the corresponding radius $r_i^*$, we first guess the color of the optimal center that is assigned to $\ell_i$ and choose the facility of that color from $E(\gamma_i)$ that has the maximum capacity among all the centers that are within the radius $r_i^*$ of $\ell_i^*$. Finally, by choosing $\epsilon_1, \epsilon_2, \epsilon_3$ appropriately we have a $(3 + \epsilon)$-approximation algorithm for CFR$k$MED. The full procedure is summarized in Algorithm 2.

**Handling fractional assignments.** Note that OPG-WC$k$MED$^\emptyset$ is defined on weighted coreset points $(W, \omega)$, such that each data point $p \in W$ in coreset represents $\omega(p) \geq 0$ clients. Furthermore, the guarantee of such coreset is that any $\beta$-approximation on coreset $(W, \omega)$ translates into $\beta \cdot (1 + 4\epsilon)$ factor approximation for the original instance, where $\epsilon > 0$ is the coreset error parameter. Let $E$ be a $\beta$-approximate solution to $(W, \omega)$ with (possibly fractional) mapping $\mu : W \times E \to \mathbb{R}_{\geq 0}$. Now, it is sufficient to have a mapping $\mu' : C \to E$ for the original clients whose cost is at most that of $\mu$ on $(W, \omega)$, which would imply that $(E, \mu')$ is a $(1 + 4\epsilon) \cdot \beta$ solution to the original instance, due to the coreset guarantee. Now, a key observation is that given a set $E$ of facilities, whose fractional cost on $(W, \omega)$ is at most $\beta$OPT, we can always find an integral assignment $\mu_I : W \times E \to \mathbb{Z}$ with the cost at most $\beta$OPT. This is because all the weights in the coreset $(W, \omega)$ are integral, and all the capacities are integral, which means we can obtain $\mu_I$ simply by computing the min-cost flow in the corresponding network. Once we have $\mu_I$, we can easily obtain $\mu'$.

**Running time.** Our algorithm for CFR$k$MED invokes Lemma D.10 at most $\mathsf{O}(2^{tk})$ times, with each invocation running in $(\mathsf{O}(|W| \, k \, \epsilon_3^{-1} \log n))^{\mathsf{O}(k)} \cdot n^{\mathsf{O}(1)}$ time. By Corollary D.5, the coreset size is $|W| = \mathsf{O}(k^2 \epsilon_1^{-3} \log^2 n)$ for CFR$k$MED and $|W| = \mathsf{O}(k^5 \epsilon_1^{-3} \log^5 n)$ for CFR$k$MEANS. Substituting the appropriate values for $|W|$ and choosing $\epsilon_1, \epsilon_2, \epsilon_3 > 0$ suitably yields the claimed overall running time of $(\mathsf{O}(2^t k \epsilon^{-1} \log n))^{\mathsf{O}(k)} \cdot n^{\mathsf{O}(1)}$. When $t$ is constant the running time is $(\mathsf{O}(k \epsilon^{-1} \log n))^{\mathsf{O}(k)} \cdot n^{\mathsf{O}(1)}$. This concludes our proof. $\qquad\qquad\square$

---

**Algorithm 2** FPT approximation algorithm for CFR$k$MED

**Input:** $I = (C, F, \mathbb{G}, \vec{\alpha}, \vec{\beta}, \zeta, k, t)$, an instance of CFR$k$MED

$\qquad\quad$ $\epsilon_3$, a real number

**Output:** $S^*$, a subset of facilities

6 $(W, \omega) \leftarrow \text{CORESET}(I, \epsilon_1)$

7 **foreach** $\vec{\gamma} \in \{0, 1\}^t$ **do**

8 $\quad\lfloor\ E(\vec{\gamma}) \leftarrow \{f \in F : \vec{\gamma} = \vec{\chi}_f\}$

9 $\mathbb{P} \leftarrow \{E(\vec{\gamma}) : \vec{\gamma} \in \{0, 1\}^t\}$

10 $S^* \leftarrow \emptyset$

11 **foreach** *multiset* $\{E(\vec{\gamma}_1), \cdots, E(\vec{\gamma}_k)\} \subseteq \mathbb{P}$ *of size* $k$ **do**

12 $\quad$ **if** $\vec{\alpha} \leq \sum_{i \in [k]} \vec{\gamma}_i \leq \vec{\beta}$, *element-wise* **then**

13 $\qquad$ $(\mathbb{E}, M) \leftarrow$ duplicate facilities to make $\{E(\vec{\gamma}_1), \ldots, E(\vec{\gamma}_k)\}$ disjoint and get a mapping

14 $\qquad$ $J \leftarrow (W, \mathbb{E}, \zeta, \omega, k)$

15 $\qquad$ $S \leftarrow$ get approximate solution for OPG-WC$k$MED$(J, \epsilon_3)$ using Lemma D.10

16 $\qquad$ **if** $\mathsf{cost}(C, S) < \mathsf{cost}(C, S^*)$ **then**

17 $\qquad\quad\lfloor\ S^* \leftarrow S$

18 **return** $S^*$

---

## D.5. Dichotomy of **FPT**-approximability

When the number of groups is $t = k^3 \cdot o(\log n)$, Theorem 4.5 already provides a $(3 + \epsilon)$ and $(9 + \epsilon)$ approximation for $k$-median and $k$-means objectives, respectively in FPT($k$) time (using standard tools in the literature). Thus, our results present a dichotomy in the FPT-approximability landscape of the problem: it remains hard to approximate for FPT($k$) algorithms to a non-trivial factor (better than $\Delta$, the distance aspect ratio of the metric space) when $t = O(k^3 \log n)$, whereas when $t = k^3 \cdot o(\log n)$ (in fact, for $t = f(k) \cdot o(\log n)$, for any function $f$), it admits a constant-factor FPT($k$) approximation.

# E. Detailed Experimental Evaluation of Polynomial-Time Algorithms

In this section, we implement and empirically evaluate our polynomial-time algorithm for the fair-range clustering problems, which achieves $O(\log k)$ and $O(\log^2 k)$ approximation factors for the $k$-median and $k$-means objectives, respectively. Our primary goal is to assess the scalability of our approach and to demonstrate that it remains practical on problem instances of modest size, particularly when the number of groups $t$ and centers $k$ are relatively small. These experiments demonstrate that our algorithm is, to the best of our knowledge, the first method for the general variant of fair-range clustering (and its capacitated variant) with intersecting groups that can practically solve problem instances of modest size and comes with provable theoretical guarantees. As highlighted by our complexity results and prior work (Thejaswi et al., 2021; 2022; 2024; Chen et al., 2024; Hotegni et al., 2023), designing practical algorithms for fair-range clustering has remained open, with most existing approaches rely on heuristics to present practical algorithms or targeting relatively easier problem variants with disjoint groups. In contrast, with this work, we present an algorithm that is simultaneously practical and equipped with formal guarantees on the approximation ratio.

## E.1. Experimental setup

Here, we describe our experimental setup including synthetic data generators, real-world datasets and the configuration of hardware used in our experiments. We use $n$ indicate the number of data points in general metric spaces and $n''$ to indicate number of data points (vertices) in the binary-tree metric.

**Synthetic generic metric instances.** We generate synthetic data using standard numerical libraries such as `numpy` and `scipy`. The generator takes as input the number of data points $n$, the facility probability $p$, the number of groups $t$, the number of dimensions (or attributes) $z$ for each data point, and the maximum capacity of facilities $c_{\max}$. We sample an $n \times z$ matrix with entries in $[0, 1]$, treating each row as a point in $\mathbb{R}^z$, which induces a metric space $(U, d)$. Each point independently becomes a facility with probability $p$ and a client with probability $1 - p$. Clients have unit demand, no capacity, and no group membership. For each facility $u \in U$, we draw an integer capacity $c_u \in \{0, \ldots, c_{\max}\}$; unless otherwise specified, we set $c_{\max} = \lceil \frac{n}{k} \rceil$, where $k$ is the number of cluster centers. We then assign to each facility a random 0-1 group-membership vector of length $t$. Given a fixed random seed, this procedure is fully reproducible, allowing us to regenerate the same synthetic instance with a fixed seed.

**Synthetic binary-tree metric instances.** We generate tree-metric instances by first sampling a complete binary tree on $n''$ nodes and then randomly designating each leaf as either a client or a facility. This assignment depends on a (user-defined) parameter $p \in [0, 1]$, which decides the probability that a leaf becomes a facility (and $1 - p$ that it becomes a client).

- *Tree structure and distances.* Nodes are indexed $\{1, \ldots, n''\}$ and arranged as a full binary tree in array order: node $i$ has parent $\lfloor \frac{i-1}{2} \rfloor$ (for $i > 1$) and children at indices $2i$ and $2i + 1$ when these are $\leq n''$. For every existing edge, we independently draw an integer distance between 0 and maximum distance parameter $d_{\max}$; the distance between two nodes is the sum of edge distances along the unique path in the tree. Unless explicitly specified $d_{\max} = 10$.

- *Leaves, clients, and facilities.* After the tree is constructed, we mark all nodes with no children as leaves. Each leaf independently becomes a facility with probability $p$ (a parameter given to the generator as input), and a client with probability $1 - p$.

- *Capacities and group membership.* Clients have unit demand, no capacity, and no group membership. For each facility, we draw an integer capacity $u$ uniformly from $\{1, \ldots, c_{\max}\}$, where $U_{\max}$ is a parameter controlling the maximum capacity (for scalability experiments, we set $c_{\max} = \lceil \frac{n''}{k} \rceil$). We then assign the facility to a random nonempty subset of the $t$ protected groups, with an additional parameter limiting the maximum number of groups a facility can belong to. Concretely, for each facility we sample a subset $B \subseteq \{1, \ldots, t\}$, $B \neq \emptyset$, and encode it as a 0-1 group-membership vector $g \in \{0, 1\}^t$ with $g_j = 1$ if and only if $j \in B$. Internal (non-leaf) nodes are purely structural: they have neither demand nor capacity and no group membership.

This procedure generates a random binary tree metric with clients and capacitated facilities as leaves, where facilities may belong to multiple intersecting groups.

**Real world datasets.** We use real-world datasets from the UCI Machine Learning Repository (Kelly et al., 2026), specifically we use Heart Failure (`Heart`), Student Performance (`Student-mat` and `Student-perf`), and the National Poll on Healthy Aging (`NPHA`) datasets. A summary of dataset statistics is given in Table 2. We preprocessed each dataset as follows. First for categorical attributes, we apply one-hot encoding: for each categorical label of the attribute we create

*Table 2.* Statistics of real-world datasets. Here, $n$ denotes the number of data points and $z$ the number of attributes (dimension of data points). $n_c$ and $n_f$ are the numbers of clients and facilities, respectively; we set the facility probability to $p = 0.3$, so each data point is independently designated as a facility with probability $p$. The columns under no. of points report the number of data points in each group $\{G_1, \ldots, G_4\}$, respectively. Similarly, the columns under no. of facilities report the number of facilities in each group. Note that for experiments with three groups ($t = 3$) we only consider groups $G_1, G_2$ and $G_3$.

| Dataset | $n$ | $z$ | $n_c$ | $n_f$ | no. of data points | | | | no. of facilities | | | |
|---|---|---|---|---|---|---|---|---|---|---|---|---|
| | | | | | $G_1$ | $G_2$ | $G_3$ | $G_4$ | $G_1$ | $G_2$ | $G_3$ | $G_4$ |
| Heart | 299 | 13 | 205 | 94 | 194 | 105 | 74 | 151 | 64 | 30 | 23 | 39 |
| Student-mat | 395 | 33 | 274 | 121 | 187 | 208 | 366 | 199 | 59 | 62 | 114 | 56 |
| Student-perf | 649 | 59 | 458 | 191 | 266 | 383 | 468 | 325 | 85 | 106 | 134 | 83 |
| NPHA | 714 | 15 | 505 | 209 | 321 | 393 | 50 | 353 | 92 | 117 | 17 | 85 |

separate indicator columns with $\{0, 1\}$ values. We then normalize all columns to unit norm; this avoids introducing spurious distances that could arise from large numeric values in the original dataset. Given an input probability $p$ (set to $p = 0.3$ in our experiments), each data point is independently designated as a facility with probability $p$ and as a client with probability $1 - p$. Facility capacities are drawn uniformly at random from $\{1, \ldots, c_{\max}\}$, where $c_{\max}$ is an input parameter; unless otherwise specified, we set $c_{\max} = \lceil \frac{n}{k} \rceil$ in our experiments.

We construct groups $G_1$ and $G_2$ using the sex attribute in all datasets (*i.e.*, $G_1$ is Male and $G_2$ is Female). The third group $G_3$ is dataset specific: in Heart dataset we use age as threshold (age $\leq 50$); in Student-mat and Student-perf datasets, we again use age, with threshold age $\leq 18$; and in NPHA dataset we use employment attribute (employment $= 1$). The groups $G_1$ and $G_2$ are disjoint, while the third groups $G_3$ enables intersection between the groups. For experiments with four groups ($t = 4$), we introduce an additional group by randomly assigning each facility to $G_4$ with probability $0.5$, further increasing group intersections.

**Implementation.** Our implementation is written in python programming language using standard numerical libraries such as numpy and scipy for efficient matrix and linear-algebra computations. The source code will be made available as open-source in the final version of the manuscript, and the supplementary material contains of the code to support reproducibility.[19]

The implementation has two main components. In the first phase, given an instance in a general metric space, we embed it into a binary tree metric. In the second phase, we solve the fair-range $k$-median (or $k$-means) problem exactly on this binary tree using a dynamic program, which is the computational bottleneck of our approach. To improve the running time of the dynamic program, we also employ heuristics to prune the search space, retaining only the $c$ lowest-cost $(\kappa, b)$ states at each node of the binary tree and discarding the rest (see "Heuristic pruning to improve running time" for details). Since experiments for the $k$-means objective can be reproduced by simply replacing distances with squared distances, we focus our empirical evaluation on the $k$-median objective. The results for the $k$-means objective are qualitatively similar.

**Baseline.** As a baseline, we perform brute-force enumeration over all subsets of facilities of size at most $k$. For each candidate subset, we first verify that the induced group associations—called constraint pattern—of the selected facilities satisfy the fair-range constraints $\vec{\alpha}$ and $\vec{\beta}$. Given a feasible candidate subset, we then solve a minimum cost flow problem that enforces the facility capacity constraints and yields a feasible assignment of clients to facilities; we use this assignment to compute the corresponding $k$-median cost. To validate the correctness of our dynamic programming algorithm on binary trees, we compare the solution costs obtained by the dynamic program with those from the brute-force baseline on instances where enumeration is computationally feasible. However, the brute-force method fails to scale even to relatively small instances with $n = 127$. Consequently, we use the brute-force implementation mainly to verify correctness and do not report runtime comparisons for large instances: for the modest-sized real-world instances used in our main experiments, brute-force enumeration is intractable, and comparing running times on only very small instances would not be particularly informative.

**Hardware.** All experiments are executed on a MacBook Air commodity laptop with an M2 processor and 24 GB of main memory. The experiments uses a single CPU core without any explicit architecture specific optimizations.

---

[19]During implementation, we used large language models (*e.g.*, Copilot and ChatGPT) to assist in implementation and code refinement. All generated code was reviewed and verified by the authors, and we bear the responsibility for its correctness.

*Table 3.* Results of embedding general metric spaces into a binary tree metric. Here, $n = |U|$ denotes the number of data points in the original metric space $(U, d)$, and $n''$ the number of nodes in the resulting binary-tree metric $(U'', d'')$. Max diff column reports $\max_{u,v \in U} (d''(u, v) - d(u, v))$, the maximum difference in pairwise distances between the binary tree metric $d''$ and the original metric $d$. Columns Avg$(d)$ and Avg$(d'')$ reports the average pairwise distance between data points under $d$ and $d''$, respectively. In both cases, we only consider distances between the original data points $U$; distances between dummy vertices introduced in the binary tree are ignored.

| $n$ | $n''$ | time (s) | $\max(d'') - \max(d)$ | avg$(d)$ | avg$(d'')$ | avg$(d'')$ − avg$(d)$ | $\frac{\text{avg}(d'')}{\text{avg}(d)}$ |
|---|---|---|---|---|---|---|---|
| 30 | $49 \pm 1$ | $0.01 \pm 0.00$ | $4.64 \pm 0.65$ | $1.26 \pm 0.06$ | $3.03 \pm 0.34$ | $1.77 \pm 0.32$ | $2.40 \pm 0.24$ |
| 40 | $69 \pm 1$ | $0.01 \pm 0.01$ | $4.79 \pm 0.54$ | $1.28 \pm 0.02$ | $3.12 \pm 0.25$ | $1.84 \pm 0.24$ | $2.44 \pm 0.19$ |
| 60 | 109 | $0.03 \pm 0.01$ | $4.74 \pm 0.78$ | $1.26 \pm 0.02$ | $3.12 \pm 0.28$ | $1.86 \pm 0.28$ | $2.48 \pm 0.22$ |
| 80 | 149 | $0.05 \pm 0.00$ | $4.70 \pm 0.40$ | $1.27 \pm 0.01$ | $3.00 \pm 0.19$ | $1.73 \pm 0.17$ | $2.36 \pm 0.12$ |
| 100 | 189 | $0.06 \pm 0.01$ | $4.90 \pm 0.79$ | $1.25 \pm 0.02$ | $3.12 \pm 0.29$ | $1.86 \pm 0.29$ | $2.49 \pm 0.23$ |
| 200 | 389 | $0.15 \pm 0.01$ | $4.37 \pm 0.46$ | $1.26 \pm 0.01$ | $2.82 \pm 0.12$ | $1.56 \pm 0.12$ | $2.24 \pm 0.10$ |
| 400 | 789 | $0.44 \pm 0.02$ | $4.94 \pm 0.58$ | $1.27 \pm 0.01$ | $3.07 \pm 0.14$ | $1.80 \pm 0.14$ | $2.41 \pm 0.11$ |
| 800 | 1589 | $1.37 \pm 0.06$ | $4.30 \pm 0.67$ | $1.27 \pm 0.01$ | $2.86 \pm 0.14$ | $1.59 \pm 0.14$ | $2.26 \pm 0.12$ |
| 1600 | 3189 | $4.77 \pm 0.22$ | $4.45 \pm 0.41$ | $1.27 \pm 0.00$ | $2.77 \pm 0.10$ | $1.51 \pm 0.10$ | $2.19 \pm 0.08$ |
| 3200 | 6389 | $17.02 \pm 0.29$ | $4.05 \pm 0.68$ | $1.27 \pm 0.00$ | $2.67 \pm 0.17$ | $1.41 \pm 0.17$ | $2.11 \pm 0.13$ |

### E.2. Experimental results

Our first set of experiments study the effect of embedding arbitrary metrics into binary tree metric and quantify the resulting distortion in distances. The second set of experiments studies the empirical loss in approximation quality in our pipeline, which consists of embedding step followed by the dynamic program that solve the problem exactly on the binary tree. Our third set of experiments examines the scalability of the dynamic program on binary trees and then investigates the running-time improvements via pruning heuristics. Finally, we apply our methods to real-world datasets to demonstrate their practical applicability.

**Embedding general metric spaces to binary-tree metric.** Recall that the construction proceeds in two stages. First, we build a tree-star metric $d'$ by computing vanilla cluster centers, using a single-swap local-search heuristic with a maximum of $s_{\max}$ swaps. Then, we run 5 independent local-search executions with random initialization of cluster centers and select the solution with lowest-cost. Using these centers we construct a tree-star metric, which we then convert into a binary tree metric $d''$ by introducing dummy vertices and zero distance edges. For simplicity, we restrict our experiments to Euclidean metrics. Our main focus in these experiments is to compare $(i)$ the change in pairwise distances between the original (Euclidean) metric and the resulting binary-tree metric, $(ii)$ what is the change in the the size of the original instance (number of data points) and the number of vertices in the constructed binary tree? Both aspects are relevant, while the former influences the approximation quality and the later affects scalability.

In Table 3, we report the running time and difference in distances of general metric instances into a binary-tree metric as we increase the number of data points increases, with $n \in \{100, 200, \ldots, 3200\}$, and with the number of groups fixed to $t = 3$ and the number of cluster centers fixed to $k = 5$. For each instance, we generate $n$ points in $\mathbb{R}^{10}$, yielding the original metric space $(U, d)$. Group memberships are assigned at random, and each point is independently designated as a facility with (user defined) probability $p \in [0, 1]$ and as a client with probability $1 - p$, in our experiments we set $p = 0.5$. For every configuration of $n, t, k$, we generate five random instances (with different seeds) and report the mean and standard deviation of all reported quantities.

The size of the resulting binary-tree metric scales linearly with $n$: the number of tree nodes is approximately $n'' \approx 2n$, with less than $n$ vertices overhead from the added dummy vertices. The embedding step is computationally faster, for an instance with $n = 3200$, it takes less than 20 seconds, with very little variance across independent runs. In terms of distortion between the original and binary tree metric, the maximum difference in pairwise distances (*i.e.*, $\max_{u,v \in U} (d''(u, v) - d(u, v))$) stays between $4 - 6$ for all instances and does not increase significantly as $n$ increases, with little variance among independent executions. The average pairwise distance in the original metric is approximately 1.26, whereas the corresponding average in the tree metric lies between $2.7 - 3.2$, yielding an (approximate) additive distortion in distances of about $1.5 - 1.9$. The gap between the average distances in original and tree metrics decreases slightly as $n$ increases, suggesting that the embedding preserves distances reasonably well in aggregate, even though (some) individual pairwise distances may be stretched. Overall, our results indicate that embedding general metric spaces to binary-tree metrics are scalable and sufficiently accurate to serve as a practical substitute for designing scalable solutions to solve the problem on binary-trees.

*Table 4.* Comparison of running time and solution quality between our polynomial-time approximation algorithm and the brute-force baseline with number of groups $t = 3$ and number of centers $k = 5$ fixed. Here $n$ is the number of data points in the original instance and $n''$ is the number of vertices in the binary-tree embedding. Columns `Embed`, `DP`, and `BF` report the mean and standard deviation of running times (in seconds) for the embedding step, the dynamic program on the tree, and brute-force enumeration, respectively. The column (`approx factor`) reports the empirical approximation factor of our algorithm relative to the optimal solution returned by brute force. The last column $\left(\frac{\text{avg}(d'')}{\text{avg}(d)}\right)$ report the ratio of average pairwise distances in the binary-tree metric embedding $(U'', d'')$ and the original metric space $(U, d)$. We terminated experiments that took more than $10^4$ seconds ($\approx 2.8$ hours).

| | | Running time (s) | | | Embed + DP | Embed |
|---|---|---|---|---|---|---|
| $n$ | $n''$ | Embed | DP | BF | approx factor | $\frac{\text{avg}(d'')}{\text{avg}(d)}$ |
| 30 | $49 \pm 1$ | $0.01 \pm 0.00$ | $0.01 \pm 0.01$ | $3.80 \pm 2.61$ | $2.30 \pm 0.31$ | $2.40 \pm 0.24$ |
| 40 | $69 \pm 1$ | $0.02 \pm 0.01$ | $0.02 \pm 0.02$ | $17.57 \pm 16.37$ | $2.12 \pm 0.23$ | $2.44 \pm 0.19$ |
| 60 | 109 | $0.03 \pm 0.01$ | $0.25 \pm 0.06$ | $372.27 \pm 164.38$ | $1.90 \pm 0.06$ | $2.48 \pm 0.22$ |
| 80 | 149 | $0.05 \pm 0.01$ | $0.77 \pm 0.12$ | $1769.82 \pm 639.31$ | $1.83 \pm 0.09$ | $2.36 \pm 0.22$ |
| 100 | 189 | $0.06 \pm 0.01$ | $1.28 \pm 0.36$ | – | – | $2.49 \pm 0.23$ |
| 200 | 389 | $0.16 \pm 0.00$ | $13.83 \pm 1.77$ | – | – | $2.24 \pm 0.23$ |
| 400 | 789 | $0.46 \pm 0.05$ | $124.23 \pm 8.71$ | – | – | $2.42 \pm 0.11$ |
| 800 | 1589 | $1.42 \pm 0.11$ | $763.79 \pm 20.49$ | – | – | $2.26 \pm 0.12$ |
| 1600 | 3189 | $4.73 \pm 0.17$ | $5000.43 \pm 60.83$ | – | – | $2.19 \pm 0.08$ |

**Experiments on solving the capacitated fair-range clustering.** In Table 4, we compare the running time and solution quality of polynomial-time approximation algorithm (tree embedding + dynamic program) against the brute-force exact algorithm for the fair-range $k$-median problem (the results for the $k$-means objective is qualitatively similar). We report mean and standard deviation of running times on synthetic datasets with $n \in \{30, 40, 60, 80, 100, 200, 400, 800, 1600\}$, keeping $t = 3$ and $k = 5$ fixed. For each configuration of $n, t, k$, we generate data in $[0, 1]^{10} \subset \mathbb{R}^{10}$ with 10 features, inducing a metric space $(U, d)$. Group memberships are assigned uniformly at random to obtain intersecting groups, and each point independently becomes a facility with probability $p = 0.5$ and a client with probability $1 - p$. For every configuration, we generate 5 random instances to report mean and standard deviations to assess variance across independent instances. We set $\alpha = (1)^t$ and $\beta = (k)^t$ to make sure that the scalability experiments are reported for the worst case scenario, where we need to consider all feasible solutions. Where brute-force enumeration is computationally feasible ($n \in \{30, 40, 60, 80\}$), we also report the empirical approximation factor of our algorithm, defined as the ratio between the cost returned by the tree-embedding followed by dynamic programming pipeline and the optimal cost obtained via brute force. We again show the mean and standard deviation of this ratio over 5 independent instances for each such $n$.

Across all instances where brute-force enumeration is feasible, we observe that the empirical approximation factor of our $O(\log k)$-approximation algorithm is less than factor 2.7 for the smallest instances with $n = 30$. For larger instances with $n = 80$, the approximation factor is close to 2, with very little variance across independent runs. In terms of runtime, our algorithm is at least three orders of magnitude faster than brute-force enumeration: for $n = 80$, it runs in about one second, whereas the brute-force method requires at least 2000 seconds. Moreover, for $n = 1600$ our algorithm completes in less than 1.5 hours, where as brute force enumeration is difficult to scale for $n > 80$ (we terminated experiments that took more than $10^4$ seconds to execute.) These experiments indicate that our approach is not only theoretically sound but also practically implementable and efficient on modest-sized datasets , despite the underlying problem being computationally hard according to our complexity results.

**Scalability of dynamic programming algorithm on binary trees.** As established in Table 4, the running time of our polynomial-time approximation algorithm is dominated by the dynamic program that solves the problem exactly on (binary) trees, while the embedding phase hardly take a few seconds to complete for modest size datasets in the order of thousands. So scalability of our method to large datasets is very much dependent on the scalability of our dynamic programming algorithm. In the next set of experiments, we evaluate the scalability of the exact dynamic programming algorithm on binary trees in a regime that has relatively small number of centers and groups. We fix $\vec{\alpha} = (1)^t$ and $\vec{\beta} = (k)^t$, *i.e.*, a vector of $t$ ones and a vector of $t$ copies of $k$, so that all $\vec{\gamma}$ are feasible under the fair-range constraints. This choice avoids scalability artifacts that could arise if certain random instances trivially eliminated many states (feasibility patterns) and thereby made the problem artificially easier. In Figure 8, we report the scalability of our dynamic programming algorithm for solving the fair-range $k$-median problem on binary trees to optimality (the results are quantitatively similar for $k$-means objective). For each combination of $n$, $t$, and $k$, we generate five synthetic instances at random and run the algorithm with the same fixed $\vec{\alpha}$

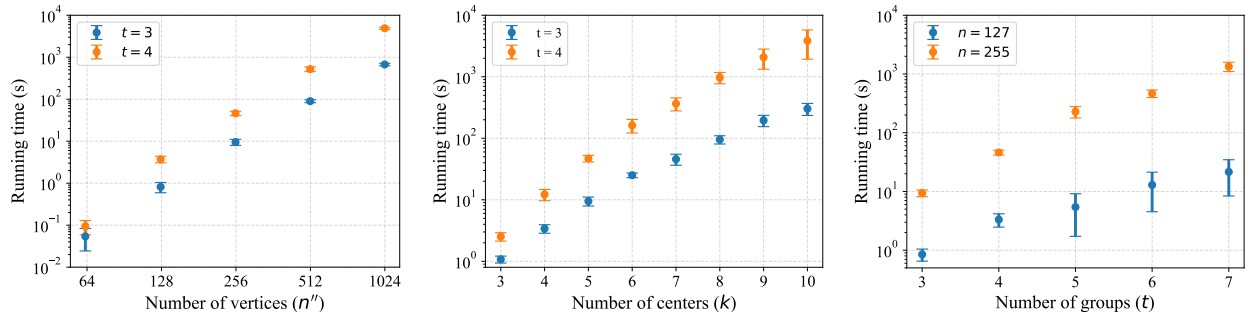

*Figure 8.* Scalability of our dynamic programming algorithm for computing exact solutions on binary tree metric. The left panel shows runtime as a function of the number of vertices $n'' \in \{63, 127, \ldots, 1023\}$, with the number of centers fixed at $k = 5$ and the number of groups $t \in \{3, 4\}$. The middle panel shows scaling with respect to the number of centers $k \in \{3, 4, \ldots, 10\}$ for a fixed tree size $n'' = 255$ and $t \in \{3, 4\}$. The right panel shows runtime as a function of the number of groups $t \in \{3, 4, \ldots, 7\}$, with the number of centers fixed at $k = 5$ and tree sizes $n'' \in \{127, 255\}$.

and $\vec{\beta}$ values and report the mean running times with the standard deviation.

Figure 8 left panel illustrates that the algorithm scales to trees with more than 1000 vertices when both $k$ and $t$ are relatively small ($k = 5, t \in \{3, 4\}$). For each configuration of $n, t$, and $k$, we run 5 independent trials and plot the mean running time together with the standard deviation. We observed little variance across independent executions. For the largest instances with over one thousand vertices and $t = 3$, the algorithm requires less than 11 minutes to find the optimal solution in all independent trials. When increasing to $t = 4$, the runtime remains less than 2 hours. These results demonstrate that, despite the problem's inherent computational hardness, our algorithm is practical on modest-sized datasets when numbers of centers and groups is small.

Figure 8, middle panel, shows that the algorithm scales up to $k = 10$ for instances with $n = 255$ vertices and $t \in \{3, 4\}$ groups. For $t = 3$, we observe little variance in running times across independent executions. In contrast, for $t = 4$, the running times exhibit higher variability. This is because, as the number of groups increases, the number of feasible states satisfying the fair-range constraints at each vertex can vary substantially when populating the dynamic programming table. This variability translates into differences in running time, which become particularly pronounced as $t$ increases. For a binary-tree instance with $n = 255, t = 4$, and $k = 10$, the algorithm successfully computes an optimal solution in under 2 hours on a commodity laptop.

Figure 8, right panel, illustrates the scalability of our implementation as we increase the number of groups $t \in \{3, 4, \ldots, 7\}$ with fixed $k = 5$ and $n \in \{127, 255\}$. For instances with $n = 255, t = 7$, and $k = 5$, our implementation finds an optimal solution in under 20 minutes across all independent trials. We observe little variance in running time for the larger tree size $n = 255$. For the smaller trees ($n = 127$), the variance appears more pronounced in the plots, but this is largely an artifact of the log-scale visualization. In absolute terms, the differences in runtime are on the order of only a few seconds, since all executions are very fast in this regime.

**Heuristic pruning to improve running time.** To further improve scalability, we apply a beam-search–style truncation of the DP table, retaining only the $c$ lowest-cost states per edge. Each state is a pair $(\vec{\kappa}, b)$ with $\vec{\kappa} \in \{0, \ldots, k\}^t$ and $b \in [-n, n]$. This reduces the per-edge complexity from being combinatorial in the total number of states to $\mathsf{O}(c^2)$, since at most $c^2$ state pairs are considered when computing each entry $T[e, \vec{\kappa}, b]$. Consequently, smaller values of $c$ can substantially improve tractability, even though the full state space still grows rapidly with $t$ and $k$. While this heuristic does not always guarantee a feasible solution, our experiments show that with $c = n \cdot t \cdot k^2$, the pruned DP recovers the optimal solution in roughly half the running time of the exact dynamic program. In Figure 9, we compare the running times of the exact and pruned DPs for $n \in \{63, 127, \ldots, 1023\}$ with $t = 3, k = 4$, and 5 random trials per configuration. We report the mean and standard deviation and observe little variance across runs. The pruning heuristic, which keeps at most $n \cdot t \cdot k^2$ states per edge, finds the optimal solution in all instances. For $n = 1023$, the exact DP takes more than twice as long as the pruned version to obtain the same optimal solution, indicating that this pruning can be practically effective in regimes with many feasible solutions.

Although this pruning strategy is practical in settings where the constraints are not too tight, it can perform poorly when the number of feasible solutions is small, *i.e.*, when the fair-range constraints are tight and only few $\vec{\kappa}$ satisfy the fair-range

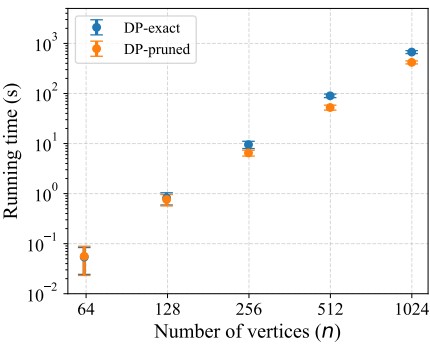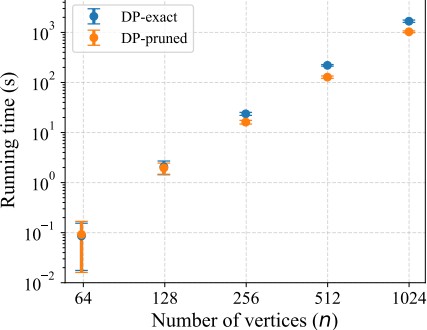

*Figure 9.* Running-time comparison of the pruning technique applied to the dynamic programming algorithm. The plots report the running times for $k = 5$ (left panel), and for $k = 6$ (right panel), with $n'' \in \{63, 127, \ldots, 1023\}$ and $t = 3$ fixed in both cases.

constraints. In such cases, feasible combinations of $(\vec{\kappa}^\ell, b^\ell)$ and $(\vec{\kappa}^r, b^r)$ for the left and right child edges may be discarded if they do not fall within the top $c$ states at those edges. The algorithm may then fail to find any feasible solution and thus cannot recover an optimal one. Note that here there is an additional possibility of loosing the exact combination of $(\vec{\kappa}^\ell, b^\ell)$ and $(\vec{\kappa}^r, b^r)$ that gives the optimal solution, here for a state $\kappa, b$ to find an optimal solution both $\kappa^\ell$ and $\kappa^r$ should sum to $\kappa$ as well as the capacity in/out flow $b^\ell$ and $b^r$ should also sum to $b$, this is a difficult constraint to achieve both while only storing top $c$ states. An interesting future work would be to refine the pruning rule so that it preferentially retain states that are more likely to participate in feasible patterns, for example by exploiting how restrictive the lower and upper bounds are, though this still cannot guarantee a feasible solution in all cases. However, when the fair-range constraints are relatively relaxed, empirically we observed that near-optimal solution can be obtained while storing relatively smaller number of $(\vec{\kappa}, b)$-states per edge (bounded by $c$).

**Experiments with real-world data.** In this section, we present our experimental results on real-world datasets. Each dataset is first embedded into a binary-tree metric using the local-search–based vanilla clustering with $s_{\max} = 10000$ swaps (see Section 4.1 and the experiments in "Embedding general metric spaces into binary-tree metrics" for details of the embedding procedure). Table 5 summarizes the quality of these embeddings. In particular, we report the maximum difference between pairwise distances in the original metric $d$ and the tree metric $d''$, and observe that these discrepancies are smaller than for the synthetic instances used in our earlier scalability experiments. Across all datasets, the average pairwise distances in the embedded tree metrics change by at most a factor of at most 3 compared to their original metric spaces, indicating that the embeddings preserve distances reasonably well in practice.

After constructing the binary-tree metric, we run our dynamic program to compute an exact solution, using $\vec{\alpha} = (1)^t$ and $\vec{\beta} = (k)^t$. Note that these settings are likely to yield the worst-case running times: every group-association vector is feasible, so there are many candidate solutions and the DP has limited opportunity to prune the state space. Any stricter fair-range constraints $\vec{\alpha}, \vec{\beta}$ would only reduce the number of feasible patterns and thus improve the runtime. In Figure 4, we report the running time of our $O(\log k)$-approximation algorithm for fair-range clustering under the $k$-median objective. For each dataset, we vary the number of cluster centers $k \in \{5, \ldots, 10\}$ and plot the total running time, including both the embedding step and the subsequent dynamic program on the binary tree. Even on commodity hardware (a MacBook Air with an M2 processor, using a single core), our implementation solves the largest instance (`NPHA`) with $n = 714$, $t = 3$ and $k = 10$ in less than 10 minutes. These results further demonstrate the practical viability of our approach and, to the best of our knowledge, provide the first implementation that scales to fair-range clustering with provable guarantees.

*Table 5.* Results of embedding real-world datasets in Euclidean metric into a binary tree metric for $k \in \{5, 10\}$ with $t = 3$ groups; and $k = \{5, 9\}$ with $t = 4$ groups. Here, $n = |U|$ denotes the number of data points in the original metric space $(U, d)$, and $n''$ the number of nodes in the resulting binary-tree metric. Column time reports the running time in seconds to embed the dataset. The column $\max(d'') - \max(d)$ reports the maximum difference (*i.e.* $\max_{u,v \in U} (d''(u, v) - d(u, v))$) in pairwise distances between the binary tree metric $d''$ and the original metric $d$. Columns Avg($d$) and Avg($d''$) reports the average pairwise distance between data points under $d$ and $d''$, respectively. In both cases, we only consider distances between the original data points $U$; distances between dummy vertices introduced in the binary tree are ignored.

| Dataset | $n$ | $n''$ | time (s) | $\max(d'')-\max(d)$ | avg($d$) | avg($d''$) | avg($d'')-$avg($d$) | $\frac{\text{avg}(d'')}{\text{avg}(d)}$ |
|---|---|---|---|---|---|---|---|---|
| $t = 3, k = 5$ | | | | | | | | |
| Heart | 299 | 587 | 0.24 | 0.987 | 0.177 | 0.440 | 0.263 | 2.486 |
| Student-mat | 395 | 779 | 0.35 | 0.911 | 0.224 | 0.546 | 0.322 | 2.438 |
| Student-perf | 649 | 1287 | 0.83 | 0.847 | 0.296 | 0.678 | 0.382 | 2.291 |
| NPHA | 714 | 1417 | 0.90 | 0.420 | 0.107 | 0.224 | 0.117 | 2.093 |
| $t = 3, k = 10$ | | | | | | | | |
| Heart | 299 | 577 | 0.46 | 1.071 | 0.177 | 0.528 | 0.351 | 2.983 |
| student-mat | 395 | 769 | 0.68 | 0.981 | 0.224 | 0.614 | 0.390 | 2.742 |
| student-perf | 649 | 1277 | 1.65 | 1.236 | 0.296 | 0.781 | 0.485 | 2.639 |
| NPHA | 714 | 1407 | 1.70 | 0.695 | 0.107 | 0.277 | 0.170 | 2.589 |
| $t = 4, k = 5$ | | | | | | | | |
| Heart | 299 | 587 | 0.19 | 0.902 | 0.177 | 0.425 | 0.249 | 2.411 |
| Student-mat | 395 | 779 | 0.31 | 0.806 | 0.224 | 0.530 | 0.306 | 2.366 |
| Student-perf | 649 | 1287 | 0.79 | 1.221 | 0.296 | 0.719 | 0.422 | 2.429 |
| NPHA | 714 | 1417 | 0.87 | 0.441 | 0.107 | 0.220 | 0.113 | 2.056 |
| $t = 4, k = 9$ | | | | | | | | |
| Heart | 299 | 579 | 0.34 | 0.968 | 0.177 | 0.501 | 0.324 | 2.831 |
| Student-mat | 395 | 771 | 0.56 | 1.223 | 0.224 | 0.620 | 0.396 | 2.768 |
| Student-perf | 649 | 1279 | 1.24 | 1.329 | 0.296 | 0.795 | 0.499 | 2.686 |
| NPHA | 714 | 1409 | 1.40 | 0.487 | 0.107 | 0.235 | 0.128 | 2.196 |

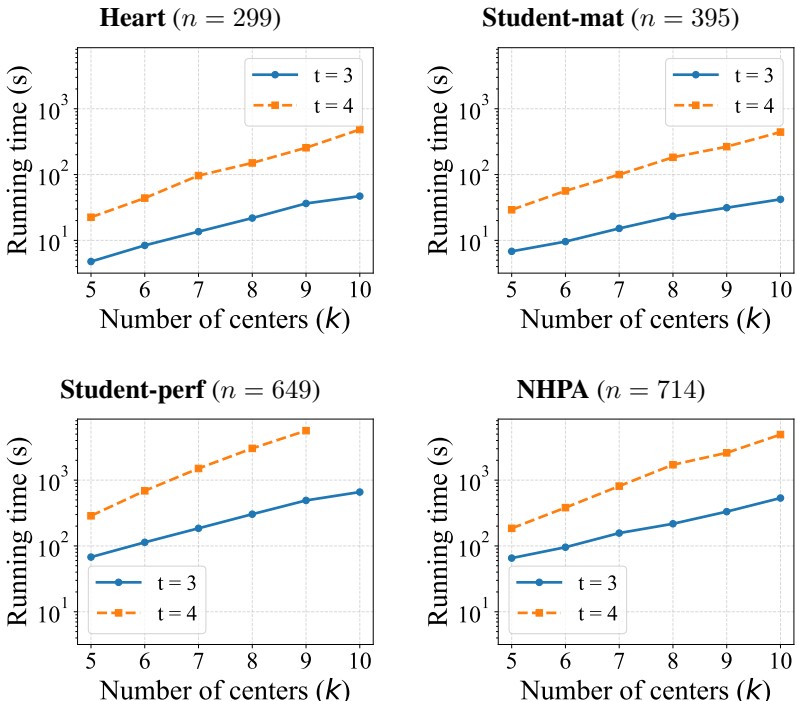

*Figure 10.* Scalability of the $O(\log k)$-approximation algorithm on real-world datasets. We report the total running time of our pipeline, *i.e.*, embedding the original metric into a binary-tree metric followed by the dynamic program that solves the problem exactly on the binary tree. For each dataset (indicated in the plot title together with the number of data points), we report runtimes for $t \in \{3, 4\}$ and $k \in \{5, \ldots, 10\}$. Experimental instances that exceeded $10^4$ seconds of execution time were terminated.

