# OpenReview forum: "Capacitated Fair-Range Clustering: Hardness and Approximation Algorithms"
_ICML.cc/2026/Conference — ICML 2026 regular_

### Official Review · Reviewer_5Xh2 · 2026-03-03

**Soundness:** 2
**Presentation:** 3
**Significance:** 3
**Originality:** 3
**Overall Recommendation:** 4
**Confidence:** 4

**Summary:**

The paper considers a fair clustering problem with the following set up: given a set of clients, a set of facilities and a metric defining respective distances, the task is to select a number $k$ of facilities such that the total median (or mean) cost for each client to its closest facility is minimized. The fairness aspect is expressed by a set of, possibly intersecting, groups of facilities and a range (upper and lower bound): a solution is only feasible (i.e., fair) if it picks a number of facilities from the respective range of every group.
While it is known to be NP-hard whether any feasible set of $k$ facilities exists, the authors show that

(1) the optimization problem remains hard to approximate even if finding any feasible set is simple

(2) for instances with a constant number of groups, they give $polylog(k)$-approximations in polynomial time and constant factor approximations running in FPT-time with respect to $k$.

Moreover, they provide an empirical evaluation of their algorithms on a set of artificial and natural benchmarks, demonstrating feasible running times on medium sized inputs.

**Compliance With Llm Reviewing Policy:**

Affirmed.

**Final Justification:**

The paper is technically sound and on an interesting clustering problem. My two main concerns in the original review were the extend of the impact of this work and some minor technical/clarity issues. In the rebuttal, the author(s) addressed the latter sufficiently. The rating of the impact is of course highly subjective, and I notice that some at least one other reviewer also raises concerns about this, while the author(s) list valid points supporting the novelty and impact. In conclusion, I think the presented results have merit to some parts of the ICML community, which leads me to weakly support acceptance. I update my score accordingly.

**Key Questions For Authors:**

1. Can you provide more motivation on why introducing capacities to the established fair ranged clustering problem is reasonable - for example is there literature on other capacitated clustering problems with fairness constraints?

2. You prove hardness for many groups and give approximation algorithms with $polylog(k)$-approximation guarantees for a constant number of groups. Are there any hardness results for constant numbers of groups or could there be exact or constant-factor-approximate polynomial-time / FPT($k$) algorithms for instances in this regime?

3. What would you like to express by oracle-feasibility? Am I missing something in that every instance with at least one feasible solution is oracle-feasible? Would one of the proposed alternatives work?

**Limitations:**

yes

**Strengths And Weaknesses:**

In the considered clustering problem, fairness is expressed not among the clients which are grouped together but among the facilities, exploring an interesting angle on fairness. It is hard to judge on how many applications would require the whole set of requirement assumed here (capacities + fairness + small cluster cost), potentially limiting the impact. That said, the stated variant seems to be a natural generalization of a well-motivated problem and the authors describe a fitting motivational setting as well. To me, the contribution is mainly in the provided approximation algorithms. Having the hardness even under the promise that there is a feasible solution at all is a bonus but in my opinion does not add too much value compared to the existing hardness result - in particular since the concept of oracle-feasibility seems to not play a role in any of the other results of the paper.

The overall presentation of the paper is fine, design decision are sufficiently well motivated and most definitions are clear.
Another strength is that the authors demonstrate practical feasibility of their algorithms by an empirical evaluation on various data sets.

All proofs are in the appendix with sketches in the main part.
I was not able to have a detailed look at all proofs due to the length of the appendix (> 15 pages of proofs).
The summaries in the main part give an overview on the most important aspects and make the claimed results reasonable.
For a good part of the results I had a closer look - for them I could verify correctness, albeit some flaws in the proofs themselves, see "Detailed Feedback". The main part itself also contains some (fixable) formal flaws, most notably the definition of "oracle-feasibility" seems off (see Detailed Feedback). I thereby trust that the claimed results stand on a solid mathematical foundation, though some details might need to be fixed.

In summary, the provided results are interesting and well-presented, but my concerns are a potentially limited impact for ICML and some technical issues. I am open to adjust my ranking based on the authors' response to my questions, in particular regarding the impact of their work.

 Detailed Feedback:

 line 132 (right): "the empirical approximation factor is below 2.7 across all instances, despite the worst-case $O(\log k)$ bound". While I appreciate the comparison between the empirical approximation factor and the mathematical guarantee, the word "despite" seems unfitting (depending on the hidden constants, it might just as well be that $O(\log k) = 2.7$ for $k=5$, indeed $\log_2(5) \approx 2.3$).

ln 149: consider including the metric $d$ in the definition of an instance, i.e., $I=(C,F,d,\mathbb{G},\ldots)$

Definition 3.1 seems a bit off. It seems that every instance I with $\mathcal{F}(I)\neq \emptyset$ is oracle-feasible. As the oracle is allowed to be instance-specific, it could just output a hard-coded feasible solution in linear time. Why not simply say that the problem is NP-hard even if some feasible solution is given as part of the input (or a promise of such a solution)?

ln 177 (right)  "is optimal" -> "is close to optimal" or similar

ln 216 (right) it might improve clarity to state that it rules out $o(D/m)$-approximations (instead of $o(D)$-approximations) and state that a satisfying assignment has cost $m$. Of course, as we can let $D$ can be an arbitrary polynomial in $n \ge m$, this preserves the claimed statement

Theorem 3.4: It is stated for Median and Means, however the proofs in the appendix seem only to consider the Median case. I imagine that the arguments easily transfer, but this could/should be mentioned

Technical overview of Section 3:
- I am missing a discussion on why the created instances are oracle-feasible (this seems to be provided in the appendix, but I consider it relevant enough to be mentioned in the main part - especially given that the only motivation for the hardness proof is to show that it is hard on such instances).
- It is claimed (in (ii)) that the distance aspect ratio is $D$, however, it seems the distance aspect ratio seems to be $4D$ (for example, the distance between assignment 10 in $G_1$ and assignment 11 in $G_2$ have distance $4D$ in your constructed instance.)
- In the construction in the appendix, the edges incident to dummy $s$ have unit weight while in the technical overview they are described with weight $D$


ln 265 (left) "An illustration of the reduction
is shown on the right." -> there's no figure on the right, are you referring to Figure 1?


It might be worth pointing out that your FPT($k$) results for constant $t$ are in fact FPT($t+k$) results, which is an even stronger statement (as far as I can see, your exponential dependency on $t$ is only for base $O(k)$, never on base $n$).

line 330 (left)  disjoint -> distinct?


line 691: bounded by -> bounded from above by ?

Also, why is $g$ allowed to map to values smaller than $1$? Because for $g(n)<1$ there can be no $g(n)$-FT$k$MED instance as the aspect ratio is always at least 1, right? In particular, Theorems C.3 and C.4 seem factually incorrect for $g(n) < 1$. In fact, I believe you even need $g(n) > 2$, see the comment for Line 796.

Line 796: It is confusing to set $g(n) \coloneqq 2D+2$ as $g$ is part of the input. It seems you instead want to set $D \coloneqq g(n)/2 - 1$, which also shows that in fact your proof only works for $g(n) \ge 4$ (so that $D\ge 1$).


line 771: the contrapositive is not that "at least $(7/8+\epsilon)$ fraction" is satisfied but that "more than $(7/8+\epsilon)$ fraction is satisfied. The proof should still work as it seems that your calculations for Claim C.8 also support this stronger statement

The above comments also apply to the proof of Theorem C.11

line 861: should $g(n)$ be $p(n)$?

---

> ### Author Rebuttal · Authors · 2026-03-28
>
> We thank the reviewer for the detailed feedback, which will help improve our work. We will correct all typographical errors in the revised manuscript. Below, we provide point-by-point responses to the concerns raised.
>
> **[Applications]** see comments to Reviewer HiSQ under [Applications].
>
> **[Definition 3.1]** Yes, you are correct, we can hard-code a feasible solution for each instance to obtain an instance-specific oracle. However, the key point is that prior hardness results do not apply in this setting, as they rely on the NP-hardness of finding a feasible solution, which is no longer an issue here. In fact, allowing an instance-specific oracle only strengthens our result: even when a feasible solution is readily available (e.g., via hardcoding), the problem remains hard to approximate, highlighting that the difficulty lies in the clustering task itself. We believe such instances are practically relevant, as domain experts often have sufficient background knowledge to identify instance-specific feasible solutions easily (see [Applications]). In contrast, earlier hardness results stemmed from the difficulty of feasibility and did not capture the complexity of the underlying clustering task. We will clarify this distinction in the revision.
>
> Please also see our response under [Oracle feasibility].
>
> **[Theorem 3.4]** Our results are stated for the k-median objective for simplicity, but they extend to the k-means objective as well. We will make this explicit in the revised manuscript.
>
> **[Technical overview of section 3]**  The created instance is oracle-feasible because for every assignment to the variables of the 3SAT formula, we get a feasible solution to the instance since the feasible solutions precisely correspond to the assignments of the variables.
>
> About the distance aspect ratio in the constructed instance, you are right, it should be 2D+2, and not D. Also, the dummy node s has unit edge weights. About your observation that $g(n) \ge 4$ is correct. We will clarify all these observations.
>
> **[Hardness for constant number of groups]** For a constant number of groups, the hardness actually follows from the capacitated version (without fairness), since in this setting, we only have 1 group. The current best hardness of approximation for the capacitated version is $1+2/e-\epsilon$ for both polytime and FPT algorithms, which itself is due to the hardness of the $k$-median problem (without fairness and capacities). A similar result $1+8/e-\epsilon$ holds for $k$-means.
>
> **[Hardness of approximation]**  Please see [Oracle feasibility] on why our hardness result has practical implications.
>
> **[Oracle feasibility]** The notion of oracle-feasibility captures instances where a feasible solution can be found in polynomial time. However, such solutions need not be optimal. These instances are particularly relevant because they bypass known hardness results, which arise from the difficulty of finding any feasible solution and therefore do not capture the complexity of the clustering objective itself. In many practical settings, feasibility is straightforward, but the real challenge lies in finding a feasible solution that also optimizes the clustering objective. Also see responses to reviewer HiSQ [Implications of hardness result] and [Novelty of hardness].
>
> We show that similar inapproximability persists even for instances where finding a feasible solution is easy, highlighting the inherent complexity of the clustering task itself. By modeling such an oracle, we capture scenarios where experts with domain knowledge can readily identify a feasible solution. While earlier inapproximability results do not apply in these settings, we show that comparable hardness guarantees still hold.
>
> **[Relevance to ICML]** There has been substantial work on algorithmic fairness and fair clustering published at venues ICML, NeurIPS, ICLR, and AAAI, and we cite several relevant papers [7–10,18]. In this context, we believe our work is both timely and relevant to the community. We provide novel hardness of approximation results that are applicable to practical instances and complement with algorithms that are effective in practically relevant regimes (small k and t). We also demonstrate empirically that our methods perform well in practice.
>
> About the question of related work on fairness in capacitated clustering, for instance, [1] studies a different notion of fairness with capacities, and [2] focuses on fair clustering with lower bounded capacity requirements rather than upper bounded capacities, as in our work. Thus, our setting and contributions are complementary to these. Also see further related work in Appendix A.
>
> [1] Fair-Capacitated Clustering. 2021. Tai Le Quy, Arjun Roy, Gunnar Friege, Eirini Ntoutsi
>
> [2] Fair Clustering with Minimum Representation Constraints. 2025. Connor Lawlessa, Oktay Gunluk.
>
> Overall, we believe our contributions align closely with ongoing research on fair clustering at ICML and related venues.

---

> > ### Author Rebuttal · Reviewer_5Xh2 · 2026-04-02
> >
> > I thank the author(s) for their detailed response. I have one follow-up question on the oracle-feasibility, as I feel that my point here was missed. I will try to elaborate:
> >
> > I understand why you are introducing oracle-feasibility and how this distinguishes your result from the known NP-hardness. I am merely concerned that the definition of oracle-feasibility is unnecessarily complicated and thus harms clarity:
> >
> > Line 208:  "We say the instance I is oracle-feasible if there exists an (instance specific) oracle $O_I$ that runs in polynomial time and outputs some feasible solution in $F(I)$."
> >
> > As the oracle is instance-specific, the above definition is just equivalent to saying $F(I) \neq \emptyset$ (as then, every algorithm that just outputs any hardcoded solution in $F(I)$ is an oracle).
> >
> > Line 211: "The $FRkMED^O$ problem, given an instance $I$ of $FRkMED$ with a promise that $I$ is oracle-feasible, asks
> > to find a feasible solution $(S, ρ)$ from $F(I)$ that minimizes ..."
> >
> > By the above, $FRkMED^O$ is simply $FRkMED$ reduced to the set of instances with $F(I)\neq \emptyset$ - which would be a much clearer definition in my opinion.
> >
> > Also, you claim that hardness still holds even when there is access to such an oracle. First, the quoted statement in Line 211 does not give this (we only have a promise of such an oracle), but I agree that your hardness proof holds even if an oracle is provided as part of the input. But again, why would you not simply assume that instead of the oracle you are directly given a feasible (but not necessarily optimal) solution as part of the input - as these two notions are equivalent for hardness results.
> >
> > So my follow-up question would be: Is there anything I am missing here in that oracle-feasibility means something more than just having a solution, and if not, is there a reason why you prefer this notion over the one of $F(I)\neq \emptyset$?

---

> > > ### Author Response · Authors · 2026-04-04
> > >
> > > We thank the reviewer for acknowledging our responses and for the follow-up questions. We apologize for the earlier misunderstanding and appreciate the insightful question, which allows us to clarify our perspective.
> > >
> > > To begin with, we agree with the reviewer’s observation that $\text{FR}k\text{Med}^O$ is equivalent to $\text{FR}k\text{Med}$ for instances where $F(I) \neq \emptyset$, and we will clarify this in the revised version of the paper.
> > >
> > > That said, our use of oracles is motivated by the following considerations.
> > >
> > > **Conceptual motivation.** Instance-specific oracles naturally model *domain experts*, who can provide feasible solutions based on implicit knowledge, heuristics, or experience, without explicitly revealing the underlying process. We do not model how such solutions are obtained, but only assume access to them. More broadly, this abstraction captures settings where feasibility arises from external or black-box mechanisms, such as legacy systems, heuristic or learning-based methods, simulation engines, or regulatory checks. While this is formally equivalent to restricting attention to instances with $F(I) \neq \emptyset$, we believe the oracle viewpoint provides a more faithful and flexible abstraction of these practical scenarios.
> > >
> > > **Complexity-theoretic perspective.** A secondary motivation is that oracles are a standard and well-established concept in computational complexity. In our context, they can be viewed as representing the existence of efficient procedures for obtaining feasible solutions. This perspective allows us to isolate and study the intrinsic complexity of the clustering objective, independent of feasibility concerns. More broadly, the oracle-based formulation provides a flexible framework for understanding the *fine-grained complexity* of the problem and enables natural extensions. For instance, one may consider uniform procedures (as opposed to instance-specific ones) and analyze the problem under such models. This is particularly relevant in structured settings, such as low-dimensional Euclidean or planar metrics, where efficient methods for finding (approximate) feasible solutions may exist, allowing one to focus on the core optimization task. Furthermore, this framework naturally captures scenarios where access to such procedures incurs a computational cost (e.g., limited query budget, time, or space). While we do not explore these extensions in this paper, similar perspectives have been successful in the past.
> > >
> > > Finally, we clarify that we do not assume the oracle is part of the input; rather, we restrict attention to instances that admit such an oracle and establish hardness within this class. Including an oracle (or a feasible solution) in the input would define a different problem than $\text{FR}k\text{Med}$. While our intuition aligns with the reviewer’s suggestion, the oracle-based formulation allows us to formalize it cleanly while preserving the original problem definition.
> > >
> > > We also note that Definition 3.1 (Line 211) already operates under a promise that a polynomial-time oracle returns a feasible solution. Replacing this with the condition $F(I)\neq \emptyset$ would still correspond to a promise setting, namely that the given instance satisfies $F(I) \neq \emptyset$.
> > >
> > > We hope this clarifies our perspective on oracle-based feasibility. We will revise the manuscript to make this point explicit, as the reviewer’s suggestion helps improve clarity.

---

### Official Review · Reviewer_5mJ1 · 2026-03-10

**Soundness:** 4
**Presentation:** 4
**Significance:** 3
**Originality:** 3
**Overall Recommendation:** 5
**Confidence:** 3

**Summary:**

The paper studies k-median and k-means clustering problems with
two additional constraints, namely capacity of facilities and
upper/lower bounds based on demographic fairness. Since these
optimization problems are NP-hard (even without the capacity and
fairness constraints), the focus is on efficient approximation
algorithms with both forms of constraints. The authors develop
general hardness results with respect to efficient approximation
and also efficient approximations under very reasonable
conditions (e.g., constant number of demographic groups). Experimental
results using synthetic and real-world datasets are included to
illustrate the performance of the algorithms.

**Compliance With Llm Reviewing Policy:**

Affirmed.

**Key Questions For Authors:**

(a) Page 3, right column, lines 2--4: These lines should mention
that the approximation algorithms mentioned in Theorem 4.5 are
randomized algorithms.

(b) Page 4, right column, statement of Theorem 3.3: The statement here
includes only the result for k-median. Is there a different version of the
theorem for k-means?

(c) It is not clear to this reviewer whether the algorithms referred to
in Theorem 4.5 can be considered (in a strict sense) to be FPT
algorithms with respect to k. The standard definition of an FPT algorithm
(which is stated as Definition B.2 in the supplement) requires that
the running time should be of the form O(f(k) n^{O(1)}), where f(k) is
a function only of k and the "n^{O(1)}" part does NOT involve k.
The running time stated in Theorem 4.5 involves the term
"O((log n)^{O(k)})", which is technically not permitted in a strict
definition of an FPT algorithm. (Of course, the correctness of the
algorithm and its running time, as explained in Section E.4 are not
affected by this comment.)

Some minor typos:

(a) Page 3, left column, line 132: "constant many" ---> "a constant number of".

(b) Page 3, right column, the line preceding "Experimental Results":
"requirements" ---> "requirement".

(c) Page 4, left column, line 167: "lie" ----> "lies".

(d) Page 4, left column, line 181: "referred as" ----> "referred to as".

**Limitations:**

Yes

**Strengths And Weaknesses:**

Strengths:

(a) Both the hardness results and the approximation algorithms presented
in the paper significantly extend current results for the k-median and
k-means clustering problems.

(b) The algorithms developed in the paper are nontrivial and they
use a nice blend of techniques from the literature (e.g., embedding
using tree metrics, use of coresets) along with new ideas (e.g.,
an exact fixed parameter tractable algorithm for the fair capacitated
versions of clustering problems).

(c) The writing style is very good. The overviews presented for the
several of the results are very helpful in understanding the ideas
behind the proofs.  the discussion on prior work on the topic is
excellent.

Weakness:

A (very) minor weakness is that the experimental results on real-world
datasets use rather small datasets. (The largest real-world dataset
is of size 714. However, larger synthetic datasets are considered.)
This reviewer believes that further research is very likely to improve
the running times of these algorithms and make them scale to larger
datasets.

---

> ### Author Rebuttal · Authors · 2026-03-28
>
> We thank the reviewer for their feedback. We will address the typographical and grammatical issues, and other suggestions in the revised manuscript. Below, we give responses to your comments.
>
> **[Datasets and scalability]** We emphasize that our algorithm is the first with theoretical guarantees for intersecting groups that can handle modestly sized instances (e.g., up to 1600 data points on synthetic datasets), whereas prior approaches largely rely on heuristics. All experiments were conducted on commodity hardware (a MacBook Air laptop with an M2 chip). For comparison, Thejaswi et al. (KDD 2022, Table 2) report runtimes exceeding 14,000 seconds for instances with n=200, t=4, and k=6, whereas our method solves a comparable instance (e.g., n = 299, t=4, k=6) in under 20 seconds (see Figure 1). At the same time, we stress that our approach is designed for moderate-scale inputs, and developing methods with theoretical guarantees that scale to truly large datasets remains an important direction for future work.
>
> **[Theorem 3.3]** The theorem also applies to k-means objective; we will update the statement accordingly in the revised manuscript.
>
> **[Theorem 4.5]** In parameterized complexity, running times of the form \((\log n)^{O(k)}\) are fixed-parameter tractable with respect to \(k\). In particular, $(\log n)^{O(k)}= 2^{O(k \log\log n)}$, and hence when $k < \log n/\log\log n$,  this is polynomial in $n$. For the other case, we have that $\log n$ is upper bounded by $O(k\log k)$, and hence, our running time is bounded by $k^{O(k)}$. Therefore, in both cases, it is bounded by $k^{O(k)} n^{O(1)}$, as desired. Such running times have been common in prior works (e.g., Thejaswi et al., 2022; Cohen-Addad et al., 2019) and is explicitly noted in Footnote 1 of Thejaswi et al. (2022). We will clarify this in the revised manuscript.

---

> > ### Author Rebuttal · Reviewer_5mJ1 · 2026-04-03
> >
> > I thank the author(s) for the careful rebuttal which addresses all my concerns. I have no further questions.

---

> > > ### Author Response · Authors · 2026-04-06
> > >
> > > We thank the reviewer for acknowledging our rebuttal responses.

---

### Official Review · Reviewer_qFWk · 2026-03-13

**Soundness:** 4
**Presentation:** 3
**Significance:** 3
**Originality:** 2
**Overall Recommendation:** 5
**Confidence:** 3

**Summary:**

The paper studies the capacitated clustering problem under demographic fairness constraints. The capacitated clustering problem aims to select k-clustering where each center has a capacity to serve individuals. In this variant, the paper assumes that there is a color of each possible cluster center and there is a lower- and upper- bound for how many centers should be selected from each color. The goal is to minimize the social cost (total distance between every individual and their assigned center).

In this model, the paper demonstrated when color count (t) is part of the input: it is NP hard to approximate within any polynomial factor. They further extend this result and demonstrate that assuming *Gap-ETH, there is no algorithm better than brute-force trying.* Assuming the number of colors is a constant (a standard assumption in this line of work), they provided O(log k) and O(log^2 k) approximate polynomial time algorithms for k-median and k-means problems respectively. Finally, they demonstrated constant factor approximation algorithm when k is a constant. They also validate their approach experimentally on both synthetic and real data. With synthetic data they compare approximation ratio and runtime of their algorithm vs brute-force algorithm. With real data, the brute-force algorithm can not terminate within reasonable time so they only demonstrate the runtime of their algorithm for different values of k (cluster size) and t (color size).

**Compliance With Llm Reviewing Policy:**

Affirmed.

**Key Questions For Authors:**

- I recommend defining distance aspect ratio very briefly.
- Motivation of forming clique-star metric is unclear. For instance Lemma 4.3 states that we should start from k-clique-star metric and convert it to tree metric. Why don’t we start from the original metric? What does break if we start from the original metric? Providing a brief motivation here would improve the discussion.
- There is a line of research in computational soical choice regarding to fair clustering. It would be nice to cite this literature too. Some references:
    - Proportionally Representative Clustering Aziz, Lee, Chu, Vollen
    - Proportionally Fair Clustering Chen, Fain, Lyu, Munagala
- Another line of research regarding to citizens' assemblies which might be also related to this paper:
    - Boosting Sortition via Proportional Representation Ebadian, Micha
    - Temporal Panel Selection in Ongoing Citizens' Assemblies Kalayci, Micha
- The experiments with synthetic data uses k=5 and t=3 which is a scenario that we select almost 1 center from each color. It is not clear whether this family really capture the difficulty of the problem. Can you explain more whether the experiments actually capture the difficulty of the problem well? Can you extend experiments for larger values of k and t?
- Do you know O(log n) and O(log^2 n) approximations are tight?
- For a non-expert person reading the paper, the contribution of the paper and novelty is hidden. For instance, it is not clear whether 3SAT reduction part is a common technique or a novel construction. It is also not mentioned whether k-clique-star metric appeared in the literature before or not. It would be nice to address this issue too.

**Limitations:**

Yes

**Strengths And Weaknesses:**

Strength:

- Demographic fairness is a concept getting popular recently. They study this notion in the context of capacitated clustering (arguably the most practically relevant clustering variant) gives the paper a strong and timely motivation.
- The paper provides a comprehensive theoretical results for the problem, covering hardness, conditional lower bounds, and approximation algorithms across different parameter regimes.

Weaknesses:

- Experiments section is weak. Experiments with synthetic data is tested against scenarios where k=5 and t=3. In these scenarios the problem is almost like the capacitated problem without fairness constraints.
- The paper can locate their results better in the literature by providing which parts are novel and which parts are following existing literature.

---

> ### Author Rebuttal · Authors · 2026-03-28
>
> We thank the reviewer for their constructive feedback. Below, we provide point-by-point responses to the issues raised in the review.
>
> **[Distance aspect ratio]** Due to space constraints, we moved the definition to Appendix B. We will clarify this in the revised manuscript.
>
> **[k-clique-star metric]** We emphasize that the k-clique-star metric is not introduced ad hoc, but is a core component of both our framework and the framework of Adamczyk et al. ESA 2019 (referred to there as $\ell$-centered instances), which achieves similar guarantees for capacitated clustering.
>
> The motivation for transforming a general metric into a $k$-clique-star metric before embedding into a tree is as follows: (i) A direct embedding into a tree incurs **$O(\log n)$ distortion**, leading to an $O(\log n)$ approximation (for k-median) even when combined with our exact tree algorithm. (ii) Our goal, however, is an **$O(\log k)$ approximation**, which is significantly stronger.
> The key insight is that the k-clique-star metric has a **highly sparse structure**: it consists of a k-clique, with all remaining vertices attached as leaves. Thus, **apart from the clique, the metric is already a tree.**
>
> We therefore embed only the k-clique into a tree, incurring $O(\log k)$ distortion restricted to these vertices. If this clique corresponds to a good approximate solution S (i.e., the centers), we can bound the overall cost increase by combining the bounded distortion within the clique, and the guarantee of S.
>
> **[References]** We will include the suggested references in the revised manuscript.
>
> **[Experiments]**  We are not entirely sure which experiment the reviewer is referring to when mentioning $k=5$ and $t=3$. If this corresponds to Table 1, then for the scalability experiments we set $\vec{\alpha} = (1)^t$ and $\vec{\beta} = (k)^t$, i.e., each group must have at least 1 and at most $k$ centers (and not at most one center from each group). Under this setting, all combinations satisfy the fairness constraints, and hence the dynamic program must consider the full space of possibilities. We emphasize that this setting is not like capacitated clustering without fairness, since these groups intersect and facilities in the intersection satisfy multiple group requirements, making the situation complex. This is a different setting compared to a setting where there are no groups (or fairness constraints).
>
> Additionally, if instead we impose stricter bounds (i.e., $\vec{\alpha} > (1)^t$ or $\vec{\beta} < (k)^t$), the number of feasible combinations decreases, leading to strictly fewer DP iterations and faster running time. Note that when $\vec{\alpha}=(2)^t$, $k=5$ and $t=3$, a feasible solution must select at least one center that belongs to multiple groups. Our experiments already capture this setting.
>
> In addition, we evaluate scalability for $t=\{3,4\}$ and $k=\{5,\dots,10\}$ in Figure 4, with additional scalability experiments in the appendix for different values of $k$ and $t$.
>
> **[Tightness of approximation results]** We do not know whether the claimed approximation ratios are tight in these regimes, and currently do not have matching lower bounds establishing tightness. In fact, the approximation ratios for capacitated versions are also not known to be tight either and have been stuck at $O(\log k)$ factor for 25 years.
>
> **[Novelty of results]** The reduction from 3SAT to (capacitated) fair-range clustering is, to the best of our knowledge, novel, and we are not aware of any prior reductions of this form for our problem. We will clarify this in the revision.
> Please see our response to reviewer HiSQ detailed comment on the novelty of our hardness construction and algorithmic results.

---

> > ### Author Rebuttal · Reviewer_qFWk · 2026-04-01
> >
> > Thanks for the detailed rebuttal. My concerns are addressed. I still suggest revising the manuscript to better state the novelty in the hardness section and give more intuition for the k-clique star metric. I keep my score and support acceptance to ICML.

---

> > > ### Author Response · Authors · 2026-04-02
> > >
> > > We thank the reviewer for their acknowledgement and support for acceptance of the paper. We will incorporate all suggested changes in the revised manuscript.

---

### Official Review · Reviewer_HiSQ · 2026-03-13

**Soundness:** 3
**Presentation:** 3
**Significance:** 2
**Originality:** 2
**Overall Recommendation:** 4
**Confidence:** 2

**Summary:**

This work studies the capacitated fair-range clustering, where in addition to the clustering objectives (k-means and k-median), the solution has to respect capacities of the facilities as well as “fair-range” constraints, where the number of centers chosen from each set is constrained to be between a lower and an upper bound. The first contribution of the authors is to show that the hardness of the problem does not come only from the “fair-range” constraints part of the problem. Second, they give a polynomial time O(log k) - approximation algorithm for the problem.

**Compliance With Llm Reviewing Policy:**

Affirmed.

**Final Justification:**

I have upgraded my score after the response. However, my confidence has now reduced after going over newer parts of the paper.
The authors give a justification as to why isolating hard parts of the problem has practical applications. I am not however completely convinced about the novelty and originality of the algorithmic results.

**Key Questions For Authors:**

NA

**Limitations:**

Yes

**Strengths And Weaknesses:**

Strengths
1. The paper is well written and the claims in the paper are correct and verifiable.
2. It increases our understanding of the problem by isolating the hard parts of the problem.

Weaknesses
1. Although I appreciate and like the contribution of showing that even in the case where fair-range constraints can be easily satisfied, the problem is hard, it has no immediate implications. Further, it is not too surprising as well.
2. The algorithms are not too unexpected either. Embedding of general metrics into tree metrics and then solving the problem has been done quite regularly. Further, usually, the embedding into a tree metric is done to get a linear-time algorithm which is not the case here.
3. The paper does not provide a comprehensive overview of the work on capacitated and fair-clustering individually. While I understand the space constraint, it would be good to have a concrete idea as to how difficult the problems are individually and how much harder they get when put together. And I say also because of the nature of the problem that it is not easy to reason and think about it due to it being a “complicated problem”.

---

> ### Author Rebuttal · Authors · 2026-03-28
>
> We thank the reviewer for their feedback. Below, we provide a point-by-point response to the questions and clarify the issues raised.
>
> **[Novelty of hardness]** To the best of our knowledge, no hardness of approximation results are known for instances where feasible solutions can be found in polynomial time. Our work provides the first such hardness construction, which crucially leverages the group structure inherent to our problem.
>
> In particular, in our construction, feasible solutions can be found easily, so the hardness arises purely from optimizing the clustering objective, rather than from satisfying constraints. This sharply contrasts with prior hardness results (Thejaswi et al. 2022), where hardness arises because finding a feasible solution is difficult, entirely bypassing the clustering aspect of the problem. Our result thus isolates and captures the intrinsic complexity of the clustering objective under structured feasibility constraints.
>
> Please also see responses to Reviewer 5Xh2 under [Oracle feasibility].
>
> **[Implication of hardness result]** Consider overlapping groups with women, master’s degree and high income. If sufficiently many candidates belong to all groups, then a feasible solution can be obtained by greedily choosing candidates that belong to all groups. However, when distances capture how well the selected candidates serve the population, such a feasible solution need not minimize the k-median cost. Hence, the computational difficulty lies not in finding any fair solution, but in finding a fair solution that also minimizes the k-median objective. Our results show that even in such settings, where feasibility is trivial, the problem remains inapproximable.
>
> Please see our response to reviewer 5Xh2 under [Oracle feasibility] and  [Practical applications] for implications of our hardness result in practical scenarios.
>
> **[Applications]** Many real-world facility location problems require equity, which is captured jointly by capacity and fairness (representation) constraints. Capacities ensure that no single facility is overloaded, promoting equitable service quality, while fairness in center selection ensures that infrastructure is distributed across different regions or demographic groups, promoting equitable access. For example, in planning hospitals, schools, disaster relief sites, or EV charging stations, each facility can serve a limited population, yet policymakers or companies should ensure that centers are not concentrated solely in high-demand or affluent areas. Thus, equity arises both in how much each facility serves and where facilities are located.Consequently, fairness is not merely an assignment-level constraint, but fundamentally influences the selection of centers, which is essential in many public infrastructure and policy-driven applications.
>
> Additionally, our results, both hardness and $O(\log k)$- $O(\log^2 k)$-approximation algorithms, also apply to uncapacitated fair-range clustering, yielding the first algorithm with theoretical guarantees that is also practical for fair-range clustering.
>
> **[Algorithmic results]** Our polynomial-time $O(\log k)$ (and $O(\log^2k)$)  approximation algorithms build on the tree embedding framework of Adamczyk et al. ESA 2019, which addresses capacitated variants without fairness constraints. We note that tree embedding based algorithms have a well-established history, originating from the seminal work of Bartal (FOCS, 1996), and have been applied to other clustering variants. That said, it is not at all clear a priori that these techniques extend naturally to (capacitated) fair-range clustering, especially in the light of strong inapproximability results this problem admits. Our main contribution is to extend and adapt this framework to simultaneously handle capacities and fairness constraints.
> In addition, we provide a polynomial-time exact algorithm for tree metrics when the number of colors is constant, a standard and well-motivated setting in fair clustering. This exact algorithm serves as a key subroutine in our adaptation of the framework. We also present FPT approximation algorithms.
>
> **[Related work]** Due to space constraints, we moved the further related work to Appendix A, where we cover the literature most relevant to our work. If the reviewer brings to our attention any missing references, we would be happy to include them in the revised version.

---

> > ### Author Rebuttal · Reviewer_HiSQ · 2026-04-04
> >
> > Thank you for the detailed rebuttal and apologies for the delay in response.
> > I have upgraded my score, but would still like to make the following points -
> > 1. I understand the example for the hard case. However I feel this is an incremental progress unless there is a work-around this as well. For example if we assume something about both the hard constraints - maybe stability for clustering and easy satisfiability for the fairness constraints, something better can be achieved.
> > 2. I still regard the tree-embedding algorithms to be incremental in nature and they are not too surprising.

---

> > > ### Author Response · Authors · 2026-04-06
> > >
> > > We thank the reviewer for their acknowledgements and follow-up comments.
> > >
> > > Regarding point (1), we are not entirely clear on the specific concern being raised. It would be helpful if the reviewer could elaborate further and clarify the concerns raised. In particular, we are unsure what is meant by “stability of clustering” or “easily satisfiable fairness constraints” in this context, and how this relates to our setting. As it stands, we are not able to fully interpret the concern or provide a detailed response. We would be happy to address this once more clarification is provided to us.
> > >
> > > We would like to emphasize that in our hardness reduction, we assume that a feasible solution can be found easily. In case, if this is meant by “easily satisfiable fairness constraints”, then, yes, our hard instances have this property, and this is what distinguishes them from the earlier hardness results, in which the intractability stemmed from finding a feasible solution. Moreover, our hardness reduction rules out non-trivial approximation factors, similar to earlier reduction, which we find novel and significantly increases our understanding of the computational complexity of the problem (as mentioned by other reviewers).
> > >
> > > We would also appreciate it if the reviewer could elaborate on the assessment that our hardness of approximation results are incremental. In particular, if there are prior results from which our claims can be easily derived or extended, we would be happy to check these results and update our manuscript accordingly.
> > >
> > > Regarding point (2), we would like to build on our earlier clarification. Tree-embedding techniques are well-understood and widely used, which explains their popularity. However, while they have been successfully applied to several clustering variants, it is not clear *a priori* that they extend to settings with fairness constraints while preserving the same approximation guarantees as in capacitated clustering.
> > >
> > > In our view, this extension is not immediate, particularly in light of inapproximability results, which hold even on **tree metrics**. Thus, even in the simplified setting induced by embeddings, the problem retains significant complexity.
> > > To our knowledge, despite extensive work on fair clustering, there are no prior algorithms that simultaneously provide strong theoretical guarantees and are practically implementable in this setting. Our work aims to achieve this combination; if similar results (or extensions) exist, we would appreciate pointers to related work and will incorporate them in the revision.

---

### Decision · Program_Chairs · 2026-04-30

**Decision:**

Accept (regular)

**Comment:**

This paper studies clustering with both capacity constraints and fair-representation range constraints, giving new hardness results and approximation algorithms for this natural fair-clustering variant. The main strength is that it both sharpens our understanding of the core computational obstacle and provides practical algorithms with provable guarantees in this regime. All reviewers ultimately support acceptance.